# EUBRL: Epistemic Uncertainty Directed Bayesian Reinforcement Learning

**Jianfei Ma, Wee Sun Lee**
National University of Singapore
{ma-j, leews}@comp.nus.edu.sg

## Abstract

At the boundary between the known and the unknown, an agent inevitably confronts the dilemma of whether to explore or to exploit. Epistemic uncertainty reflects such boundaries, representing systematic uncertainty due to limited knowledge. In this paper, we propose a Bayesian reinforcement learning (RL) algorithm, `EUBRL`, which leverages epistemic guidance to achieve principled exploration. This guidance adaptively reduces per-step regret arising from estimation errors. We establish nearly minimax-optimal regret and sample complexity guarantees for a class of sufficiently expressive priors in infinite-horizon discounted MDPs. Empirically, we evaluate `EUBRL` on tasks characterized by sparse rewards, long horizons, and stochasticity. Results demonstrate that `EUBRL` achieves superior sample efficiency, scalability, and consistency.

## 1 Introduction

In a completely unknown environment, what compels an agent to seek new knowledge? This drive is captured by the concept of exploration, which lies at the heart of reinforcement learning, from $\epsilon$-greedy to Boltzmann exploration (Sutton & Barto, 2018). Yet, these heuristics often fall short in more challenging environments, particularly those with sparse rewards, long horizons, or stochasticity. Epistemic uncertainty (Der Kiureghian & Ditlevsen, 2009) characterizes the degree of unknownness, providing a principled basis for exploration. However, it remains unclear how to most effectively leverage this uncertainty to guide learning.

Bayesian RL (Duff, 2002) provides a framework for modeling a world of uncertainty. An agent seeks to maximize cumulative rewards based on its current belief, interact with the environment, and update that belief—without knowing the true dynamics and rewards. From the agent's perspective, the world is epistemically uncertain. It must balance exploration and exploitation to find a near-optimal solution. By placing a prior over both transitions and rewards, epistemic uncertainty arises from limited data: the less familiar the agent is with a region of the environment, the more it is incentivized to explore it. Nonetheless, higher uncertainty also raises the risk of unreliable estimates. A common approach is to add the uncertainty as a "bonus" directly to the reward, a strategy known as *optimism in the face of uncertainty* (Kolter & Ng (2009); Sorg et al. (2012)). However, even small errors in the reward can propagate into an inaccurate value function, potentially resulting in unnecessary exploration and slower convergence.

When measuring the efficiency of an algorithm's exploration, metrics such as regret (Lai & Robbins, 1985; Auer et al., 2008)—the cumulative difference from the optimal value function—or sample complexity (Kakade, 2003)—the number of steps that are not $\epsilon$-optimal—are commonly used. An algorithm is said to be minimax-optimal (Lattimore & Hutter, 2012; Dann & Brunskill, 2015) if its bounds match the corresponding lower bound up to logarithmic factors. While previous works based on optimism (Kakade, 2003; Auer et al., 2008; Strehl & Littman, 2008; Kolter & Ng, 2009) or sampling (Strens, 2000; Osband et al., 2013) have been shown to achieve strong theoretical guarantees, their use of uncertainty quantification and empirical evaluation of exploration capabilities remain limited, leaving room for improvement in practical problems, particularly those requiring sustained and efficient exploration.

In this paper, we propose `EUBRL`, an **E**pistemic **U**ncertainty directed **B**ayesian **RL** algorithm for principled exploration. We use probabilistic inference to model epistemic uncertainty as part of the

agent's objective. This approach guides the agent to explore regions with high epistemic uncertainty while mitigating the impact of unreliable reward estimates. Our contributions are both theoretical and empirical:

- We prove that `EUBRL` is nearly minimax-optimal in both regret and sample complexity for infinite-horizon discounted MDPs, with epistemic uncertainty adaptively reducing the per-step regret.
- We instantiate prior-dependent bounds and demonstrate their applications using conjugate priors.
- We demonstrate that `EUBRL` excels across diverse tasks with sparse rewards, long horizons, and stochasticity, achieving superior sample efficiency, scalability, and consistency.

To the best of our knowledge, our result is the first to achieve nearly minimax-optimal sample complexity in infinite-horizon discounted MDPs, without assuming the existence of a generative model (Gheshlaghi Azar et al., 2013).

## 2 PRELIMINARY

An infinite-horizon discounted Markov Decision Process (MDP) is defined by a tuple $\mathcal{M} = (\mathcal{S}, \mathcal{A}, P, r, \gamma)$, where $\mathcal{S}$ and $\mathcal{A}$ are the state and action spaces, both of finite cardinality, denoted by $S$ and $A$, respectively, $P$ the transition kernel $P(\cdot|s, a)$, $r$ the expected reward function, and $\gamma \in [0, 1)$ the discount factor. We assume the source distribution of rewards has bounded support in $[0, R_{\max}]$. A policy $\pi$ is a mapping from states to actions, whose performance is measured by the expected return $V^\pi(s) = \mathbb{E}\left[\sum_{l=0}^{\infty} \gamma^l r(s_{t+l}, a_{t+l})|s_t = s, \pi\right]$. The goal is to find the optimal policy $\pi^\star(s) = \arg\max_\pi V^\pi(s)$, $\forall s \in \mathcal{S}$, whose value function is $V^\star(s)$. We denote the maximum value function as $V_\gamma^\uparrow := \frac{R_{\max}}{1-\gamma}$ and, whenever applicable, $V_H^\uparrow := H R_{\max}$ for its finite-horizon counterpart.

### 2.1 BAYESIAN RL

We consider the Bayes-adaptive MDP (BAMDP) (Duff, 2002) to model the agent's learning process. Given a prior $b_0$, the uncertainty over both the transitions and rewards—or equivalently, possible MDPs—is explicitly modeled. A policy is Bayes-optimal if it maximizes expected return in the belief-augmented state space $(s, b) \in \mathcal{S} \times \mathcal{B}$, where $b$ is a belief over MDPs. Formally, it solves the Bellman optimality equation under the posterior predictive transition model $P_b$ and posterior predictive mean reward $r_b$ of the corresponding BAMDP. However, this solution requires full Bayesian planning (Poupart et al., 2006; Kolter & Ng, 2009; Sorg et al., 2012), which is computationally expensive and typically intractable because the belief-augmented state space can be too large to enumerate, and the belief must be recalculated every time a new state is encountered. Consequently, agents generally must approximate Bayes optimality. One simple yet effective alternative is the mean MDP (Kolter & Ng, 2009; Sorg et al., 2012), which fixes the belief during planning. This is essentially equivalent to an MDP $(\mathcal{S}, \mathcal{A}, P_b, r_b, \gamma)$ given a belief $b$. When indexed by time, $b_t$ refers to the posterior given all data up to time $t$. By solving the corresponding mean MDP, we obtain a policy $\pi_t$ derived from the subjective value function $V^t$ and its objective evaluation in the underlying MDP, $V^{\pi_t}$. Our goal is to find the optimal policy $\pi^\star$ by repeatedly solving the mean MDP during interaction, alternating between posterior learning and policy optimization.

### 2.2 METRICS FOR EXPLORATION

We define per-step regret as $\mathbf{\Delta}_t := V^\star(s_t) - V^{\pi_t}(s_t)$. Regret and sample complexity are defined from different angles:

$$\textbf{Regret} \quad \sum_{t=1}^{T} \mathbf{\Delta}_t, \qquad\qquad \textbf{Sample Complexity} \quad \sum_{t=1}^{\infty} \mathbf{1}(\mathbf{\Delta}_t > \epsilon).$$

Low regret does not imply low sample complexity, and vice versa. Regret, which is more cost-oriented, focuses on how much you lose while learning, whereas sample complexity cares about learning efficiency, i.e., the number of samples needed to learn properly.

A lower bound is best achievable for regret, $\widetilde{\Omega}\left(\frac{\sqrt{SAT}}{(1-\gamma)^{1.5}}\right)$ (He et al., 2021), and for sample complexity, $\widetilde{\Omega}\left(\left(\frac{SA}{\epsilon^2(1-\gamma)^3}\right)\log\frac{1}{\delta}\right)$ (Strehl et al., 2009; Lattimore & Hutter, 2012). When an algorithm's upper bound matches these lower bounds up to logarithmic factors, it is considered minimax-optimal; if it holds only in the asymptotic regime of large $T$ or small $\epsilon$, it is considered nearly minimax-optimal.

# 3 METHODOLOGY

## 3.1 EPISTEMIC UNCERTAINTY

Learning an imperfect model is of epistemic nature, where the uncertainty arises from a lack of knowledge and is, in principle, reducible by observing more data. In general, epistemic uncertainty captures the degree of disagreement in the belief—a distribution over model parameters $\mathbf{w}$, e.g. the transition probability vector or the reward location and scale. For example, for transitions, we have:

$$\mathcal{E}_T(s,a) = f \circ g(P_b(s'|s,a)) - \mathbb{E}_{\mathbf{w}\sim b(\mathbf{w})}\left[f \circ g(P(s'|s,a,\mathbf{w}))\right],$$

for some functions $f$ and $g$ that take a scalar or a distribution as input. Intuitively, it reflects the likelihoods' deviation from the "average". When $f(x) = -x^2, g(p) = \mathbb{E}_{p(x)}[x]$, it corresponds to the variance $\mathrm{Var}_{\mathbf{w}\sim b}\left(\mathbb{E}[s'|s,a,\mathbf{w}]\right)$ (Kendall & Gal, 2017). When $f(p) = \mathcal{H}(p), g(p) = p$, it corresponds to mutual information $\mathrm{MI}(s,a) = \mathcal{H}\left(P_b(s'|s,a)\right) - \mathbb{E}_{b(\mathbf{w})}\left[\mathcal{H}\left(P(s'|s,a,\mathbf{w})\right)\right]$ (Hüllermeier & Waegeman, 2021). A similar argument holds for rewards $\mathcal{E}_R(s,a)$ by substituting $s'$ with $r$.

We adopt a generalized formulation of epistemic uncertainty to integrate both sources:

$$\mathcal{E}_b(s,a) \coloneqq h(\mathcal{E}_T(s,a), \mathcal{E}_R(s,a)).$$

In this paper, we consider $h(x,y) = \eta(\sqrt{x} + \sqrt{y})$, where $\eta$ is a scaling factor.

## 3.2 PROBABILISTIC INFERENCE AND EPISTEMIC GUIDANCE

Traditionally, RL aims to maximize cumulative reward. A pivotal question is how to account for epistemic uncertainty in this objective to balance exploration and exploitation. One common approach is optimism-based methods, modifying rewards with an additive bonus $\tilde{r} = r_b + \eta r_{\mathrm{bonus}}$. However, this can be misleading when $r_b$ is uncertain. In this regard, we utilize probabilistic inference to model epistemic uncertainty directly in the objective, disentangling exploration and exploitation and making it more resilient to unreliable reward estimates (see discussion in Appendix A).

Probabilistic inference has a rich history in decision-making (Todorov, 2008; Toussaint, 2009; Levine, 2018). It has been shown that standard RL can be formulated as an inference problem by introducing a binary "optimality" random variable $\mathcal{O}_t$:

$$\max_\pi \mathbb{E}_{P(\tau)}\left[\log \prod_{t=0}^\infty P\left(\mathcal{O}_t = 1|s_t, a_t\right)\right]$$

with an exponential transformation $P(\mathcal{O}_t = 1|s_t, a_t) \propto \exp\left(r(s_t, a_t)\right)$ and $\tau$ denoting a trajectory.

We introduce the notion of *probability of uncertainty*, representing the degree of uncertainty, governed by a binary "uncertainty" variable $U_t$. Marginalizing over this variable, we obtain a lower bound on per-step likelihood:

$$\log P\left(\mathcal{O}_t = 1|s_t, a_t\right) = \log \mathbb{E}_{U_t}\left[P\left(\mathcal{O}_t = 1|s_t, a_t, U_t\right)|s_t, a_t\right]$$
$$\geq \mathbb{E}_{U_t}\left[\log P\left(\mathcal{O}_t = 1|s_t, a_t, U_t\right)|s_t, a_t\right].$$

Note that since $U_t$ is binary, if we adopt the same exponential transformation, which intensifies the higher uncertainty, we obtain the *epistemically guided reward*:

$$r_b^{\mathrm{EUBRL}}(s,a) \coloneqq (1 - P(U = 1|s,a))\, r_b(s,a) + P(U = 1|s,a)\mathcal{E}_b(s,a).$$

Intuitively, when uncertain, EUBRL focuses more on epistemic uncertainty, as an intrinsic reward, encouraging exploration; when confident, it is more committed to exploiting what has been learned. We call this kind of behavior *epistemic guidance*. The probability of uncertainty $P(U = 1|s,a)$ naturally disentangles the two ends, being more indifferent to reward estimates in the early stage and becoming more committed as evidence accumulates. Although its definition can vary, $P(U = 1 \mid s,a)$ must reflect epistemic uncertainty. For simplicity, we choose $P(U = 1 \mid s,a) = \frac{\mathcal{E}_b(s,a)}{\mathcal{E}_{\mathrm{max}}}$, where $\mathcal{E}_{\mathrm{max}}$ is typically determined by the prior, and adopt the shorthand $P_U(s,a)$ henceforth.

### 3.3 Algorithm

The full Algorithm 1 is shown below. It alternates between posterior updating and policy learning. The belief update is in closed form due to conjugacy. Moreover, both epistemic uncertainty and posterior predictives can be expressed in closed form. Once a belief is updated, we derive the posterior predictive transition model $P_b$ and posterior predictive mean reward $r_b$, from which we construct an MDP $\mathcal{M} = (\mathcal{S}, \mathcal{A}, P_b, r_b^{\text{EUBRL}}, \gamma)$, where $r_b^{\text{EUBRL}}$ is the epistemically guided reward. The MDP is solved using value iteration (Sutton & Barto, 2018). For large-scale settings, approximate methods such as tree search may be required (Guez et al., 2012).

---

**Algorithm 1** EUBRL

> **Input:** prior $b_0$, scaling factor $\eta$, discount factor $\gamma$ or horizon $H$
> $s_0 \sim P(s_0)$
> $\pi_0 \leftarrow$ Solve $\mathcal{M} = (\mathcal{S}, \mathcal{A}, P_{b_0}, r_{b_0}^{\text{EUBRL}}, \gamma)$
> **for** $t \leftarrow 0$ to $T - 1$ **do**
>     Act: $a_t = \pi_t(s_t)$
>     Interact: $s_{t+1}, r_t \sim P(s_{t+1}, r_t | s_t, a_t)$
>     Update belief: $b_{t+1} \leftarrow$ Belief Update$(s_{t+1}, r_t)$
>     **if** time to reset **then**
>         $s_{t+1} \leftarrow s \sim P(s_0)$
>     **end if**
>     **if** time to update policy **then**
>         $\pi_{t+1} \leftarrow$ Solve $\mathcal{M} = (\mathcal{S}, \mathcal{A}, P_{b_{t+1}}, r_{b_{t+1}}^{\text{EUBRL}}, \gamma)$
>     **end if**
> **end for**

---

Notably, our algorithm is a general recipe that depends on the combination of reset and policy update, and generalizes to both infinite-horizon discounted MDPs and finite-horizon episodic MDPs. For finite-horizon episodic MDPs, the policy is updated and the episode is reset every $H$ steps. For infinite-horizon discounted MDPs, the policy is updated at every step and there is no reset.

Unlike prior optimism-based approaches, our method features simplicity, avoiding intricate designs like knowness (Kakade, 2003; Strehl & Littman, 2008; Sorg et al., 2012) and tailored bonuses (Azar et al., 2017; Dann et al., 2019), and can, in principle, work with any Bayesian model. Compared to Bayesian RL (Kolter & Ng, 2009; Sorg et al., 2012), the key difference is our reward formulation.

## 4 Theoretical Analysis

In this section, we aim to answer two key questions: **(1)** What is the role of epistemic guidance, and **(2)** How efficient is the exploration for EUBRL. Theoretically, an algorithm is considered efficient in exploration if it achieves sublinear regret or polynomial sample complexity, the latter being known as PAC-MDP (Kakade, 2003; Strehl & Littman, 2008). Many algorithms have been shown to be efficient in exploration. In particular, (He et al., 2021) has shown that achieving nearly minimax-optimality for regret is possible in infinite-horizon discounted MDPs. However, it is not clear whether this holds for sample complexity. We show that EUBRL achieves both nearly minimax-optimal regret and sample complexity, providing insight into how epistemic guidance adaptively reduces per-step regret. Our analysis builds on the concept of quasi-optimism (Lee & Oh, 2025), which established minimax-optimality in finite-horizon episodic MDPs—yet its applicability to infinite-horizon MDPs remains unexplored. Unlike finite-horizon episodic MDPs, which feature clear separation into episodes and allow backward induction over horizons, infinite-horizon MDPs are more involved due to the coupling of trajectories and the stationarity of value functions.

We commence our analysis from a frequentist perspective, with $\mathcal{E}_b(s, a) = \frac{1}{\sqrt{N^t(s,a)}}$, where $N^t(s, a)$ denotes the number of visits to $(s, a)$ prior to the $t$-th step. Both the transition and reward are estimated using maximum likelihood estimators, corresponding to the empirical means $\hat{P}$ and $\hat{r}$. We then extend and instantiate this framework to the Bayesian setting, deriving prior-dependent bounds for a specific class of priors and examining their applications to commonly used priors[1].

---

[1] All results presented here remain valid for finite-horizon episodic MDPs.

### 4.1 REGRET DECOMPOSITION

The per-step regret, a central quantity in both regret and sample complexity, can be decomposed as follows:

$$V^\star(s) - V^{\pi_t}(s) = \underbrace{V^\star(s) - \widetilde{V}^t(s)}_{\text{Quasi-optimism}} + \underbrace{\widetilde{V}^t(s) - V^t(s)}_{\text{Complexity}} + \underbrace{V^t(s) - V^{\pi_t}}_{\text{Accuracy}},$$

Each term is defined by its purpose and consequence: **Quasi-optimism** is a weaker form of optimism than typically assumed in theoretical works, allowing more relaxed requirements on algorithmic components. **Complexity** stems from introducing $\widetilde{V}^t(s)$, an auxiliary value function that ensures quasi-optimism and adapts the previous analysis to our framework. **Accuracy** reflects the extent to which the agent's internal model diverges from the true environment, serving as a key indicator of how effectively an agent explores the environment and builds its model.

We aim to bound these terms individually. To do so, we define an auxiliary sequence $\{\lambda_t\}_{t=1}^\infty$ where $\lambda_t \in (0, 1], \forall t \in \mathbb{N}$. These values, derived from Freedman's inequality (Freedman, 1975) and refined by Lee & Oh (2025), are used to bound quasi-optimism and accuracy. Furthermore, to bound the accuracy term, we require $V^t(s) = \mathcal{O}(V_\gamma^\uparrow)$ to ensure the agent's subjective value function remains bounded. Without loss of generality, we set the positive multiplicative constant $C = 1$. In addition, for notational simplicity, we denote $\Phi_t := R_{\max}\lambda_t$.

Consequently, by invoking Corollaries 2–3 and Lemma 14, we bound the terms with respect to the epistemic uncertainty $\mathcal{E}_b$, the maximum value function $V_\gamma^\uparrow$, and the auxiliary sequence $\{\lambda_t\}_{t=1}^\infty$, combining them to yield:

**Theorem 1** (Bound of Per-step Regret). *For infinite-horizon discounted MDPs, with probability at least $1 - \delta$, it holds that for all $s \in \mathcal{S}, t \in \mathbb{N}$,*

$$V^\star(s) - V^{\pi_t}(s) \le \left(\frac{9}{2} - \mathfrak{R}^t(s)\right)\lambda_t V_\gamma^\uparrow + 2J_\gamma^t(s) + \mathcal{O}\left(\Phi_t\left(1 + \frac{\Phi_t}{V_\gamma^\uparrow}\right)\right),$$

*where we define the following as **Epistemic Resistance***

$$\mathfrak{R}^t(s) := 2P_U^t\left(s, \pi_t(s)\right) + \frac{9}{7}P_U^t\left(s, \pi^\star(s)\right).$$

Here, $J_\gamma^t(s)$, a Bellman-like function involving error terms, is bounded in Lemmas 20–21 by addressing the challenge of trajectory coupling through a simple observation that grouping terms by time step creates a martingale difference sequence (Durrett, 2019).

Intuitively, *epistemic resistance* adaptively reduces the per-step regret based on the unfamiliarity of the actions chosen by the current policy and the optimal policy. The greater the uncertainty of these actions, the lower the per-step regret, which highlights the critical role of epistemic uncertainty. In fact, the reduction of total regret is even more pronounced, as indicated by the following bound.

**Lemma 1** (Lower Bound of Epistemic Resistance). *Given a uniform $\lambda_t = \lambda, \forall t \in \mathbb{N}$, it holds that*

$$\sum_{t=1}^T \mathfrak{R}^t(s_t)\lambda_t V_\gamma^\uparrow \ge \frac{23R_{max}}{7(1-\gamma)}\left(\frac{2}{\mathcal{E}_{max}}\left(\sqrt{T} - 1\right) + 1\right)\lambda,$$

*for any $T \in \mathbb{N}$.*

That is, the regret bound of our method must be no worse than that without epistemic guidance.

### 4.2 FREQUENTIST BOUNDS

**Theorem 2.** *For infinite-horizon discounted MDPs, for any fixed $T \in \mathbb{N}$, with probability at least $1 - \delta$, it holds that*

$$Regret(T) \le \widetilde{\mathcal{O}}\left(\frac{\sqrt{SAT}}{(1-\gamma)^{1.5}} + \frac{S^2A}{(1-\gamma)^2}\right).$$

Note that when $T \geq \frac{S^3 A}{1-\gamma}$, the regret matches the lower bound, implying nearly minimax-optimality. This result improves the state-of-the-art frequentist bound from (He et al., 2021).

**Theorem 3.** *Let $\epsilon \in (0, V_\gamma^\uparrow]$, $\delta \in (0, 1]$, and $\mathcal{M} = (\mathcal{S}, \mathcal{A}, P, r, \gamma)$ be any MDP. There exists an input $\eta = \mathcal{E}_{max} \Upsilon + R_{max} \sqrt{m}$, such that if EUBRL is executed on MDP $\mathcal{M}$, with probability at least $1 - \delta$, $V^{\pi_t}(s_t) \geq V^\star(s_t) - \epsilon$ is true for all but $\widetilde{\mathcal{O}}\left( \left( \frac{SA}{\epsilon^2(1-\gamma)^3} + \frac{S^2 A}{\epsilon(1-\gamma)^2} \right) \log \frac{1}{\delta} \right)$ steps.*

Here, $\Upsilon$ is a function of $(S, A, \delta, \lambda, V_\gamma^\uparrow)$, and $m$ a critical point where the complexity term is sufficiently bounded (see Table 4). Note that when $\epsilon \in \left[ 0, \frac{1}{S(1-\gamma)} \right]$, the sample complexity matches the lower bound, implying nearly minimax-optimality. This result, to the best of our knowledge, is the first online algorithm to achieve such a bound without assuming a generative model (Gheshlaghi Azar et al., 2013).

### 4.3 FROM FREQUENTIST TO BAYESIAN

In this section, we instantiate prior-dependent bounds and demonstrate their applications using conjugate priors, building upon the frequentist results. To bridge the gap between the frequentist and Bayesian settings, we formalize key properties of priors that ensure expressivity while facilitating the analysis of regret and sample complexity.

Due to space limitations, we only outline the conceptual ideas here and defer the details to Definitions 13–16. A prior is *decomposable* if the difference between the posterior predictive and the ground truth can be decomposed into a frequentist bound and a prior bias; a prior is *weakly informative* if the posterior predictive is close to the empirical mean. If the prior is *uniform*, the prior bias admits a universal constant, and if *bounded*, the prior predictive mean of the reward is bounded.

**Definition 1.** Let $\mathfrak{C}$ be defined by the class of *decomposable* or *weakly informative* priors whose rate of epistemic uncertainty is $\Theta\left( \frac{1}{\sqrt{n}} \right)$.

This class can be quite expressive, as it can be either correlated or independent over state-actions, including hierarchical priors (Neal, 2012).

**Theorem 4.** *Let $\mathcal{M} = (\mathcal{S}, \mathcal{A}, P, r, \gamma)$ be any MDP. For any prior $b_0 \in \mathfrak{C}$, there exists an instance of EUBRL such that, when executed on $\mathcal{M}$, it achieves, with probability at least $1 - \delta$, a prior-dependent bound on regret, or alternatively, on sample complexity, depending on the choice of $\eta$. If, furthermore, $b$ is assumed to be uniform and bounded, these bounds are nearly minimax-optimal.*

The significance of this result is that, depending on the priors, we can achieve even tighter bounds. In addition, it can be nearly minimax-optimal despite dependence on the prior. We demonstrate its applications with the two most commonly used priors: Dirichlet for transitions and Normal or Normal-Gamma for rewards.

**Corollary 1.** *Let $b_0$ denote the joint distribution consisting of a Dirichlet prior $\mathrm{Dir}(\alpha \mathbf{1}_{S \times 1})$ on the transition probability vector and a Normal prior $\mathcal{N}(\mu_0, \frac{1}{\tau_0})$ on the mean reward with known precision $\tau$ for all $(s, a) \in \mathcal{S} \times \mathcal{A}$. Then $b_0 \in \mathfrak{C}$ and is uniform and bounded, and hence achieves nearly minimax-optimality when used with EUBRL.*

To the best of our knowledge, this is the first nearly minimax-optimality result in the Bayesian setting. Nevertheless, we also find that EUBRL can fail in certain special cases.

**Proposition 1.** *For a Normal-Gamma prior $\mathcal{NG}(\mu_0, \lambda_0, \alpha_0, \beta_0)$, there exists a parameterization and an MDP such that $\exists t \in \mathbb{N}$ for which quasi-optimism does not hold.*

Intuitively, since the epistemic uncertainty of the Normal-Gamma depends on the sample variance, when the environment is deterministic or nearly deterministic, this term can be zero, leading to a degenerate rate of epistemic uncertainty that violates the requirement of quasi-optimism. Nonetheless, this issue can be alleviated by using sufficiently small prior parameters to control prior bias.

When the prior is misspecified such that the initial epistemic uncertainty is very low, the method may also encounter difficulties and could fail to converge.

Table 1: Results on **Chain** environment. The average return and standard error are computed across 500 random seeds, with each run consisting of 1000 steps.

| Algorithm | Average Return | SE |
|---|---|---|
| PSRL | 3158 | 31 |
| RMAX | 3090 | 36 |
| BEETLE | 1754 | - |
| BOSS | 3003 | - |
| Mean-MDP | 3078 | 49 |
| BEB | 3430 | - |
| MBIE-EB | 3462 | - |
| VBRB | 3465 | $20 \sim 50$ |
| **EUBRL** | **3473** | **16** |

Table 2: Summary of tasks. For Loop, we denote $L$ as the number of loops and $L_k$ as the $k$-th loop; for DeepSea, $N$ is the side length; for LazyChain, $N$ is the balanced length. "D" stands for deterministic and "S" stochastic.

| TASK | $S$ | $A$ | $r$ | | TYPE |
|---|---|---|---|---|---|
| CHAIN | 5 | 2 | $10\,\mathbf{1}_{(s'=5)}$ | | S |
| LOOP | $4L+1$ | 2 | $2\,\mathbf{1}_{(s'=1 \text{ AND } L_1)}$ | $+$ | D |
| | | | $\mathbf{1}_{(s'=1 \text{ AND } L_{k\neq1})}$ | | |
| DEEPSEA | $N \times N$ | 2 | $\mathbf{1}_{(s'=(N,N))}$ | $-$ | D |
| | | | $\mathbf{1}_{(a=\text{RIGHT})}\frac{0.01}{N}$ | | |
| DEEPSEA | | | $\mathcal{N}(1,1)\mathbf{1}_{(s'=(N,N))}$ | $+$ | S |
| | | | $\mathcal{N}(0,1)\mathbf{1}_{(s'=(N,1))}$ | $-$ | |
| | | | $\mathbf{1}_{(a=\text{RIGHT})}\frac{0.01}{N}$ | | |
| LAZYCHAIN | $2N+1$ | 3 | $(2N-1)\,\mathbf{1}_{(s'=\text{RIGHT})}$ | $+$ | S, D |
| | | | $(N-1)\,\mathbf{1}_{(s'=\text{LEFT})}$ | $+$ | |
| | | | $0\,\mathbf{1}_{(a=\text{DO NOTHING})}$ | $-$ | |
| | | | $1\,\mathbf{1}_{(\text{OTHERWISE})}$ | | |

**Theorem 5** (Prior Misspecification). *Let $\eta = 1$. There exists an MDP $\mathcal{M}$, a prior $b_0$, an accuracy level $\epsilon_0 > 0$, and a confidence level $\delta_0 \in (0,1]$ such that, with probability greater than $1 - \delta_0$,*

$$V^{\pi_t}(s_t) < V^\star(s_t) - \epsilon_0$$

*will hold for an unbounded number of time steps.*

We construct a two-armed bandit with a misspecified prior such that the prior is confidently wrong and produces low epistemic uncertainty, leading to repeated commitment to the suboptimal arm with high probability.

In other words, this counterexample highlights the vital importance of the scaling factor $\eta$ and the priors in enabling efficient exploration.

## 5 EXPERIMENTS

In this section, we aim to measure the exploration capabilities of `EUBRL` on tasks with sparse rewards, long horizons, and stochasticity. We focus on sample efficiency, scalability, and consistency, as reflected by metrics such as the number of steps or episodes required to fully solve a task, scalability with respect to problem size, and success rate. We find that `EUBRL` generally matches or outperforms previous principled algorithms, with the advantage increasing as problem size grows. We compare `EUBRL` with both frequentist and Bayesian methods. Our benchmarks include well-known standard tasks in the Bayesian literature, Chain and Loop (Strens, 2000)—the former highly stochastic, the latter deterministic and emphasizing state-space structure—as well as more complex environments: we study DeepSea (Osband et al., 2019b;a) and design LazyChain, both featuring sparse rewards, long horizons, and deterministic and stochastic variants. A concise summary is provided in Table 2, with detailed descriptions available in Appendix C.1.

**Baselines** Frequentist algorithms based on optimism include RMAX (Brafman & Tennenholtz, 2002), which assigns unknown state-action pairs the maximum possible reward, and MBIE-EB (Strehl & Littman, 2008), which uses Hoeffding's inequality to derive a reward bonus $r_{\text{bonus}}^t = \frac{1}{\sqrt{n^t(s,a)}}$, where $n^t(s,a)$ is the number of visits up to and including the $t$-th step. Bayesian methods are flexible in incorporating prior knowledge. Sampling-based methods include PSRL (Strens, 2000; Osband et al., 2013), which acts optimally with respect to a model sampled from the belief, and BOSS, which samples multiple models and solves a merged MDP. Optimism-based Bayesian methods include BEB (Kolter & Ng, 2009), which is based on the mean-MDP with an additive bonus $r_{\text{bonus}}^t = \frac{1}{1+n^t(s,a)+\mathbf{1}^\top\boldsymbol{\alpha}}$, where $\boldsymbol{\alpha}$ are the prior parameters of the Dirichlet distribution; however, it

Table 3: Results on **Loop** environment of 2 Loops. The average return and standard error are computed across 500 random seeds, with each run consisting of 1000 steps.

| Algorithm | Average Return | SE |
|-----------|----------------|-----|
| PSRL | 377 | 1 |
| RMAX | 394 | 0 |
| Mean-MDP | 233 | 3.4 |
| BEB | 386 | 0 |
| EUBRL | **395** | 0.04 |

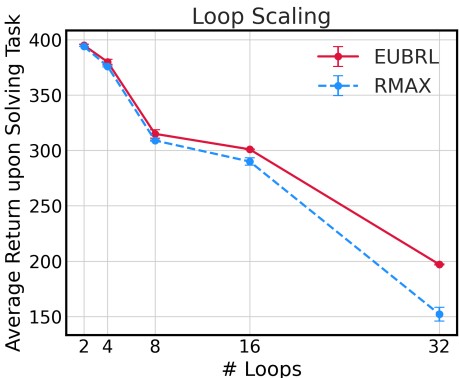

Figure 1: Scaling of number of loops, leading to more sparsity and structural difficulty. Averaged over 500 random seeds.

assumes the reward function is known, and VBRB (Sorg et al., 2012), which is based on the variance in the belief over both reward and transition. VBRB is similar to ours but, being tailored only to variance, does not include epistemic guidance. Moreover, classic Bayesian methods are worth comparing: BEETLE (Poupart et al., 2006) provides an analytic solution to BAMDP, where the Bayes-optimal policy implicitly trades off exploration and exploitation, and Mean-MDP (Poupart et al., 2006; Kolter & Ng, 2009; Sorg et al., 2012) approximates BAMDP without any reward bonus.

**Results**  As shown in Table 1 and 3, in Chain and Loop, `EUBRL` not only outperforms all relevant baselines but also exhibits low variability. Notably, Mean-MDP consistently performs subpar, highlighting the importance of a reward bonus for sustained and efficient exploration. Furthermore, we evaluated `EUBRL` against RMAX—whose inductive bias favors deterministic environments—on Loop by increasing the number of loops, which leads to more sparsity in the state space; surprisingly, even with a perfect prior—so that RMAX knows the transitions and rewards after experiencing them—it scales less favorably than `EUBRL`. This suggests that the priors in Bayesian methods may have a smoothing effect, enabling more scalable performance in sparse environments.

Another standard benchmark is DeepSea, a hard-exploration problem where a dithering strategy may require an exponentially large amount of data, and the success probability decays exponentially as the problem size increases (Osband et al., 2019b). As depicted in Figure 2, for the deterministic variant, most methods are able to solve the task. Surprisingly, PSRL (or Thompson sampling in the bandit setting)—despite being an effective sampling strategy for exploration—does not scale well as the problem size increases, likely because their sampling is excessively frequent, causing unnecessary exploration and fluctuations near convergence. Additionally, BEB, a Bayesian method, also based on the mean MDP, does not leverage any posterior information in the reward bonus, making it less flexible across different environments and resulting in slower convergence. On the other hand, the stochastic variant is a harder problem, with stochastic rewards, additional competing sources, and randomized transitions. We consider two priors for `EUBRL`: one more conservative and the other more exploratory, denoted as `EUBRL+`. We find that our method is more sample-efficient, requiring fewer steps to solve the task, and more scalable and consistent. Notably, `EUBRL+` perfectly solves the task without failure—a result not observed in previous works.

Lastly, we introduce a new environment called LazyChain, which involves long horizons, sparse rewards, and myopia. The only positive rewards are at the two ends, with the left end being suboptimal. Starting from the middle of the chain, the agent can move at a per-step cost or choose to `do nothing`, incurring no cost but making it impossible to obtain higher rewards. Even upon reaching the left end, the agent receives a positive immediate reward, yet the cumulative reward remains zero, hindering effective credit assignment. To succeed, the agent must sufficiently explore the chain to reach both ends and overcome the myopia. Results in Figure 3 show that `EUBRL` consistently outperforms other methods, exhibiting better sample efficiency and scalability, even under heavy noise injection in the transitions. A comparison with DeepSea is provided in Remark 1.

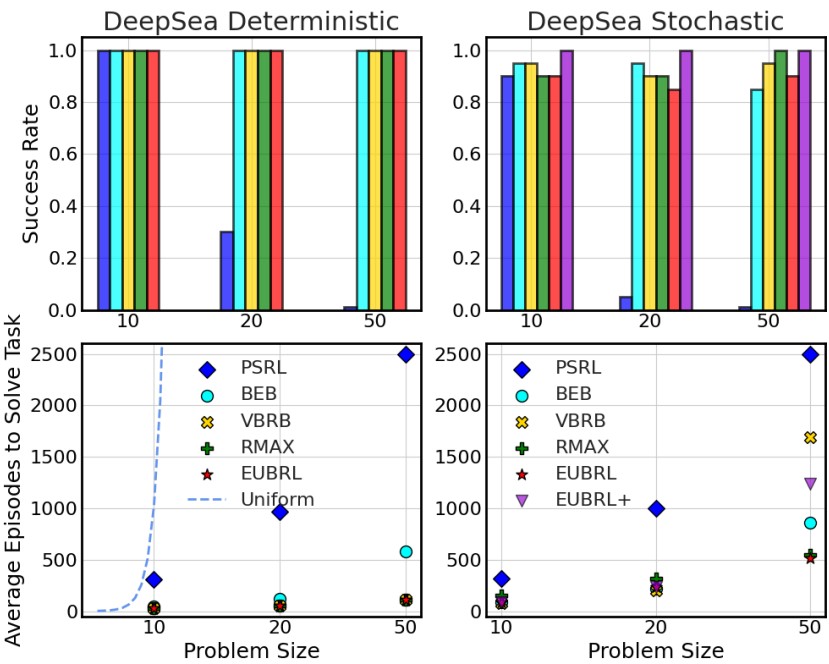

Figure 2: Success rate and average episodes to solve task, reported for both deterministic and stochastic variants over different problem sizes ($S = N \times N$). Averaged over 20 random seeds.

**Prior Selection** We discuss the selection and incorporation of priors. We use independent Dirichlet (Dearden et al., 1999) and Normal-Gamma priors for transitions and rewards. Although Proposition 1 suggests that Normal-Gamma may be degenerate, we find that it adapts more smoothly to changes. Since we have diverse stochastic environments, the sample variance can help inform epistemic uncertainty. In contrast, Normal-Normal assumes the precision $\tau$ (the reciprocal of variance) is fixed, entirely disregarding variability.

Moreover, in practice—for example, in navigation tasks where per-step transitions are similar across different states—it is beneficial to use a *tied* prior, maintaining a single global Dirichlet prior that is aggregated and shared among all states. As shown in Figure 3, EUBRL (Tied Prior) indeed reduces the number of samples required for convergence and achieves a higher overall success rate.

From Section 3.1, we know that the definition of epistemic uncertainty is not unique. Beyond variance, one information-theoretic measure is mutual information, which quantifies the reduction in uncertainty after collecting additional evidence. We find that mutual information is more exploratory than variance. As shown in Figure 3, EUBRL (MI), although taking slightly more steps, achieves the highest overall success rate.

## 6 RELATED WORKS

**Bayesian RL** Bayesian RL maintains a posterior over uncertain quantities and uses this uncertainty to guide policy selection. From bandits (Thompson, 1933; Kaufmann et al., 2012) to MDPs (Dearden et al., 1999; Strens, 2000; Kolter & Ng, 2009), this idea enables effective exploration strategies that are otherwise impossible with simple dithering. BAMDP (Duff, 2002) formally represents uncertainty over MDPs by augmenting the state with beliefs, allowing derivation of a Bayes-optimal policy, though it is generally intractable. Approximate methods include mean-MDP (Poupart et al., 2006), sparse sampling (Wang et al., 2005), and approximate inference (Wang et al., 2012). Despite being Bayesian, most of these works make limited use of uncertainty quantification, without fully leveraging the posterior. VBRB (Sorg et al., 2012) employs variance similar to ours; however, it is motivated by Chebyshev's inequality and lacks epistemic guidance.

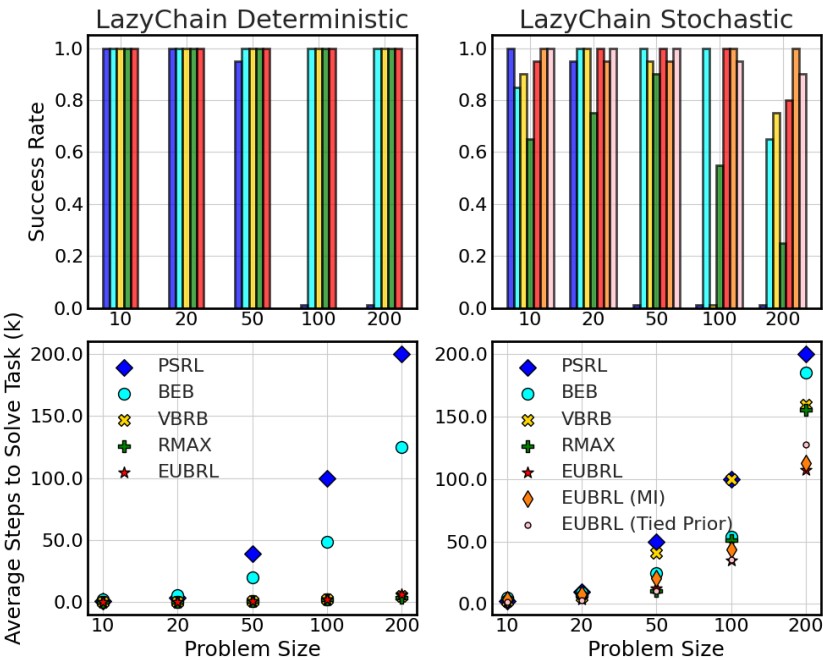

Figure 3: Success rate and average steps to solve task, reported for both deterministic and stochastic variants over different problem sizes ($S = 2N + 1$). Averaged over 20 random seeds.

**Provably Efficient RL**  The idea of knownness (Kakade, 2003), combined with Hoeffding's inequality, underlies the PAC-MDP (Strehl & Littman, 2008; Strehl et al., 2009) and PAC-BAMDP (Kolter & Ng, 2009; Araya-López et al., 2012) guarantees, though these bounds are loose compared to our frequentist results. He et al. (2021) shows that nearly minimax-optimal regret is achievable in infinite-horizon discounted MDPs, but whether similar sample complexity guarantees hold remains unclear. Although several works achieve nearly minimax-optimal regret (Azar et al., 2017) or sample complexity (Dann & Brunskill, 2015; Dann et al., 2019) in the finite-horizon setting using refined concentration bounds (Lee & Oh, 2025), the infinite-horizon setting is generally more challenging due to trajectory coupling and value function stationarity.

**Uncertainty Quantification**  Epistemic uncertainty—arising from knowledge gaps—has deep roots in cognition: it elicits curiosity (Kidd & Hayden, 2015) and enhances memory for surprising information (Kang et al., 2009). Mathematically, this manifests as surprise or disagreement in one's belief, captured by mutual information (Hüllermeier & Waegeman, 2021) or variance (Kendall & Gal, 2017). Despite this rich foundation, formal study of epistemic uncertainty in Bayesian RL remains limited. As an intrinsic motivation emerging naturally from Bayesian inference, epistemic uncertainty offers a versatile, principled approach to learning—yet a critical open question remains: how to capture it across multiple hierarchies, minimizing the need of hand-crafted rewards.

## 7 CONCLUSION

In this paper, we introduce EUBRL, a Bayesian RL algorithm that leverages epistemically guided rewards for principled exploration. The epistemic guidance naturally disentangles exploration and exploitation and adaptively reduces per-step regret. Theoretically, we prove that EUBRL achieves nearly minimax-optimal regret and sample complexity for a class of sufficiently expressive priors, with concrete instantiations for the two most commonly used priors. Empirical results demonstrate the strong exploration capabilities of EUBRL on tasks with sparse rewards, long horizons, and stochasticity, achieving superior sample efficiency, scalability, and consistency. Scalable epistemic uncertainty estimation and efficient Bayesian planning with function approximation remain open and promising directions for future research. See discussion in Appendix B.3.

ACKNOWLEDGMENTS

This research is supported by the Ministry of Digital Development and Information (MDDI) under the Singapore Global AI Visiting Professorship Program (Award No. AIVP-2024-002).

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

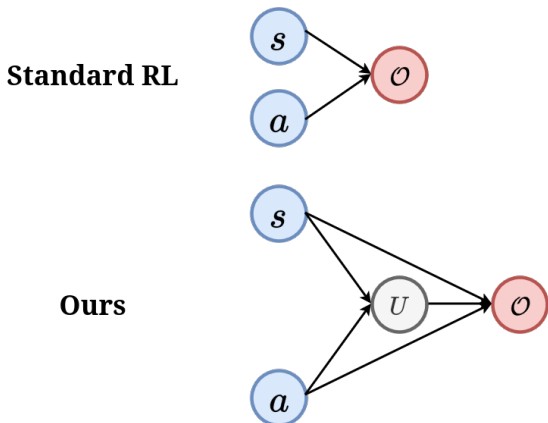

Figure 4: Comparison between standard RL and our formulation as represented by probabilistic graphical models (PGMs). We introduce the variable of "uncertainty" $U$, which partitions the optimality $\mathcal{O}$ into distinct cases: one is when certain, the other is when uncertain.

## A  CONTROL AS INFERENCE

In this section, we expand the discussion on the motivation for using probabilistic inference to incorporate epistemic uncertainty into the agent's learning objective. In decision-making problems, an agent seeks to accumulate knowledge through interactions with the environment. Standard RL achieves this by maximizing accumulated rewards; however, this approach often becomes overly exploitative of known rewards and ignorant of unknown potential rewards. Consequently, the learning process is not truly *active*, as it is driven primarily by observed rewards rather than by the uncertainty surrounding them.

By contrast, UCB-based bonuses introduce a proxy for "uncertainty," such as the inverse of visit counts, to promote more active exploration. However, these methods often fail to distinguish whether a reward estimate is reliable or not when the uncertainty is high. This limitation motivates us to disentangle exploration from exploitation via epistemically guided rewards, which are anchored in probabilistic inference.

Specifically, probabilistic graphical models provide a principled framework to condition on additional hidden variables (e.g., $U$). As illustrated in Figure 4, this conditioning allows us to partition optimality ($\mathcal{O}$) into two cases (or more, depending on modeling choices): one in which the agent is uncertain, corresponding to epistemic uncertainty, and one in which it is certain, corresponding to reward estimates. This mechanism thereby focuses more on exploration when epistemic uncertainty is high, while shifting to exploitation when the agent is confident in its existing knowledge.

## B  MODEL SPECIFICATION

### B.1  HIERARCHICAL REWARD MODEL

In our setting, $r(s, a)$ denotes the expected reward, which is a deterministic function. The source distribution can be modeled as either $P(r|s, a)$ or $P(r|s, a, s')$. Notably, the former is independent of the outcome following the action, making it less expressive and potentially misrepresenting the underlying generative process. For example, in many scenarios, the reward is meaningful only when feedback is received, which depends on the next state $s'$. In such cases, the reward distribution $P(r|s, a)$ effectively becomes a mixture distribution:

$$P(r|s, a) = \mathbb{E}_{P(s'|s, a)}[P(r|s, a, s')].$$

If we were to use simple distributions e.g. Normal distribution to model $P(r|s, a)$, then it will underrepresent the more complex true mixture distribution. On the other hand, if $P(r|s, a)$ can be

sufficiently represented by a class of distributions, its feedback-dependent counterpart $P(r|s, a, s')$ must be representable with similar complexity. Moreover, learning $P(r|s, a, s')$ and the transition model $P(s'|s, a)$ is disentangled, allowing computation to be appropriately allocated. In contrast, learning $P(r|s, a)$ implicitly learns $P(s'|s, a)$, which makes learning more difficult and prevents reuse of existing knowledge. Therefore, we adopt modeling $P(r|s, a, s')$ in our implementation and aggregate epistemic uncertainty in reward as follows:

$$\mathcal{E}_R(s, a) := \mathbb{E}_{P_b(s'|s,a)} \left[ \mathcal{E}_R(s, a, s') \right].$$

## B.2 EPISTEMIC UNCERTAINTY

Variance-based epistemic uncertainty involves evaluating the expectation of either the transition model or the reward model. However, unlike real-valued distributions, it is meaningless to take an expectation over a categorical distribution, since the numeric representation does not correspond to the actual categories. To address this, we encode the state using one-hot encoding when calculating epistemic uncertainty for the Dirichlet distribution. The exact formulations are provided in Section H.2.

The maximum epistemic uncertainty $\mathcal{E}_{\max}$, as required by the epistemically guided reward, is fully determined by the priors and therefore does not introduce any additional degrees of freedom. Although epistemic uncertainty is generally non-increasing, for certain priors it may not be strictly so—for example, the Normal-Gamma prior for the reward, which incorporates the sample variance. Therefore, it is safer to track the maximum epistemic uncertainty throughout learning, ensuring that $P_U$ remains well-defined.

For mutual information-based epistemic uncertainty, it is worth noting that a closed-form solution exists for the Dirichlet distribution. By leveraging the known moments of the Dirichlet, we obtain:

$$\mathrm{MI}_b(s, a) = E_{b(\mathbf{w})}[D_{\mathrm{KL}} \left( P(s'|s, a, \mathbf{w}) \| P_b(s'|s, a) \right)]$$
$$= \sum_i \frac{\alpha_i}{\alpha_0} \left[ \psi(\alpha_i + 1) - \psi(\alpha_0 + 1) - \log \frac{\alpha_i}{\alpha_0} \right],$$

where $\psi$ is the digamma function.

## B.3 DISCUSSION ON FUNCTION APPROXIMATION

Although our work is not intended for immediate deployment with deep function approximators, we believe that the conceptual idea of epistemically guided reward could inspire future research. The main barriers we foresee are the efficiency and quality of both epistemic uncertainty estimation and Bayesian planning. Furthermore, we discuss how our theoretical results could be extended beyond the current setting.

**Epistemic Uncertainty Estimation** Existing approximate Bayesian methods can be leveraged for this purpose, e.g., deep ensembles (Lakshminarayanan et al., 2017), Bayes by Backprop (Blundell et al., 2015), MC dropout (Gal & Ghahramani, 2016), Bayesian hypernets (Krueger et al., 2017; Dwaracherla et al., 2020), or Langevin Monte Carlo (Welling & Teh, 2011; Ishfaq et al., 2024). However, these methods typically require multiple models or samples, which can significantly hinder computational efficiency, particularly when integrated with Bayesian planning. Meanwhile, some efforts (Fan & Ming, 2021; Sasso et al., 2023) have been made to scale PSRL to continuous state and action spaces using Bayesian linear regression, offering a lighter-weight alternative. When epistemic uncertainty is quantified via mutual information, active learning (Gal et al., 2017) and Bayesian experimental design (Rainforth et al., 2024) provide tractable estimators. In particular, Sukhija et al. (2023) model the dynamics with Gaussian processes (Williams & Rasmussen, 2006), deriving a tractable upper bound on mutual information, which demonstrates strong zero-shot capability on novel tasks. Nevertheless, a key open question remains: can we construct a well-calibrated epistemic uncertainty estimator that does not rely heavily on sampling?

**Bayesian Planning** One can leverage sparse and smart sampling strategies, such as employing lazy sampling (Guez et al., 2012) or reusing a set of pre-sampled models (Wang et al., 2012; Lu & Van Roy, 2017). Additionally, trajectories can be simulated in latent space using Monte Carlo

estimates, similar to Dreamer (Hafner et al., 2020), while policy optimization can be performed via reparameterized policy gradients (Heess et al., 2015). Despite these advances, computational efficiency remains suboptimal, and the accuracy of the solution is unclear.

**Approximation Error**  When approximation is involved, a natural question arises: do the theoretical results in the paper still hold? We argue that our theoretical results remain meaningful in the approximate setting. For example, when an exact MDP solver is not available, we may need to resort to an approximate one, whose solution we denote by $\hat{\pi}_t$. The per-step regret can then be re-expressed as follows:

$$V^\star(s) - V^{\hat{\pi}_t}(s) = V^\star(s) - V^{\pi_t}(s) + \underbrace{V^{\pi_t}(s) - V^{\hat{\pi}_t}(s)}_{\text{Approximation Error}}.$$

As shown, the per-step regret can be decomposed into two components: the first part corresponds to the results established in the paper, while the second part captures the quality of the approximate solver. If a reasonably good approximation is available, we can derive similar regret and sample complexity bounds, albeit with an additional term reflecting the approximation error.

This property is appealing because, given the same solver, our method will always outperform alternative approaches. It also implies that EUBRL can be integrated with existing solvers, such as tree search-based or rollout-based methods, as discussed previously.

Overall, scalable epistemic uncertainty estimation and efficient Bayesian planning with function approximation remain open and promising directions for future research, providing a fundamental basis for enabling active exploration in increasingly complex environments. Moreover, a more comprehensive theoretical analysis of the approximate setting is another direction worth pursuing.

## C  EXPERIMENTAL SETUP

### C.1  ENVIRONMENTS

Our benchmarks include standard tasks from the Bayesian literature, Chain and Loop, originally introduced by (Strens, 2000) as testbeds for smart exploration. These tasks are challenging due to the presence of multiple suboptimal policies and the fact that the optimal policy produces rewards that are distant. The two tasks differ in their characteristics: in Chain, transitions are highly stochastic and actions may not always have the intended effect, whereas Loop, although deterministic, has a state-space structure that makes exploration difficult.

In addition to these, we evaluate more complex and larger-scale tasks. In particular, DeepSea (Osband et al., 2019b), as implemented in Bsuite (Osband et al., 2020), requires deep exploration as a core capability for RL agents. It has been shown that dithering strategies such as $\epsilon$-greedy or Boltzmann exploration fail to achieve deep exploration and may require exponentially many episodes to learn anything meaningful. In contrast, optimistic or randomized strategies can solve the task in the optimal number of episodes (Osband et al., 2019b). DeepSea has two variants: one deterministic, where both transitions and rewards are predictable, and one stochastic, which introduces noise to transitions and rewards and includes competing reward sources, making it more challenging. Notably, no algorithm has been shown to consistently succeed across different problem sizes in the stochastic setting.

Furthermore, we introduce a new environment called LazyChain, where effective credit assignment is bottlenecked by exploration, and efficient exploration is hindered by myopia. In this environment, seemingly promising immediate rewards may not provide meaningful feedback for learning the value function. LazyChain also has deterministic and stochastic variants. Details of all these environments are provided below.

**Chain**  (Strens, 2000) is a five-state problem with two abstract actions, {left, right}. Each action has a probability of "slipping", which causes it to produce the opposite effect. The optimal behavior is to always choose the action right; however, if the other action is chosen, the agent will be reset to the leftmost state.

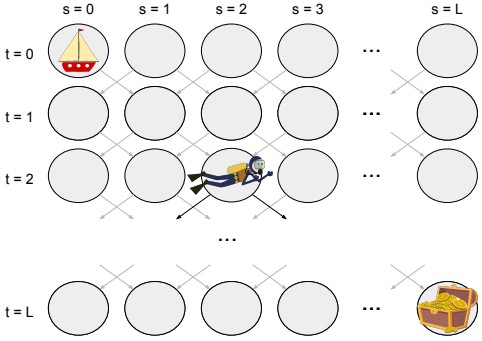

Figure 5: Visualization of DeepSea.

**Loop**   (Strens, 2000) was originally proposed as a two-loop problem jointly connected at a single start state. It is a deterministic environment consisting of nine states and two actions, $\{a, b\}$. Repeatedly taking action $a$ causes traversal of the right loop, yielding a reward of 1 when the agent returns to the start state, while repeatedly taking action $b$ traverses the left loop, yielding a reward of 2. If a single step is missed in the left loop, the agent is immediately sent back to the start state, while all other actions taken within the right loop continue the traversal. This problem makes exploration difficult due to the structure of state-space and sparse rewards.

We generalize this environment to more than two loops, with only one loop yielding the optimal reward. The size of the action space expands to match the number of loops, with each action corresponding to entering a specific loop. Similarly, any incorrect action taken within the optimal loop causes a reset to the start state, while the other loops continue their traversal.

**Deep Sea**   (Osband et al., 2019b;a) consists of $S = N \times N$ states and two actions, $\{\text{left}, \text{right}\}$, which move the agent diagonally and terminate exactly after $N$ steps per episode (see Figure 5). In the deterministic variant, there is only one positive reward at the bottom-right cell, representing a treasure. However, there is a per-step cost of $r = -\frac{0.01}{N}$ to discourage the agent from moving in that direction, while no cost is incurred when moving the other way. In the stochastic variant, a "bad" transition is generated with probability $\frac{1}{N}$ when moving toward the treasure, introducing a high degree of uncertainty from which the agent may not recover. Moreover, additive noise $\mathcal{N}(0, 1)$ is applied to the rewards at both the treasure cell and the bottom-left cell.

**Lazy Chain**   is a balanced chain, with the initial state in the middle and the two halves of equal length (see Figure 6). The only positive rewards are at the two ends; however, the left end is suboptimal. The per-step cost for moving along the chain is $r = -1$. To test the exploration capability of an algorithm, we introduce another action, `do nothing`, which leaves the agent in the current state with no cost incurred. Notably, although the left end gives a positive reward, accounting for the cost to reach it, the cumulative reward will be zero. This makes credit assignment even harder, as it fails to distinguish between "worth nothing because nothing happened" and "worth nothing because a lot of bad things and one good thing happened." Without proper exploration, an agent may either converge confidently on the suboptimal path or be unable to receive any positive rewards, eventually leading to a myopic solution—remaining in the same state. There are two variants of this environment: the deterministic version, in which transitions are fully predictable, and the stochastic version, in which actions may be flipped at each time step with probability $p = 0.2$. In addition, the agent is reset to the middle of the chain whenever either end is reached.

*Remark* 1. Notably, unlike DeepSea, where the probability of error decays and is limited to the "right" action with no adverse effect, LazyChain maintains a constant error probability, affects all movement actions, and produces opposite effects, making larger problem sizes increasingly challenging, potentially exponentially so. Additionally, DeepSea is episodic, terminating exactly in $N$ steps, whereas LazyChain may take an arbitrarily long time to explore the chain, even indefinitely, if one keeps choosing `do nothing`. Moreover, LazyChain has more suboptimal solutions, since

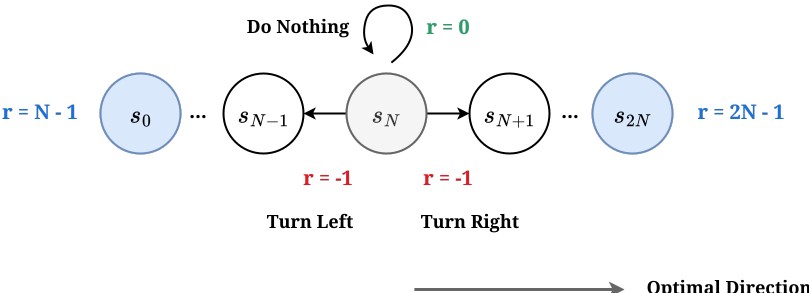

Figure 6: Visualization of LazyChain.

DeepSea is isomorphic to the right half of the chain of LazyChain, where any action that does not head toward the treasure is considered a failure.

## C.2 METRICS

The metrics we choose reflect the key aspects of interest: sample efficiency, scalability, and consistency. These are measured by the number of steps or episodes required to solve a task, scalability with respect to problem size, and consistency in terms of success rate. Additionally, we report the average return whenever applicable.

**Average Return** The cumulative return up to a given time, averaged across all random seeds.

**Average Steps / Episodes to Solve Task** The number of steps or episodes required to solve the task, averaged across all random seeds.

**Success Rate** The proportion of successful runs among all runs.

An algorithm is considered successful only if it matches the optimal policy exactly for consecutive episodes, which makes this a stricter condition for task completion. Meanwhile, an algorithm is halted if it succeeds in solving the task, fails to solve the task completely, or exceeds the maximum allowable steps $T_{\max}$. For instance, Chain and Loop are stopped exactly at the specified number of environment steps. In contrast, for DeepSea, we use a limit of $T_{\max} = 50 \cdot N^2$, where $N$ is the side length, and for LazyChain, we use $T_{\max} = 1000 \cdot N$, where $N$ is a balanced length. This choice both facilitates computational efficiency and allows evaluation of exploration under constraints. In other words, this threshold can be viewed as a penalty for any algorithm that fails the task, inflating the metric for the average steps or episodes required to solve the task.

In sparse reward environments, we expect an efficient algorithm to achieve faster convergence, as indicated by the average number of steps or episodes required to solve the task, and higher consistency, as reflected by the success rate. However, in practice, algorithms may exhibit a trade-off between convergence and consistency.

## C.3 HYPERPARAMETERS

To ensure fairness, all hyperparameters are tuned via line search for best performance for each method. Moreover, we ensure identical priors and modeling choices across Bayesian methods. Scaling factors are adjusted per algorithm, as they are algorithm-dependent.

We model rewards using Normal-Gamma model $\mathcal{NG}(\mu_0, \lambda_0, \alpha_0, \beta_0)$. We set $\mu_0 = 0$ and $\alpha_0 = 2$. Furthermore, we impose $\lambda_0 = \beta_0$, resulting in a single tunable parameter. This configuration leads to an initial epistemic uncertainty $\mathcal{E}_R(s, a, s') = 1, \forall (s, a, s') \in \mathcal{S} \times \mathcal{A} \times \mathcal{S}$. In addition, we model transitions using Dirichlet-Multinomial model. The Dirichlet prior is parameterized by a single parameter $\alpha$, yielding to a uniform prior $\mathrm{Dir}(\mathbf{1}^\top \alpha)$. A higher value of $\alpha$ indicates a stronger prior belief, whereas a lower value makes the prior less informative. On top of that, we have a tunable

scaling factor $\eta$. For LazyChain, since the maximum reward scales with the state space size, the scaling factor $\eta$ has to be adaptive.

For Bayesian algorithms like EUBRL, VBRB, PSRL, and BEB, we perform a sweep over Dirichlet parameter $\alpha \in \{1.0, 1 \times 10^{-1}, 1 \times 10^{-2}, 1 \times 10^{-3}, 1 \times 10^{-4}, 1 \times 10^{-5}, 1 \times 10^{-6}, 1 \times 10^{-8}\}$ for transitions, and $\beta_0 \in \{1.0, 5 \times 10^{-1}, 1 \times 10^{-1}, 5 \times 10^{-2}, 1 \times 10^{-2}, 1 \times 10^{-3}, 1 \times 10^{-4}\}$ for rewards. All algorithms use the same prior. The scaling factor is tuned individually for each algorithm, in proportion to the maximum rewards or the state space size. We find that VBRB and BEB perform better when the scaling factor is smaller, whereas EUBRL benefits from a slightly larger value. It is worth noting that PSRL does not require a scaling factor; it promotes exploration by repeatedly sampling from the belief. However, this sampling mechanism can lead to numerical instability when the Dirichlet parameter is too small (e.g. below the threshold $1 \times 10^{-3}$). For fairness, we therefore clip the Dirichlet parameter at this threshold, which only affects state–action pairs for which no observations have been made. Moreover, since BEB does not utilize the posterior to compute the reward bonus, it is unnecessary to adopt the reward modeling discussed in the previous section; we follow the same practice as in the original paper.

For the frequentist algorithm RMAX, the maximum reward is assumed to be known to the algorithm, and we tune the knowness parameter $m \in \{1, 3, 5, 10, 20\}$. We find that a small knowness parameter is generally beneficial in deterministic environment, but not well suited for stochastic environments, which require sustained exploration. A moderate value of the knowness parameter performs best in stochastic environment; otherwise, the algorithm spends excessive time exploring, which scales as $\mathcal{O}(mSA)$.

The discount factor is kept the same for all algorithms. However, its value may differ across tasks, depending on the task horizon. For smaller-scale tasks like Chain and Loop, we choose $\gamma = 0.95$, which trades off accuracy and computational efficiency. For long-horizon tasks, we choose $\gamma = 0.99$ for DeepSea, while $\gamma = 0.999$ for LazyChain, since LazyChain may require more time to explore due to the inaction and stronger stochasticity.

### C.4 CODE ACCESSIBILITY

The code supporting this work is available at https://github.com/MagiFeeney/EUBRL, licensed under GPLv3 to ensure the software and its derivatives remain free and open-source. Our code is based on https://github.com/dustinvtran/bayesrl, which we substantially revised, extended, and optimized while aligning the baselines with the original implementation. We appreciate the efforts of the authors in releasing their code.

## D NOTATIONS AND LOGARITHMIC TERMS

In this section, we summarize the notation and logarithmic terms used exclusively for the analysis of both finite- and infinite-horizon settings. To begin with, we denote $PV(s, a) := \mathbb{E}_{P(s'|s,a)}[V(s')]$ for any distribution $P$ and function $V$.

### D.1 FINITE-HORIZON EPISODIC MDPS

Whenever we refer to $k$ or $h$, they denote the episode and a particular step of that episode, respectively. We define $\Delta_h(V)(s, a) := V_h(s) - PV_{h+1}(s, a)$. Furthermore, we define $N^k(s, a)$ as the number of visits to $(s, a)$ before the $k$-th episode, and $n_h^k(s, a)$ as the number of visits up to and including the $h$-th step of the $k$-th episode. It is useful to define stopping time $\nu_k$ as follows:

$$\nu^k := \begin{cases} \min\{h \in [H] : n_h^k(s_h^k, a_h^k) > 2N^k(s_h^k, a_h^k)\}, & \text{if } h \text{ exists.} \\ H + 1, & \text{otherwise.} \end{cases}$$

Intuitively, the stopping time is the first time step within an episode at which the number of visits has more than doubled compared to before the episode.

We define the error terms $\beta^k(s, a)$ associated with $\frac{1}{N^k(s,a)}$:

$$\beta^k(s, a) := P_U^k(s, a)\eta^k \mathcal{E}^k(s, a) + \frac{1}{N^k(s, a)} \left( \frac{(4 - P_U^k(s))V_H^\uparrow \ell_1}{\lambda_k} + 30 V_H^\uparrow S\ell_{3,k}(s, a) \right).$$

Table 4: Summary of logarithmic terms and additional notations used in the analysis, with shorthand notation. Each term is specialized for finite- and infinite-horizon MDPs, where symbol $\square$ takes either episodes or steps as input.

| Shorthand | Finite-horizon episodic MDPs | Infinite-horizon discounted MDPs |
|---|---|---|
| $\ell_1$ | $\log\left(\frac{24HSA}{\delta}\right)$ | $\log\left(\frac{24SA}{\delta}\right)$ |
| $\ell_{2,\square}$ | $\log\left(1+\frac{kH}{SA}\right)$ | $\log\left(1+\frac{t}{SA}\right)$ |
| $\ell_{3,\square}$ | $\log\left(\frac{12SA(1+\log kH)}{\delta}\right)$ | $\log\left(\frac{12SA(1+\log t)}{\delta}\right)$ |
| $\ell_{3,\square}(s,a)$ | $\log\left(\frac{12SA(1+\log N^k(s,a))}{\delta}\right)$ | $\log\left(\frac{12SA(1+\log N^t(s,a))}{\delta}\right)$ |
| $\ell_{4,\square}$ | $\log\frac{12H}{\delta}$ | $\log\frac{12t}{\delta}$ |
| $\ell_{5,\epsilon}$ | $\log\left(1+280B(\epsilon)H\right)$ | $\log\left(1+140B(\epsilon)\right)$ |
| $\ell_{6,\epsilon}$ | $\log\log\frac{V_H^\uparrow e}{\epsilon}$ | $\log\log\frac{V_\gamma^\uparrow e}{\epsilon(1-\gamma)}$ |
| $B(\epsilon)$ | $\frac{R_{\max}^2 H^2 \ell_1}{\epsilon^2}+\frac{R_{\max}HS(2\ell_1+\ell_{6,\epsilon})}{\epsilon}$ | $\frac{R_{\max}^2\ell_1}{\epsilon^2(1-\gamma)^3}+\frac{R_{\max}S(2\ell_1+\ell_{6,\epsilon})}{\epsilon(1-\gamma)^2}$ |
| $m_\square$ | $\frac{V_H^{\uparrow 2}}{R_{\max}^2 \lambda_k^2}$ | $\frac{V_\gamma^{\uparrow 2}}{R_{\max}^2 \lambda_t^2}$ |
| $m$ | $\frac{V_H^{\uparrow 2}}{R_{\max}^2 \lambda}$ | $\frac{V_\gamma^{\uparrow 2}}{R_{\max}^2 \lambda}$ |
| $\Upsilon_\square$ | $\frac{7V_H^\uparrow \ell_1}{\lambda_k}$ | $\frac{7V_\gamma^\uparrow \ell_1}{\lambda}$ |
| $\Upsilon$ | $\frac{7V_H^\uparrow \ell_1}{\lambda}$ | $\frac{7V_\gamma^\uparrow \ell_1}{\lambda}$ |
| $\eta^\square$ | $\mathcal{E}_{\max}\Upsilon^k + R_{\max}\sqrt{m_k}$ | $\mathcal{E}_{\max}\Upsilon^t + R_{\max}\sqrt{m_t}$ |
| $\eta$ | $\mathcal{E}_{\max}\Upsilon + R_{\max}\sqrt{m}$ | $\mathcal{E}_{\max}\Upsilon + R_{\max}\sqrt{m}$ |

Based on this, we define a Bellman-like function $J_h^k(s)$, which uses $\beta^k(s,a)$ as rewards while following the latest policy $\pi_h^k$ and the true transition $P$:

$$J_{H+1}^k(s) := 0$$
$$J_h^k(s) := \min\left\{\beta^k(s,\pi_h^k(s)) + PJ_{h+1}^k(s,\pi_h^k(s)), V_H^\uparrow\right\} \text{ for } h \in [H].$$

### D.2 INFINITE-HORIZON DISCOUNTED MDPs

Whenever we refer to $t$, it denotes the time step, which is the same as the environment step. Analogously, we have $\Delta_\gamma(V)(s,a) := V(s) - \gamma PV(s,a)$. In addition, we define $N^t(s,a)$ as the number of visits to $(s,a)$ prior to the $t$-th step[2], and $n^t(s,a)$ as the number of visits up to and including the $t$-th step.

The stopping time $\nu_t$ is defined as follows:

$$\nu_t := \begin{cases} \min\{\tau \in [t,T] : n^\tau(s_\tau,a_\tau) > 2N^t(s_\tau,a_\tau)\}, & \text{if } \tau \text{ exists.} \\ T+1, & \text{otherwise.} \end{cases}$$

The main difference from the finite-horizon setting is that, for every time step $t$, we look ahead to determine a stopping time $\nu_t$, rather than relying on a single stopping time that applies to an entire episode.

Similar to the finite-horizon setting, we define the error terms $\beta^t(s,a)$ associated with $\frac{1}{N^t(s,a)}$:

$$\beta^t(s,a) := P_U^t(s,a)\eta^t \mathcal{E}^t(s,a) + \frac{1}{N^t(s,a)}\left(\frac{(4-P_U^t(s))V_\gamma^\uparrow \ell_1}{\lambda_t} + 30V_H^\uparrow S\ell_{3,t}(s,a)\right).$$

---

[2] Although at first sight this definition differs from the standard visit count $n^t(s,a)$, they are essentially equivalent up to a one-step shift.

And $J_\gamma^t(s)$ uses $\beta^t(s, a)$ as rewards while following the latest policy $\pi^t$ and the true transition $P$, with discounting:

$$J_\gamma^t(s) \coloneqq \min \left\{ \beta^t(s, \pi^t(s)) + \gamma P J_\gamma^t(s, \pi^t(s)), V_\gamma^\uparrow \right\}.$$

## E  HIGH PROBABILITY EVENTS

In this section, we outline high probability events that are basis of the analysis henceforth. Let $\{\lambda_k\}_{k=1}^\infty$ be a sequence of real numbers with $\lambda_k \in (0, 1], \forall k \in \mathbb{N}$ for finite-horizon episodic MDPs. Analogously, we have $\{\lambda_t\}_{t=1}^\infty$ for infinite-horizon discounted MDPs. They arise from Freedman's inequality (Freedman, 1975), and has been enhanced recently by (Lee & Oh, 2025).

### E.1  REGRET ANALYSIS

#### E.1.1  FINITE-HORIZON EPISODIC MDPS

$$\mathbf{A}_1 \coloneqq \left\{ \left| (\hat{P}^k - P) V_{h+1}^\star(s, a) \right| \le \frac{\lambda_k}{4V_H^\uparrow} \mathrm{Var}(V_{h+1}^\star)(s, a) + \frac{3V_H^\uparrow \ell_1}{\lambda_k N^k(s, a)}, \forall (s, a) \in \mathcal{S} \times \mathcal{A}, h \in [H], k \in \mathbb{N} \right\}$$

$$\mathbf{A}_2 \coloneqq \left\{ (P - \hat{P}^k)(V_{h+1}^\star)^2(s, a) \le \frac{1}{2} \mathrm{Var}(V_{h+1}^\star)(s, a) + \frac{6V_H^{\uparrow 2} \ell_1}{N^k(s, a)}, \forall (s, a) \in \mathcal{S} \times \mathcal{A}, h \in [H], k \in \mathbb{N} \right\}$$

$$\mathbf{A}_3 \coloneqq \left\{ \left| \hat{P}^k(s' \mid s, a) - P(s' \mid s, a) \right| \le 2 \sqrt{\frac{2P(s' \mid s, a) \ell_{3,k}(s, a)}{N^k(s, a)}} + \frac{2\ell_{3,k}(s, a)}{3N^k(s, a)}, \forall (s, a) \in \mathcal{S} \times \mathcal{A}, s' \in \mathcal{S}, k \in \mathbb{N} \right\}$$

$$\mathbf{A}_4 \coloneqq \left\{ \left| \hat{r}^k(s, a) - r(s, a) \right| \le \lambda_k r(s, a) + \frac{R_{\max} \ell_1}{\lambda_k N^k(s, a)}, \forall (s, a) \in \mathcal{S} \times \mathcal{A}, k \in \mathbb{N} \right\}$$

$$\mathbf{A}_5 \coloneqq \left\{ \sum_{k=1}^K \sum_{h=1}^{\nu^k - 1} \left( P J_{h+1}^k(s_h^k, a_h^k) - J_{h+1}^k(s_{h+1}^k) \right) \le \frac{1}{4V_H^\uparrow} \sum_{k=1}^K \sum_{h=1}^{\nu^k - 1} \mathrm{Var}(J_{h+1}^k)(s_h^k, a_h^k) + 3V_H^\uparrow \log \frac{6}{\delta}, \forall K \in \mathbb{N} \right\}$$

$$\mathbf{A}_6 \coloneqq \left\{ \sum_{k=1}^K \sum_{h=1}^{\nu^k - 1} \left( P (J_{h+1}^k)^2(s_h^k, a_h^k) - (J_{h+1}^k)^2(s_{h+1}^k) \right) \le \frac{1}{2} \sum_{k=1}^K \sum_{h=1}^{\nu^k - 1} \mathrm{Var}(J_{h+1}^k)(s_h^k, a_h^k) + 6V_H^{\uparrow 2} \log \frac{6}{\delta}, \forall K \in \mathbb{N} \right\}.$$

#### E.1.2  INFINITE-HORIZON DISCOUNTED MDPS

$$\mathbf{A}_1^\gamma \coloneqq \left\{ \left| (\hat{P}^t - P) V^\star(s, a) \right| \le \frac{\lambda_t}{4V_\gamma^\uparrow} \mathrm{Var}(V^\star)(s, a) + \frac{3V_\gamma^\uparrow \ell_1}{\lambda_t N^t(s, a)}, \forall (s, a) \in \mathcal{S} \times \mathcal{A}, t \in \mathbb{N} \right\}$$

$$\mathbf{A}_2^\gamma \coloneqq \left\{ (P - \hat{P}^t)(V^\star)^2(s, a) \le \frac{1}{2} \mathrm{Var}(V^\star)(s, a) + \frac{6V_\gamma^{\uparrow 2} \ell_1}{N^t(s, a)}, \forall (s, a) \in \mathcal{S} \times \mathcal{A}, t \in \mathbb{N} \right\}$$

$$\mathbf{A}_3^\gamma \coloneqq \left\{ \left| \hat{P}^t(s' \mid s, a) - P(s' \mid s, a) \right| \le 2 \sqrt{\frac{2P(s' \mid s, a) \ell_{3,k}(s, a)}{N^t(s, a)}} + \frac{2\ell_{3,k}(s, a)}{3N^t(s, a)}, \forall (s, a) \in \mathcal{S} \times \mathcal{A}, s' \in \mathcal{S}, t \in \mathbb{N} \right\}$$

$$\mathbf{A}_4^\gamma \coloneqq \left\{ \left| \hat{r}^t(s, a) - r(s, a) \right| \le \lambda_t r(s, a) + \frac{R_{\max} \ell_1}{\lambda_t N^t(s, a)}, \forall (s, a) \in \mathcal{S} \times \mathcal{A}, t \in \mathbb{N} \right\}$$

$$\mathbf{A}_5^\gamma \coloneqq \left\{ \sum_{t=1}^T \sum_{l=0}^{\nu_t - 1} \gamma^{l+1} \left( P J^t(s_{t+l}, a_{t+l}) - J^t(s_{t+l+1}) \right) \le \frac{(1 - \gamma)}{8V_\gamma^\uparrow} \sum_{t=1}^T \mathrm{Var}\left( Y^t(s_{t+1}) \right)(s_t, a_t) + \frac{6V_\gamma^\uparrow}{1 - \gamma} \log \frac{6}{\delta}, \forall T \in \mathbb{N} \right\}$$

$$\mathbf{A}_6^\gamma \coloneqq \left\{ \sum_{t=1}^T \left( P \left( Y^t(s_{t+1}) \right)^2 (s_t, a_t) - (Y^t(s_{t+1}))^2 \right) \le \frac{1}{4} \sum_{t=1}^T \mathrm{Var}\left( Y^t(s_{t+1}) \right)(s_t, a_t) + \frac{12V_\gamma^{\uparrow 2}}{(1 - \gamma)^2} \log \frac{6}{\delta}, \forall T \in \mathbb{N} \right\}.$$

For the definition of $Y^t(s_{t+1})$, please refer to the proof of Lemma 20.

### E.2 SAMPLE COMPLEXITY

To analyze sample complexity, we consider modifying the last two events using an indicator function that only accounts for a subset of episodes or time steps deemed "bad". Since the resulting bound is almost identical, except that these "bad" indices replace the full summation, we denote such events as $\mathbf{A}_7, \mathbf{A}_8$ for finite-horizon episodic MDPs, and $\mathbf{A}_7^{\gamma}, \mathbf{A}_8^{\gamma}$ for infinite-horizon discounted MDPs.

### E.3 PUTTING ALL TOGETHER

Each undesirable event is assigned probability at most $\frac{\delta}{6}$. By the union bound, the probability of their intersection is at least $1 - \delta$. Therefore, we have the following events spanning different results:

$$\mathcal{A} := \cap_{i=1}^{6} \mathbf{A}_i$$
$$\mathcal{B} := \cap_{i=1}^{6} \mathbf{A}_i^{\gamma}$$
$$\mathcal{C} := \left(\cap_{i=1}^{4} \mathbf{A}_i\right) \cap \left(\mathbf{A}_7 \cap \mathbf{A}_8\right)$$
$$\mathcal{D} := \left(\cap_{i=1}^{4} \mathbf{A}_i^{\gamma}\right) \cap \left(\mathbf{A}_7^{\gamma} \cap \mathbf{A}_8^{\gamma}\right).$$

## F PROOFS FOR FINITE-HORIZON EPISODIC MDPS

Our proof starts with finite-horizon episodic MDPs, which are simple to illustrate and play a vital role in bridging to the infinite-horizon case.

### F.1 PRELIMINARY CONSTRUCTIONS

Since our formulation decays more aggressively than $\frac{1}{N^k}$, we need to introduce an auxiliary value function $\widetilde{V}^k$ that behaves the same as the original before a critical point $m$, however, after which the error should be manageable. That is, it is the value function of the MDP $(\mathcal{S}, \mathcal{A}, \hat{P}^k, \tilde{r}^k, H)$, where only the reward is different compared to $V^k$ of $(\mathcal{S}, \mathcal{A}, \hat{P}^k, r_{\text{EUBRL}}^k, H)$. The modified reward is defined as $\tilde{r}^k = (1 - P_U^k)\hat{r}^k + b^k$, where the bonus term $b^k$ is defined as:

$$b^k = \begin{cases} P_U^k \eta^k \mathcal{E}^k, & \text{if } N^k < m. \\ P_U^k R_{\max} + \frac{\Upsilon^k}{N^k}, & \text{otherwise.} \end{cases}$$

Here $\eta^k = \mathcal{E}_{\max} \Upsilon^k + R_{\max} \sqrt{m_k}$, for which more details can be found in Lemma 33.

The reward is increased to a degree that decays at least as fast as $\frac{1}{N^k}$, ensuring an advantage over the complexity arising from the reciprocal of visits. Although this advantage holds for an arbitrary $m$, we need to control the error between the two value functions thereafter. For this reason, we set $m_k = \frac{V_H^{\uparrow 2}}{R_{\max}^2 \lambda_k^2}$, which yields a sufficiently small error.

### F.2 QUASI-OPTIMISM WITH EPISTEMIC RESISTANCE

**Lemma 2.** *For finite-horizon episodic MDPs, under high-probability event $\mathbf{A}_1 \cap \mathbf{A}_2$, it holds that for all $s \in \mathcal{S}, h \in [H+1], k \in \mathbb{N}$,*

$$\widetilde{V}_h^k(s) + \left(\frac{3}{2} - P_U^{k,\star}(s)\right)\lambda_k H \geq V_h^{\star}(s)$$

*Proof.* Since we want to bound the error between $V^{\star}(s)$ and $V^k(s)$ for any $s \in \mathcal{S}$. The auxiliary function is served as a bridge to achieve that. Let us decompose the error $V^{\star}(s) - V^k(s)$ as follows:

$$V^{\star}(s) - V^k(s) = \underbrace{V^{\star}(s) - \widetilde{V}^k(s)}_{\text{Quasi-optimism}} + \underbrace{\widetilde{V}^k(s) - V^k(s)}_{\text{Complexity}}.$$

The complexity can be bounded by Lemma 3, and its proof will be given later. We now focus on the other part.

The proof follows the procedure of Lemma 2 in (Lee & Oh, 2025), with modifications to fit our formulation. The epistemic uncertainty guidance allows us to establish a refined induction hypothesis, thereby tightening the bound in proportion to the degree of uncertainty.

To simplify notations, we write $P_U^{k,\star}(s) := P_U^k(s, \pi^\star(s))$ and $P_U^k(s) := P_U^k(s, \pi^k(s))$. Furthermore, let $a^\star := \pi^\star(s)$, $a := \pi^k(s)$, and $\tilde{a} := \tilde{\pi}^k(s)$ denote the actions under the optimal policies corresponding to $V^\star$, $V^k$, and $\widetilde{V}^k$, respectively.

We prove by backward induction on $h$:

$$V_h^\star(s) - \widetilde{V}_h^k(s) \leq \lambda_k \left( \left(2 - P_U^{k,\star}(s)\right) V_h^\star(s) - \frac{1}{2V_H^\uparrow}(V_h^\star)^2(s) \right).$$

For the base case $h = H + 1$, both sides are 0, therefore the inequality holds. Assume it holds for $h + 1$, we will show it holds for $h$. If $\widetilde{V}_h^k = V_H^\uparrow$, then left-hand side will be no positive, therefore the inequality trivially holds. Suppose $\widetilde{V}_h^k < V_H^\uparrow$, by definition we have

$$\widetilde{V}_h^k(s) = \tilde{r}^k(s, \tilde{a}) + \hat{P}^k \widetilde{V}_{h+1}^k(s, \tilde{a}).$$

With this, we obtain:

$$
\begin{aligned}
V_h^\star(s) - \widetilde{V}_h^k(s) &= \left(r(s, a^\star) + PV_{h+1}^\star(s, a^\star)\right) - \left(\tilde{r}^k(s, \tilde{a}) + \hat{P}^k \widetilde{V}_{h+1}^k(s, \tilde{a})\right) \\
&\overset{(a)}{\leq} \left(r(s, a^\star) + PV_{h+1}^\star(s, a^\star)\right) - \left(\tilde{r}^k(s, a^\star) + \hat{P}^k \widetilde{V}_{h+1}^k(s, a^\star)\right) \\
&= r(s, a^\star) - \tilde{r}^k(s, a^\star) + \left(PV_{h+1}^\star(s, a^\star) - \hat{P}^k \widetilde{V}_{h+1}^k(s, a^\star)\right) \\
&= r(s, a^\star) - \left((1 - P_U^{k,\star}(s))\hat{r}^k(s, a^\star) + b^k(s, a^\star)\right) + \left(PV_{h+1}^\star(s, a^\star) - \hat{P}^k \widetilde{V}_{h+1}^k(s, a^\star)\right) \\
&\overset{(b)}{=} (1 - P_U^{k,\star}(s)) \left(r(s, a^\star) - \hat{r}^k(s, a^\star)\right) + \left(P_U^{k,\star}(s)r(s, a^\star) - b^k(s, a^\star)\right) \\
&\quad + \left(PV_{h+1}^\star(s, a^\star) - \hat{P}^k \widetilde{V}_{h+1}^k(s, a^\star)\right),
\end{aligned}
$$

where **(a)** is due to the optimality of $\tilde{a}$ and **(b)** by noting $r(s, a^\star) = \left((1 - P_U^{k,\star}(s)) + P_U^{k,\star}(s)\right) r(s, a^\star)$.

Since $r \leq R_{\max}$, we have:

$$P_U^{k,\star}(s)r(s, a^\star) - b^k(s, a^\star) \leq P_U^{k,\star}(s)R_{\max} - b^k(s, a^\star).$$

At this point, we note that the intermediate steps are identical to those in (Lee & Oh, 2025); therefore, we omit them here and state the resulting expression. Denote $\Upsilon^k := \frac{7V_H^\uparrow \ell_{1,k}}{\lambda_k}$, we obtain:

$$
\begin{aligned}
V_h^\star(s) - \widetilde{V}_h^k(s) &\leq -(b^k(s, a^\star) - P_U^{k,\star}(s)R_{\max}) + \frac{(7 - P_U^{k,\star}(s))V_H^\uparrow \ell_{1,k}}{\lambda_k N^k(s, a^\star)} \\
&\quad + \lambda_k(2 - P_U^{k,\star}(s))(r(s, a^\star) + PV_{h+1}^\star(s, a^\star)) - \frac{\lambda_k}{2V_H^\uparrow}(V_h^\star)^2(s) \\
&= -(b^k(s, a^\star) - P_U^{k,\star}(s)R_{\max}) + \frac{(7 - P_U^{k,\star}(s))V_H^\uparrow \ell_{1,k}}{\lambda_k N^k(s, a^\star)} + \lambda_k \left( \left(2 - P_U^{k,\star}(s)\right) V_h^\star(s) - \frac{1}{2V_H^\uparrow}(V_h^\star)^2(s) \right) \\
&\leq -(b^k(s, a^\star) - P_U^{k,\star}(s)R_{\max}) + \frac{\Upsilon^k}{N^k(s, a^\star)} + \lambda_k \left( \left(2 - P_U^{k,\star}(s)\right) V_h^\star(s) - \frac{1}{2V_H^\uparrow}(V_h^\star)^2(s) \right) \\
&\overset{(a)}{\leq} \lambda_k \left( \left(2 - P_U^{k,\star}(s)\right) V_h^\star(s) - \frac{1}{2V_H^\uparrow}(V_h^\star)^2(s) \right) \quad\quad\quad (1)
\end{aligned}
$$

where **(a)** is due to the fact of Lemma 33. Moreover, note that for $s \in \mathcal{S}$, we have $1 \leq 2 - P_U^{k,\star}(s) \leq 2$, therefore the function $f(x) = \left(2 - P_U^{k,\star}(s)\right) x - \frac{1}{2V_H^\uparrow}x^2, x \in [0, V_H^\uparrow]$ is bounded by $\left(\frac{3}{2} - P_U^{k,\star}(s)\right) V_H^\uparrow$. Substituting this for Eq. 1 completes the proof. $\qquad\square$

### F.3 BOUNDEDNESS OF COMPLEXITY

**Lemma 3.** *For all $s \in \mathcal{S}, h \in [H+1], k \in \mathbb{N}$, it holds that*

$$\widetilde{V}_h^k(s) - V_h^k(s) \leq R_{max}\lambda_k := \Phi_k.$$

We first introduce the the following elementary lemma:

**Lemma 4.** *For any $n > \max\{m, \frac{1}{\epsilon^2}\}$, we have*

$$\frac{1}{\sqrt{n}} - \frac{\sqrt{m}}{n} = \frac{\sqrt{n} - \sqrt{m}}{n} < \epsilon.$$

*Proof.* Since $n > m$, $\frac{1}{\sqrt{n}} - \frac{\sqrt{m}}{n} \geq 0$. To require $\epsilon$-accuracy, it needs $\frac{1}{\sqrt{n}} < \frac{\sqrt{m}}{n} + \epsilon$. If we have $\frac{1}{\sqrt{n}} < \epsilon \Leftrightarrow n > \frac{1}{\epsilon^2}$, then the result is desired, which is because:

$$\frac{1}{\sqrt{n}} < \epsilon < \frac{\sqrt{m}}{n} + \epsilon$$

$\square$

*Proof of Lemma 3.* The following will bound the complexity term $\widetilde{V}^k(s) - V^k(s)$. Since the two terms differ only in rewards, we first bound the difference in rewards $\Delta_r^k = |\tilde{r}^k - r_{\text{EUBRL}}^k|$.

Without loss of generality, we bound the reward for finite-horizon episodic MDPs. We set $\epsilon_k = \frac{R_{max}\lambda_k}{V_H^\uparrow}$, thereby $m_k = \frac{V_H^{\uparrow 2}}{R_{max}^2\lambda_k^2}$.

If $N^k < m_k$, the reward $\tilde{r}^k$ of $\widetilde{V}^k$ is the same as $r_{\text{EUBRL}}^k$, therefore $\Delta_r^k = 0$; otherwise, we have

$$
\begin{aligned}
\Delta_r^k &= \left| \left( P_U^k R_{\max} + \frac{\Upsilon^k}{N^k} \right) - \left( P_U^k \eta^k \mathcal{E}^k \right) \right| \\
&= \left| \left( P_U^k R_{\max} + \frac{\Upsilon^k}{N^k} \right) - \left( \left( \Upsilon^k + \frac{R_{\max}}{\mathcal{E}_{\max}}\sqrt{m_k} \right) \frac{1}{N^k} \right) \right| \\
&= \left| P_U^k R_{\max} - \frac{\frac{R_{\max}}{\mathcal{E}_{\max}}\sqrt{m_k}}{N^k} \right| \\
&= \left| \frac{1}{\sqrt{N^k}} \frac{R_{\max}}{\mathcal{E}_{\max}} - \frac{\frac{R_{\max}}{\mathcal{E}_{\max}}\sqrt{m_k}}{N^k} \right| \\
&= \left| \frac{R_{\max}}{\mathcal{E}_{\max}} \left( \frac{1}{\sqrt{N^k}} - \frac{\sqrt{m_k}}{N^k} \right) \right| \\
&= \frac{R_{\max}}{\mathcal{E}_{\max}} \left| \frac{\sqrt{N^k} - \sqrt{m_k}}{N^k} \right| \\
&= \frac{R_{\max}}{\mathcal{E}_{\max}} \left( \frac{\sqrt{N^k} - \sqrt{m_k}}{N^k} \right) \\
&\leq \frac{R_{\max}}{\mathcal{E}_{\max}} \frac{R_{\max}\lambda_k}{V_\gamma^\uparrow} \\
&= \frac{R_{\max}}{\mathcal{E}_{\max}} \frac{\lambda_k}{H} \\
&\leq R_{\max} \frac{\lambda_k}{H},
\end{aligned}
$$

where the second to last is because of Lemma 4, and the last is because of the assumption that $\mathcal{E}_{\max} \geq 1$.

By Simulation Lemma (Kearns & Singh, 2002), we know that the value functions differ at most $R_{\max}\lambda_t$.

For infinite-horizon discounted MDPs, the proof is similar, except that we need to replace the time index with $t$ and the maximum value function with $V_\gamma^\uparrow$. $\qquad\square$

### F.4 BOUNDEDNESS OF ACCURACY

**Lemma 5.** *For finite-horizon episodic MDPs, under high-probability event $\cap_{i=1}^4 \mathbf{A}_i$, it holds that for all $s \in \mathcal{S}, h \in [H+1], k \in \mathbb{N}$,*

$$V_h^k(s) - V_h^{\pi^k}(s) \le \left(3 - 2P_U^k(s) - \frac{2}{7}P_U^{k,\star}(s)\right)\lambda_k H + 2J^k(s) + \mathcal{O}\left(\frac{\Phi_k^2}{V_H^\uparrow} + \lambda_k \Phi_k\right).$$

It is convenient to define the following quantities for the analysis.

**Definition 2.** Let $D_h^k(s)$ be defined by

$$D_h^k(s) := \lambda_k\left((3 - 2P_U^k(s))V_h^\star(s) - \frac{1}{2V_H^\uparrow}(V_h^\star)^2(s)\right) + \frac{1}{7V_H^\uparrow}\left((S_k)^2 - \left(\widehat{V}_h^k(s) + S_k\right)^2\right),$$

where $\beta^k(s,a)$:

$$\beta^k(s,a) = P_U^k(s,a)\eta^k\mathcal{E}^k(s,a) + \beta_1^k(s,a) + (1 - P_U^k(s))\frac{V_H^\uparrow \ell_{1,k}}{\lambda_k N^k(s,a)}$$

$$\widehat{V}_h^k(s) := V_h^k(s) - V_h^\star(s)$$

$$S_k := \left(\frac{3}{2} - P_U^{k,\star}(s)\right)\lambda_k V_H^\uparrow + \Phi_k,$$

in which

$$\beta_1^k(s,a) := \frac{1}{N^k(s,a)}\left(\frac{3V_H^\uparrow \ell_{1,k}}{\lambda_k} + 30V_H^\uparrow S\ell_{3,k}(s,a)\right).$$

*Proof of Lemma 5.* The key to bound the accuracy term $V_h^k(s) - V_h^{\pi^k}(s)$ is to decompose it into differences:

$$V_h^k(s) - V_h^{\pi^k}(s) = \underbrace{\Delta_h\left(V^k - V^{\pi_k}\right)(s,a)}_{I_1} + P\left(V_{h+1}^k - V_{h+1}^{\pi_k}\right)(s,a).$$

By Lemma 35, we know that

$$I_1 \le \left(\Delta_h(D^k)(s,a) + 2\beta^k(s,a)\right).$$

Combining this with backward induction on $h$, we obtain

$$V_h^k(s) - V_h^{\pi_k}(s) = D_h^k(s) + 2J_h^k(s).$$

The final step is to bound $D_h^k(s)$. Denote

$$I_2 := (3 - 2P_U^k(s))V_h^\star(s) - \frac{1}{2V_H^\uparrow}(V_h^\star)^2(s)$$

$$I_3 := \left(S_k^2 - \left(\widehat{V}_h^k(s) + S_k\right)^2\right),$$

we have $D_h^k(s) = \lambda_k I_2 + \frac{1}{7V_H^\uparrow}I_3$.

We now bound $I_2$ and $I_3$ individually.

**Bounding $I_2$**

$$I_2 \le \left(\frac{5}{2} - 2P_U^k(s)\right)V_H^\uparrow \tag{2}$$

**Bounding $I_3$**

$$I_3 = -\widehat{V}_h^k(s)^2 - 2S_k\widehat{V}_h^k(s) \leq S_k^2, \ \widehat{V}_h^k(s) \in [-V_H^\uparrow, V_H^\uparrow] \tag{3}$$

$$S_k = \left(\frac{3}{2} - P_U^{k,\star}(s)\right)\lambda_k V_H^\uparrow + \Phi_k \tag{4}$$

$$S_k^2 \leq \left(\frac{3}{2} - P_U^{k,\star}(s)\right)^2 \lambda_k^2 V_H^{\uparrow 2} + \Phi_k^2 + \left(3 - 2P_U^{k,\star}(s)\right)\lambda_k V_H^\uparrow \Phi_k \tag{5}$$

Therefore,

$$D_h^k(s) = \lambda_k I_2 + \frac{1}{7V_H^\uparrow}I_3 \tag{6}$$

$$\leq \left(\frac{5}{2} - 2P_U^k(s)\right)\lambda_k V_H^\uparrow + \frac{1}{7}\left(\frac{3}{2} - P_U^{k,\star}(s)\right)^2 \lambda_k^2 V_H^\uparrow + \mathcal{O}\left(\frac{\Phi_k^2}{V_H^\uparrow}\right) + \mathcal{O}(\lambda_k \Phi_k) \tag{7}$$

$$\leq \left(3 - 2P_U^k(s) - \frac{2}{7}P_U^{k,\star}(s)\right)\lambda_k V_H^\uparrow + \mathcal{O}\left(\frac{\Phi_k^2}{V_H^\uparrow} + \lambda_k \Phi_k\right), \tag{8}$$

which completes the proof. $\qquad\qquad\square$

### F.5 BOUNDEDNESS OF $J_1^k$

**Lemma 6.** *For finite-horizon episodic MDPs, under high-probability event $\mathbf{A}_5 \cap \mathbf{A}_6$, it holds that*

$$\sum_{k=1}^{K} J_1^k(s_1^k) \leq 2\sum_{k=1}^{K}\sum_{h=1}^{\nu^k-1}\beta^k(s_h^k, a_h^k) + 6V_H^\uparrow SA \log\frac{12H}{\delta},$$

*for all $K \in \mathbb{N}$.*

**Lemma 7.** *For finite-horizon episodic MDPs, under high-probability event $\mathbf{A}_5 \cap \mathbf{A}_6$, denote $\mathcal{Y}^{(K)} := \frac{12V_H^\uparrow \ell_{1,K}}{\lambda_K} + 30V_H^\uparrow S\ell_{3,K}$, it holds that*

$$\sum_{k=1}^{K} J_1^k(s_1^k) \leq 4\mathcal{Y}^{(K)}SA \log\left(1 + \frac{KH}{SA}\right) + 6V_H^\uparrow SA \log\frac{12H}{\delta},$$

*for all $K \in \mathbb{N}$.*

### F.6 LOWER BOUND OF EPISTEMIC RESISTANCE

**Lemma 8** (Lower Bound of Epistemic Resistance). *Given a uniform $\lambda_k = \lambda, \forall k \in \mathbb{N}$, it holds that*

$$\sum_{k=1}^{K} \mathfrak{R}^k(s_1^k)\lambda_k V_H^\uparrow \geq \frac{23R_{max}}{7}\left(\frac{2}{\mathcal{E}_{max}}\left(\sqrt{HK} - \sqrt{H}\right) + H\right)\lambda,$$

*for any $K \in \mathbb{N}$.*

*Proof.*

$$\sum_{k=1}^{K} P_U^k(s_1^k, a_1^k) = 1 + \frac{1}{\mathcal{E}_{\max}} \sum_{k=2}^{K} \frac{1}{\sqrt{N^k(s_1^k, a_1^k)}}$$

$$\geq 1 + \frac{1}{\mathcal{E}_{\max}} \sum_{k=2}^{K} \frac{1}{\sqrt{(k-1)H}}$$

$$= 1 + \frac{1}{\mathcal{E}_{\max}\sqrt{H}} \sum_{k=1}^{K-1} \frac{1}{\sqrt{k}}$$

$$\geq 1 + \frac{1}{\mathcal{E}_{\max}\sqrt{H}} \int_{k}^{k+1} \frac{1}{\sqrt{x}} \, dx$$

$$= 1 + \frac{1}{\mathcal{E}_{\max}\sqrt{H}} \left(2\sqrt{K} - 2\right),$$

Note, this also holds for $P_U^{k,\star}(s_1^k, \pi_1^\star(s_1^k))$. Therefore, multiplying with $\frac{23}{7}\lambda V_H^\uparrow$ completes the proof.
$\square$

### F.7 REGRET ANALYSIS

Combining the results of Lemmas 2–5, we obtain the per-step regret:

**Theorem 6.** *Under high-probability event $\mathcal{A}$, it holds that for all $s \in \mathcal{S}, h \in [H+1], k \in \mathbb{N}$,*

$$V^\star(s) - V^{\pi^k}(s) \leq \left(\frac{9}{2} - \mathfrak{R}^k(s)\right) \lambda_k V_H^\uparrow + 2J^k(s) + \mathcal{O}\left(\Phi_k \left(1 + \frac{\Phi_k}{V_H^\uparrow}\right)\right),$$

*where we define the following as **Epistemic Resistance***

$$\mathfrak{R}^k(s) := 2P_U^k(s) + \frac{9}{7} P_U^{k,\star}(s).$$

**Theorem 7.** *For finite-horizon episodic MDPs, for any fixed $K \in \mathbb{N}$, with probability at least $1 - \delta$, it holds that*

$$Regret(K) \leq \widetilde{\mathcal{O}}(H\sqrt{SAK} + HS^2A).$$

*Proof.* From Theorem 6, we have:

$$Regret(K) \leq \frac{9V_H^\uparrow}{2} \sum_{k=1}^{K} \lambda_k - V_H^\uparrow \sum_{k=1}^{K} \mathfrak{R}^k(s_1^k)\lambda_k + 2\sum_{k=1}^{K} J^k(s_1^k) + \sum_{k=1}^{K} \mathcal{O}\left(\Phi_k\left(1 + \frac{\Phi_k}{V_H^\uparrow}\right)\right).$$

Choose $\lambda_k = \min\{1, 4\sqrt{\frac{SA\ell_1\ell_{2,K}}{K}}\}$, $\forall k \in [K]$ and denote $\Psi(K) := \frac{2\sum\limits_{k=1}^{K} \mathfrak{R}^k(s)}{9K}$, we have

$$\frac{9V_H^\uparrow}{2} \sum_{k=1}^{K} \lambda_k - V_H^\uparrow \sum_{k=1}^{K} \mathfrak{R}^k(s_1^k)\lambda_k = \frac{9V_H^\uparrow}{2}\left(1 - \frac{2\sum\limits_{k=1}^{K} \mathfrak{R}^k(s_1^k)}{9K}\right) K\min\{1, 4\sqrt{\frac{SA\ell_1\ell_{2,K}}{K}}\}$$

$$\leq 18V_H^\uparrow(1 - \Psi(K))K\sqrt{\frac{SA\ell_1\ell_{2,K}}{K}}$$

$$= 18V_H^\uparrow(1 - \Psi(K))\sqrt{SAK\ell_1\ell_{2,K}}.$$

From Lemma 7, we know that

$$2\sum_{k=1}^{K} J^k(s_1^k) \le 8\mathcal{Y}^{(K)} SA \log\left(1 + \frac{KH}{SA}\right) + 12V_H^{\uparrow} SA \log\frac{12H}{\delta}$$

$$\le \frac{96V_H^{\uparrow} SA\ell_{1,K}\ell_{2,K}}{\lambda_K} + 240V_H^{\uparrow} S^2 A\ell_{2,K}\ell_{3,K} + 12V_H^{\uparrow} SA \log\frac{12H}{\delta}.$$

$$\le 96V_H^{\uparrow} SA\ell_{1,K}\ell_{2,K} \max\left\{1, \frac{1}{4}\sqrt{\frac{K}{SA\ell_1\ell_{2,K}}}\right\} + 240V_H^{\uparrow} S^2 A\ell_{2,K}\ell_{3,K} + 12V_H^{\uparrow} SA \log\frac{12H}{\delta}.$$

$$\le 96V_H^{\uparrow} SA\ell_{1,K}\ell_{2,K} + 24V_H^{\uparrow}\sqrt{SAK\ell_1\ell_{2,K}} + 240V_H^{\uparrow} S^2 A\ell_{2,K}\ell_{3,K} + 12V_H^{\uparrow} SA \log\frac{12H}{\delta}.$$

$$\le 24V_H^{\uparrow}\sqrt{SAK\ell_1\ell_{2,K}} + 336V_H^{\uparrow} S^2 A\ell_{1,K}'(1 + \ell_{2,K}),$$

where we denote $\ell_{1,K}' := \log\frac{24HSA(1+\log KH)}{\delta}$ as an upper bound of both $\ell_{1,K}$ and $\ell_{3,K}$, and merge into the non-leading term.

Combining these two together, we get:

$$\text{Regret}(K) \le (42 - 18\Psi(K)) V_H^{\uparrow}\sqrt{SAK\ell_1\ell_{2,K}} + 336V_H^{\uparrow} S^2 A\ell_{1,K}'(1 + \ell_{2,K}) + \sum_{k=1}^{K}\mathcal{O}\left(\Phi_k\left(1 + \frac{\Phi_k}{V_H^{\uparrow}}\right)\right)$$

$$\le (42 - 18\Psi(K)) R_{\max}H\sqrt{SAK\ell_1\ell_{2,K}} + 336R_{\max}HS^2 A\ell_{1,K}'(1 + \ell_{2,K}) + \sum_{k=1}^{K}\mathcal{O}\left(\Phi_k\left(1 + \frac{\Phi_k}{V_H^{\uparrow}}\right)\right),$$

where only the last part left to resolve.

Given $\Phi_k = R_{\max}\lambda_k$, we have one additional source of $\mathcal{O}(\lambda K)$, which will be merged into the leading term. In addition, note that

$$\sum_{k=1}^{K}\mathcal{O}\left(\frac{\Phi_k^2}{V_H^{\uparrow}}\right) = \sum_{k=1}^{K}\widetilde{\mathcal{O}}\left(\frac{R_{\max}^2 SA}{KV_H^{\uparrow}}\right)$$

$$= \widetilde{\mathcal{O}}\left(\frac{R_{\max}^2 SA}{V_H^{\uparrow}}\right)$$

$$\le \widetilde{\mathcal{O}}(R_{\max}SA),$$

which only increases the non-leading term by some constants. So overall, we have:

$$\text{Regret}(K) = \widetilde{\mathcal{O}}\left(H\sqrt{SAK} + HS^2 A\right).$$

$\square$

### F.8 SAMPLE COMPLEXITY

**Theorem 8.** *For finite-horizon episodic MDPs, with probability at least $1-\delta$, the sample complexity is bounded by*

$$\widetilde{\mathcal{O}}\left(\left(\frac{H^2 SA}{\epsilon^2} + \frac{HS^2 A}{\epsilon}\right)\log\frac{1}{\delta}\right).$$

For finite-horizon episodic MDPs, the sample complexity of an algorithm is defined as the number of non-$\epsilon$-optimal episodes taken over the course of learning (Dann & Brunskill, 2015; Dann et al., 2017). If this sample complexity can be bounded by a polynomial function $f(S, A, \frac{1}{\epsilon}, \frac{1}{\delta}, H)$, then the algorithm is PAC-MDP.

The proof is analogous to that of the infinite-horizon case in Appendix G.8; therefore, we only provide a sketch.

From Theorem 6, we know that the per-step regret can be bounded as follows:

$$V^{\star}(s_1^k) - V^k(s_1^k) \leq \left(\frac{9}{2} - \mathfrak{R}^k(s_1^k)\right)\lambda_k V_H^{\uparrow} + 2J^k(s_1^k) + \Phi_k\left(1 + \left(3 - 2P_U^{k,\star}(s_1^k)\right)\lambda_k + \frac{\Phi_k}{V_H^{\uparrow}}\right)$$

$$\leq \underbrace{\left(\frac{9}{2} - \mathfrak{R}^k(s_1^k)\right)\lambda_k V_H^{\uparrow}}_{:=L_{1,k}} + \underbrace{2J^k(s_1^k)}_{:=L_{2,k}} + \underbrace{\Phi_k\left(4 + \frac{\Phi_k}{V_H^{\uparrow}}\right)}_{:=L_{3,k}}$$

We choose $\lambda_k = \frac{\epsilon}{18V_H^{\uparrow}}$, so that we have $L_{1,k} \leq \frac{\epsilon}{4}$ and $L_{3,k} \leq \frac{\epsilon}{4}$. So the remaining step is to prove that majority of episodes satisfy $J^k(s_1^k) \leq \frac{\epsilon}{4}$, which implies $L_{2,k} \leq \frac{\epsilon}{2}$.

The following notations are to connect the number of non-optimal episodes with $J^k(s_1^k)$.

Let the set of non-optimal episodes within $K$ total episodes be defined as $\Gamma_K := \{k \in [K] : J^k(s_1^k) > \frac{\epsilon}{4}\}$, and its cardinality $|\Gamma_K|$. We overload the definition of visits that occur only in $\Gamma_K$.

$$n_h^k(s,a) := \sum_{\kappa \in \Gamma_k}\sum_{\tau=1}^{H} \mathbf{1}((s_\tau^\kappa, a_\tau^\kappa) = (s,a), (\kappa < k \text{ or } \tau \leq h))$$

$$N^k(s,a) := \sum_{\kappa \in \Gamma_{k-1}}\sum_{\tau=1}^{H} \mathbf{1}((s_\tau^\kappa, a_\tau^\kappa) = (s,a))$$

$$\nu^k := \begin{cases} \min\{h \in [H] : n_h^k(s_h^k, a_h^k) > 2N^k(s_h^k, a_h^k)\}, & \text{if } h \text{ exists.} \\ H+1, & \text{otherwise.} \end{cases}$$

Akin to Lemma 26, we can bound $|\Gamma_K|$ using the fact that $J^k(s_1^k) > \frac{\epsilon}{4}$.

**Definition 3.** Let $W(K)$ be defined by

$$W(K) := \frac{3456 R_{\max}^2 H^2 SA \ell_{1,K}\ell_{2,K}}{\epsilon^2} + \frac{480 R_{\max} H S^2 A \ell_{2,K}\ell_{3,K}}{\epsilon} + \frac{24 R_{\max} H SA \ell_{1,K}}{\epsilon}.$$

**Lemma 9.** *For finite-horizon episodic MDPs, under high-probability event* **C***, it holds that*

$$|\Gamma_K| \leq W(|\Gamma_K|),$$

*for all $K \in \mathbb{N}$.*

**Proposition 2.** *For finite-horizon episodic MDPs, let $K_0$ be defined as*

$$K_0 := \left\lfloor \frac{6920 R_{max}^2 H^2 SA \ell_1 \ell_{5,\epsilon}}{\epsilon^2} + \frac{480 R_{max} H S^2 A(2\ell_1 + \ell_{6,\epsilon})\ell_{5,\epsilon}}{\epsilon} \right\rfloor.$$

*Then the sample complexity of* EUBRL *is at most $K_0$ with probability at least $1 - \delta$.*

Before proving this result, we need to bound the the other way around i.e. $W(K_0) < K_0$.

**Lemma 10.** *It holds that*

$$W(K_0) < K_0.$$

*Proof of Proposition 2.* From Lemmas 9 and 10, we know that $|\Gamma_K| \leq W(|\Gamma_K|)$ and $W(K_0) < K_0$. It implies that $|\Gamma_K| \neq K_0$ for all $K \in \mathbb{N}$. Since $|\Gamma_K|$ increases by at most 1 starting from $|\Gamma_0| = 0$, that is, $|\Gamma_{K+1}| \leq |\Gamma_K| + 1$ for all $K \in \mathbb{N}$, we conclude that $|\Gamma_K| < K_0$ for all $K \in \mathbb{N}$. Otherwise, there exists $K'$ such that $|\Gamma_{K'}| > K_0$. Assume $K'$ is the minimal such index. Then it follows that $|\Gamma_{K'-1}| = K_0$, which leads to a contradiction. $\square$

# G  PROOFS FOR INFINITE-HORIZON DISCOUNTED MDPS

The difficulty in proving quasi-optimism and bounding accuracy is that we can no longer use backward induction on the horizon, since the value function is time-independent. To resolve this, we construct Bellman-like operators to bridge this gap.

### G.1 QUASI-OPTIMISM WITH EPISTEMIC RESISTANCE

**Lemma 11.** *For infinite-horizon discounted MDPs, under high-probability event $\mathbf{A}_1^\gamma \cap \mathbf{A}_2^\gamma$, it holds that for all $s \in \mathcal{S}, t \in \mathbb{N}$,*

$$V^\star(s) - \widetilde{V}^t(s) \le \lambda_t \left( \left(2 - P_U^{t,\star}(s)\right) V^\star(s) - \frac{1}{2V_\gamma^\uparrow}(V^\star)^2(s) \right).$$

**Corollary 2.** *For infinite-horizon discounted MDPs, under high-probability event $\mathbf{A}_1^\gamma \cap \mathbf{A}_2^\gamma$, it holds that for all $s \in \mathcal{S}, t \in \mathbb{N}$,*

$$V^\star(s) - \widetilde{V}^t(s) \le \lambda_t \left( \frac{3}{2} - P_U^{t,\star}(s) \right) V_\gamma^\uparrow.$$

To prove Lemma 11, we need to define a Bellman-like operator that is a contraction mapping and monotone.

**Definition 4.** Let operator $\mathcal{T}_1$ be defined by

$$(\mathcal{T}_1 V)(s) := \left(P_U^{t,\star}(s) R_{\max} - b^t(s, \pi^\star(s))\right) + \left(1 - P_U^{t,\star}(s)\right) \left(r(s, \pi^\star(s)) - \hat{r}^t(s, \pi^\star(s))\right)$$
$$+ \gamma \left(P - \hat{P}^t\right) V^\star(s, \pi^\star(s)) + \gamma \hat{P}^t V(s, \pi^\star(s)).$$

**Lemma 12.** $\mathcal{T}_1$ *is a contraction mapping and monotone.*

*Proof.* Denote

$$M(s) := \left(P_U^{t,\star}(s) R_{\max} - b^t(s, \pi^\star(s))\right) + \left(1 - P_U^{t,\star}(s)\right) \left(r(s, \pi^\star(s)) - \hat{r}^t(s, \pi^\star(s))\right)$$
$$+ \gamma \left(P - \hat{P}^t\right) V^\star(s, \pi^\star(s))$$

For any $U, V \in [0, V_\gamma^\uparrow]^S$, we have

$$\|\mathcal{T}_1 U - \mathcal{T}_1 V\|_\infty = \sup_s \left| \left(M(s) + \gamma \hat{P}^t U(s, \pi^\star(s))\right) - \left(M(s) + \gamma \hat{P}^t V(s, \pi^\star(s))\right) \right|$$
$$= \gamma \sup_s \left| \hat{P}^t (U - V)(s, \pi^\star(s)) \right|$$
$$\le \gamma \|U - V\|_\infty.$$

Therefore, $\mathcal{T}_1$ is a contraction mapping under $\infty$-norm.

On the other hand, given $U, V \in [0, V_\gamma^\uparrow]^S$ such that $U(s) \le V(s), \forall s \in \mathcal{S}$, we have

$$(\mathcal{T}_1 U - \mathcal{T}_1 V)(s) = \hat{P}^t (U - V)(s, \pi^\star(s))$$
$$\le 0.$$

Thus, $\mathcal{T}_1$ is monotone as well. $\qquad \square$

**Lemma 13.** *Denote* $f(s) = \lambda_t \left( \left(2 - P_U^{t,\star}(s)\right) V^\star(s) - \frac{1}{2V_\gamma^\uparrow}(V^\star)^2(s) \right)$, *under high-probability event* $\mathbf{A}_1^\gamma \cap \mathbf{A}_2^\gamma$, *it holds that*

$$\mathcal{T}_1 f \le f$$

*Proof.* This follows the same procedure as the proof of Lemma 2, except that we apply Lemma 34, the boundedness of the discounted value function, and the inequalities stated in the events $\mathbf{A}_1^\gamma$ and $\mathbf{A}_2^\gamma$. $\qquad \square$

Now we prove Lemma 11.

*Proof of Lemma 11.* Denote $\Delta V := V^\star - \widetilde{V}^t$.

Since $\mathcal{T}_1$ is a contraction mapping, by the Banach fixed-point theorem, there exists a fixed point $\bar{V}$ such that $\bar{V} = \lim_{k \to \infty} (\mathcal{T}_1)^k g$ from an arbitrary initial point $g$.

Note, $\Delta V \le \mathcal{T}_1 \Delta V$. By monotonicity and contraction of $\mathcal{T}_1$ from Lemma 12, we have $\Delta V \le \mathcal{T}_1 \Delta V \le \lim_{k \to \infty} (\mathcal{T}_1)^k \Delta V = \bar{V}$. By Lemma 13, we have $\mathcal{T}_1 f \le f$, by monotonicity and contraction again, we have $\bar{V} = \lim_{k \to \infty} (\mathcal{T}_1)^k f \le \mathcal{T}_1 f \le f$. Combining two sides, we conclude that $\Delta V \le f$, which completes the proof. $\qquad \square$

### G.2 BOUNDEDNESS OF COMPLEXITY

**Lemma 14.** *For all $s \in \mathcal{S}, t \in \mathbb{N}$, it holds that*

$$\widetilde{V}^t(s) - V^t(s) \leq R_{max}\lambda_t \coloneqq \Phi_t.$$

*Proof.* See the proof of Lemma 3. $\qquad\square$

### G.3 BOUNDEDNESS OF ACCURACY

In this section, we bound the accuracy term. Although it is tempting to use the same logic as in the previous section, it is worth noting that the nuance in the definition of $J$ prevents this, as we no longer have an argument analogous to $\mathcal{T}_1 f \leq f$. We state the main result in advance, followed by its proof.

**Lemma 15.** *For infinite-horizon discounted MDPs, under high-probability event $\cap_{i=1}^4 \mathbf{A}_i^\gamma$, it holds that for all $s \in \mathcal{S}, t \in \mathbb{N}$,*

$$V^t(s) - V^{\pi_t}(s) \leq D_\gamma^t(s) + 2J_\gamma^t(s).$$

Before we dive into details, we define the following relevant quantities.

**Definition 5.** Let $D_\gamma^t(s)$ be defined by

$$D_\gamma^t(s) \coloneqq \lambda_t \left( \left(3 - 2P_U^t(s)\right) V^\star(s) - \frac{1}{2V_\gamma^\uparrow}(V^\star)^2(s) \right) + \frac{1}{7V_\gamma^\uparrow}\left( (S_t)^2 - \left(\widehat{V}^t(s) + S_t\right)^2 \right),$$

where $\beta^t(s,a)$:

$$\beta^t(s,a) = P_U^t(s,a)\eta^t \mathcal{E}^t(s,a) + \beta_1^t(s,a) + (1 - P_U^t(s))\frac{V_\gamma^\uparrow \ell_1}{\lambda_t N^t(s,a)}$$

$$\widehat{V}^t(s) \coloneqq V^t(s) - V^\star(s)$$

$$S_t \coloneqq \left(\frac{3}{2} - P_U^{t,\star}(s)\right)\lambda_t V_\gamma^\uparrow + \Phi_t,$$

in which

$$\beta_1^t(s,a) \coloneqq \frac{1}{N^t(s,a)}\left( \frac{3V_\gamma^\uparrow \ell_1}{\lambda_t} + 30V_\gamma^\uparrow S\ell_{3,t}(s,a) \right).$$

**Definition 6.** Let operator $\mathcal{T}_2$ be defined by

$$(\mathcal{T}_2 V)(s) \coloneqq \Delta_\gamma(D_\gamma^t)(s, \pi_t(s)) + 2\beta^t(s, \pi_t(s)) + \gamma P(V)(s, \pi_t(s)).$$

**Definition 7.** $\mathcal{T}$ is affine if, for any vector $V$ and $E$

$$\mathcal{T}(V + E) = \mathcal{T}V + \gamma PE.$$

**Lemma 16.** *$\mathcal{T}_2$ is a contraction mapping, monotone, and affine.*

*Proof.* The argument for contraction and monotonicity is similar to that of proof of Lemma 12. For the affine part, we observe:

$$\begin{aligned}\mathcal{T}_2(V + E) &= \Delta_\gamma(D_\gamma^t) + 2\beta^t + \gamma P(V + E) \\ &= \Delta_\gamma(D_\gamma^t) + 2\beta^t + \gamma PV + \gamma PE \\ &= \mathcal{T}_2 V + \gamma PE.\end{aligned}$$

$\qquad\square$

**Lemma 17.** *Under high-probability event $\cap_{i=1}^4 \mathbf{A}_i^\gamma$, it holds that*

$$\Delta_\gamma\left(V^t - V^{\pi_t}\right)(s, \pi_t(a)) \leq \Delta_\gamma(D_\gamma^t)(s, \pi_t(a)) + 2\beta^t(s, \pi_t(a)).$$

*Proof.* The proof is similar to that of Lemma 35, except that it uses $\widehat{V}^t(s) + S_t \geq 0$ from Corollary 2 and Lemma 14, an application of Lemma 34, and the boundedness of the discounted value for variance decomposition, together with an adjustment of some constants under the event $\mathbf{A}_3^\gamma$. $\qquad\square$

Now we prove Lemma 15.

*Proof of Lemma 15.* Denote $\Delta V := V^t - V^{\pi_t}$.

Note, $\Delta V(s) = \Delta_\gamma (\Delta V)(s, \pi_t(s)) + \gamma P (\Delta V)(s, \pi_t(s))$. By Lemma 17, we have

$$\Delta V \leq \mathcal{T}_2 \Delta V. \qquad\qquad \textbf{Condition 1}$$

For brevity, we denote $g := \beta^t + \gamma P J^t$, $f := \min\{g, V_\gamma^\uparrow\}$, and $D := D_\gamma^t$. We observe

$$\mathcal{T}_2(D + 2f) = D + 2g. \qquad\qquad \textbf{Condition 2}$$

Moreover, since $D \geq -\frac{6}{7} V_\gamma^\uparrow$ and $\Delta V \leq V_\gamma^\uparrow$, we have

$$\Delta V \leq D + 2V_\gamma^\uparrow. \qquad\qquad \textbf{Condition 3}$$

Now, we claim $\Delta V \leq D + 2f$ is true. We consider two cases:

**Case 1:** $g(s) \geq V_\gamma^\uparrow$ For any state $s$ where $g(s) \geq V_\gamma^\uparrow$, the function $f(s)$ is defined as $f(s) = V_\gamma^\uparrow$. The inequality we want to prove becomes $\Delta V(s) \leq D(s) + 2V_\gamma^\uparrow$, which is true by **Condition 3**.

**Case 2:** $g(s) < V_\gamma^\uparrow$ For states where $g(s) < V_\gamma^\uparrow$, the function $f(s)$ is now defined as $f(s) = g(s)$. We prove by contradiction. Assume there is at least one state $s$ where $g(s) < V_\gamma^\uparrow$ and the desired inequality is false.

We define an "error" function $E := \Delta V - (D + 2f)$. By the assumption, the set of states $\Xi := \{s \in \mathcal{S} : E(s) > 0\}$ is non-empty. Let $E^\star := \sup_{s \in \Xi} E(s)$, then $E^\star > 0$.

We start with **Condition 1**, that is, $\Delta V \leq T \Delta V$, and substitute $\Delta V = E + (D + 2f)$, we get

$$E + (D + 2f) \leq \mathcal{T}_2(E + D + 2f)$$

By the affinity in Lemma 16, we can write $\mathcal{T}_2(E + D + 2f) = \mathcal{T}_2(D + 2f) + \gamma P E$. By **Condition 2**, we have $\mathcal{T}_2(D + 2f) = D + 2g$. Combining this with Equation G.3, we obtain:

$$E + (D + 2f) \leq (D + 2g) + \gamma P E.$$

Rearranging it, we get:
$$E \leq 2(g - f) + \gamma P E.$$

Now, let us consider a state $s^\star$ where the error is maximal, i.e. $E(s^\star) = E^\star$. It must hold that:

$$E(s^\star) \leq 2(g(s^\star) - f(s^\star)) + \gamma (PE)(s^\star)$$
$$\leq 2(g(s^\star) - f(s^\star)) + \gamma E(s^\star).$$

Thus, we get
$$(1 - \gamma)E(s^\star) \leq 2(g(s^\star) - f(s^\star)).$$

Since $g(s) = f(s)$ whenever $g(s) < V_\gamma^\uparrow$, the above equals to zero, implying $E(s^\star) \leq 0$. This leads to a contradiction. Therefore we conclude that $\Delta V \leq D + 2f$. $\qquad\square$

Bounding $D_\gamma^t(s)$ using steps similar to those in the proof of Lemma 5, we obtain:

**Corollary 3.** *For infinite-horizon discounted MDPs, under high-probability event $\cap_{i=1}^4 \mathbf{A}_i^\gamma$, it holds that for all $s \in \mathcal{S}, t \in \mathbb{N}$,*

$$V^t(s) - V^{\pi_t}(s) \leq \left(3 - 2P_U^t(s) - \frac{2}{7} P_U^{t,\star}(s)\right) \lambda_t V_\gamma^\uparrow + 2J_\gamma^t(s) + \mathcal{O}\left(\frac{\Phi_t^2}{V_\gamma^\uparrow} + \lambda_t \Phi_t\right).$$

### G.4 PER-STEP REGRET

**Theorem 9** (Restatement of Theorem 1). *For infinite-horizon discounted MDPs, under high-probability event $\mathcal{B}$, it holds that for all $s \in \mathcal{S}, t \in \mathbb{N}$,*

$$V^\star(s) - V^{\pi_t}(s) \leq \left(\frac{9}{2} - \mathfrak{R}^t(s)\right)\lambda_t V_\gamma^\uparrow + 2J_\gamma^t(s) + \mathcal{O}\left(\Phi_t\left(1 + \frac{\Phi_t}{V_\gamma^\uparrow}\right)\right).$$

*Proof.* Combining Corollaries 2 and 3 with Lemma 14 gives the desired result. $\qquad\square$

### G.5 BOUNDEDNESS OF $J_\gamma^t$

**Lemma 18** ((Lee & Oh, 2025)). *Let $C > 0$ be a constant and $\{X_t\}_{t=1}^\infty$ be a martingale difference sequence with respect to a filtration $\{\mathcal{F}_t\}_{t=0}^\infty$ with $X_t \leq C$ almost surely for all $t \in \mathbb{N}$. Then, for any $\lambda \in (0, 1]$ and $\delta \in (0, 1]$, the following inequality holds for all $n \in \mathbb{N}$ with probability at least $1 - \delta$:*

$$\sum_{t=1}^n X_t \leq \frac{3\lambda}{4C}\sum_{t=1}^n \mathbb{E}[X_t^2|\mathcal{F}_{t-1}] + \frac{C}{\lambda}\log\frac{1}{\delta}.$$

**Lemma 19.** *For any time $T$, we have*

$$\sum_{t=1}^T \mathbf{1}(t + \nu_t \neq T + 1) \leq SA\log_2 2T.$$

*Proof.* The general idea is similar to the proof of Lemma 30 in (Lee & Oh, 2025), but unlike the episodic setting, where episodes exhibit monotonicity, the infinite-horizon setting requires special consideration to handle coupled trajectories. By focusing on each individual state-action pair, we get:

$$\sum_{t=1}^T \mathbf{1}(t + \nu_t \neq T + 1) = \sum_{t=1}^T \sum_{(s,a)\in\mathcal{S}\times\mathcal{A}} \mathbf{1}(t + \nu_t \neq T + 1, (s_{t+\nu_t}, a_{t+\nu_t}) = (s, a))$$

$$= \sum_{(s,a)\in\mathcal{S}\times\mathcal{A}} \sum_{t=1}^T \mathbf{1}(t + \nu_t \neq T + 1, (s_{t+\nu_t}, a_{t+\nu_t}) = (s, a)).$$

If $t + \nu_t \neq T + 1$, then $t + \nu_t$ is the first time step more than double the anchor $t$. Therefore, we have $n^{(t+\nu_t)}(s_{t+\nu_t}, a_{t+\nu_t}) \geq 2N^t(s_{t+\nu_t}, a_{t+\nu_t}) + 1$. Since any step that is greater than $t + \nu_t$ is an inclusion of $n^{(t+\nu_t)}$, we have $N^{(t+\nu_t+c)}(s_{t+\nu_t}, a_{t+\nu_t}) \geq 2N^t(s_{t+\nu_t}, a_{t+\nu_t}) + 1$ for any $c \in \mathbb{N}$. Based on this condition, we denote $M_T(s, a)$ as the number of steps $t \in \{1, 2, \dots, T\}$ such that $N^{(t+\nu_t+c)}(s, a) \geq 2N^t(s, a) + 1$, then, we have:

$$\sum_{t=1}^T \mathbf{1}(t + \nu_t \neq T + 1, (s_{t+\nu_t}, a_{t+\nu_t}) = (s, a)) \leq M_T(s, a).$$

We aim to bound the right-hand side above by finding contradiction between a upper and lower bound of $N^{(t+\nu_t+c)}(s, a)$. First, since there are at most $(T - t + 1)$ time steps left from the anchor $t$, we have $N^{(t+\nu_t+c)}(s, a) \leq N^t(s, a) + (T - t + 1) \leq N^t(s, a) + T$. Combining this with $N^{(t+\nu_t+c)}(s, a) \geq 2N^t(s, a) + 1$, we know that it occurs only if $N^t(s, a) < T$. Next, we prove by induction that

$$P(t) : N^t(s, a) \geq 2^{M_{t-1}(s,a)} - 1.$$

To verify this, let $c = 1$ and define a sequence of "checkpoints" that starts with $t$:

$$t_0 := t, \qquad t_{k+1} := t_k + \nu_{t_k} + 1.$$

Because $\nu_{t_k} \geq 0$, we have $t_{k+1} \geq t_k + 1$. Also $\nu_{t_k} \leq T - t_k + 1$ and $t_k \geq t \geq 1$ give $t_{k+1} \leq T$. Hence $\{t_k\}$ is a strictly increasing sequence bounded above by $T$, so after at most $T - t_0$ steps we reach $t_K = T$. If the induction statement holds for any step $t \in [t_k, t_{k+1}]$, then, by the above

progress and termination argument, it follows that all steps are covered.

Let's first verity the base case $P(1)$, for which we have $M_0 = 0$ and $N^{(1)} = 0$, therefore the inequality holds. Then assume $P(t_0)$ holds, there are two cases to consider. If $N^{(t+\nu_t+1)}(s,a) \geq 2N^t(s,a) + 1$, it implies that $(s,a)$ is the first time step that triggers the stopping of $\nu_t$, leading to $N^{(t+\nu_t+1)}(s,a) \geq 2^{M_{t-1}(s,a)+1} - 1 = 2^{M_{t+\nu_t}(s,a)} - 1$. Moreover, for each intermediate step $l$ with $1 \leq l \leq \nu_t$, $P(t_0 + l)$ holds; On the other hand, if $(s,a)$ is not the pair that triggers $\nu_t$, this means that it has not been doubled yet, implying $M_{t+\nu_t}(s,a) = M_{t-1}(s,a)$. However, there may still be some increments, and therefore $N^{(t+\nu_t+1)}(s,a) \geq N^t(s,a) \geq 2^{M_{t-1}(s,a)} - 1 = 2^{M_{t+\nu_t}(s,a)} - 1$. Thus, we conclude that the induction holds.

This gives us a lower bound, suggesting $M_{t-1}(s,a)$ cannot grow faster than logarithmically in $T$. Formally, once $M_{t-1}(s,a)$ reaches $\lfloor \log_2 T \rfloor + 1$ for some $t$, it cannot increase further, since $N^t(s,a) < T$. Therefore, we conclude that $M_T(s,a) \leq \lfloor \log_2 T \rfloor + 1 \leq \log_2 2T$, which completes the proof. $\qquad\square$

**Lemma 20.** *For infinite-horizon discounted MDPs, under high-probability event $\mathbf{A}_5^\gamma \cap \mathbf{A}_6^\gamma$, it holds that*

$$\sum_{t=1}^T J_\gamma^t(s_t) \leq \sum_{t=1}^T \sum_{l=0}^{\nu_t-1} \gamma^l \beta^t(s_{t+l}, a_{t+l}) + \frac{13 V_\gamma^\uparrow}{1-\gamma} SA \log \frac{12T}{\delta},$$

*for all $T \in \mathbb{N}$.*

*Proof.*

**Bounding $J_\gamma^t(s_t)$**   Given a $T$-path $(s_1, a_1, r_1, \ldots, s_T, a_T, r_T, s_{T+1})$ where actions are chosen from $\pi_t(s_t)$ at each time step, we decompose $J_\gamma^t(s_t)$ as follows:

$$\begin{aligned}
J_\gamma^t(s_t) &\leq \beta^t(s_t, a_t) + \gamma P J^t(s_t, a_t) \\
&= \beta^t(s_t, a_t) + \gamma P J^t(s_t, a_t) - \gamma J^t(s_{t+1}) + \gamma J^t(s_{t+1}) \\
&\;\;\vdots \\
&\leq \sum_{l=0}^{\nu_t-1} \underbrace{\gamma^l \beta^t(s_{t+l}, a_{t+l})}_{\mathbf{S}_{1,l}} + \underbrace{\gamma^{l+1}\left(P J^t(s_{t+l}, a_{t+l}) - J^t(s_{t+l+1})\right)}_{\mathbf{S}_{2,l}} + \gamma^{\nu_t} J^t(s_{t+\nu_t}).
\end{aligned}$$

Then, we take summation over $T$ steps:

$$\sum_{t=1}^T J_\gamma^t(s_t) \leq \underbrace{\sum_{t=1}^T \sum_{l=0}^{\nu_t-1} \mathbf{S}_{1,l}}_{I_1} + \underbrace{\sum_{t=1}^T \sum_{l=0}^{\nu_t-1} \mathbf{S}_{2,l}}_{I_2} + \underbrace{\sum_{t=1}^T \gamma^{\nu_t} J^t(s_{t+\nu_t})}_{I_3}.$$

**Bounding $I_3$** By Lemma 19 and $J^t(s_{T+1}) \leq V_\gamma^\uparrow$, we get:

$$
\begin{aligned}
I_3 &= \sum_{t=1}^{T} \gamma^{\nu_t} J^t(s_{t+\nu_t}) \\
&= \sum_{t=1}^{T} \left( \mathbf{1}(t + \nu_t \neq T + 1) + \mathbf{1}(t + \nu_t = T + 1) \right) \gamma^{\nu_t} J^t(s_{t+\nu_t}) \\
&= \sum_{t=1}^{T} \mathbf{1}(t + \nu_t \neq T + 1)\gamma^{\nu_t} J^t(s_{t+\nu_t}) + \sum_{t=1}^{T} \mathbf{1}(t + \nu_t = T + 1)\gamma^{\nu_t} J^t(s_{t+\nu_t}) \\
&= \sum_{t=1}^{T} \mathbf{1}(t + \nu_t \neq T + 1)\gamma^{\nu_t} J^t(s_{t+\nu_t}) + \sum_{t=1}^{T} \mathbf{1}(t + \nu_t = T + 1)\gamma^{T-t+1} J^t(s_{T+1}) \\
&\leq \sum_{t=1}^{T} \mathbf{1}(t + \nu_t \neq T + 1)\gamma^{\nu_t} J^t(s_{t+\nu_t}) + \frac{V_\gamma^\uparrow}{1 - \gamma} \\
&\leq V_\gamma^\uparrow \sum_{t=1}^{T} \mathbf{1}(t + \nu_t \neq T + 1) + \frac{V_\gamma^\uparrow}{1 - \gamma} \\
&\leq V_\gamma^\uparrow SA \log_2 2T + \frac{V_\gamma^\uparrow}{1 - \gamma}
\end{aligned}
$$

**Bounding $I_2$**

$$
\begin{aligned}
I_2 &= \sum_{t=1}^{T} \sum_{l=0}^{\nu_t - 1} \mathbf{S}_{2,l} \\
&= \sum_{t=1}^{T} \sum_{l=0}^{T-t} \mathbf{1}(l \leq \nu_t - 1)\mathbf{S}_{2,l} \\
&= \sum_{t=1}^{T} \sum_{l=0}^{T-t} \mathbf{1}(l \leq \nu_t - 1)\gamma^{l+1} \left( P J^t(s_{t+l}, a_{t+l}) - J^t(s_{t+l+1}) \right) \\
&\stackrel{\text{(a)}}{=} \sum_{\tau=1}^{T} \sum_{t=1}^{\tau} \mathbf{1}(\tau - t \leq \nu_t - 1)\gamma^{\tau-t+1} \left( P J^t(s_\tau, a_\tau) - J^t(s_{\tau+1}) \right) \\
&\stackrel{\text{(b)}}{=} \sum_{t=1}^{T} \underbrace{\sum_{\tau=1}^{t} \mathbf{1}(t - \tau \leq \nu_\tau - 1)\gamma^{t-\tau+1} \left( P J^\tau(s_t, a_t) - J^\tau(s_{t+1}) \right)}_{:=X_t},
\end{aligned}
$$

where (**a**) is due to the exchange of rows and columns and (**b**) to the reverse of the roles of indexes.

Note that in the final step, we make a bag of martingale differences with the same time index; therefore, it is not hard to verify that $X_t$ is a martingale difference sequence, with $E[X_t|\mathcal{F}_t] = 0$, $E[(X_t)^2|\mathcal{F}_t] = \mathrm{Var}\left( \sum_{\tau=1}^{t} \mathbf{1}(t - \tau \leq \nu_\tau - 1)\gamma^{t-\tau+1} J^\tau(s_{t+1}) \right)(s_t, a_t)$ and bounded as $|X_t| \leq \frac{\gamma V_\gamma^\uparrow}{1-\gamma} \leq \frac{V_\gamma^\uparrow}{1-\gamma}$. Denoting $Y^t(s_{t+1}) := \sum_{\tau=1}^{t} \mathbf{1}(t - \tau \leq \nu_\tau - 1)\gamma^{t-\tau+1} J^\tau(s_{t+1})$ and applying Lemma 18

to $\{X_t\}_{t=1}^{\infty}$ with $\lambda = \frac{1}{6}$, we get the following:

$$I_2 = \sum_{t=1}^{T} X_t$$

$$\leq \frac{(1-\gamma)}{8V_\gamma^\uparrow} \underbrace{\sum_{t=1}^{T} \mathrm{Var}\left(Y^t(s_{t+1})\right)(s_t, a_t)}_{:=L} + \frac{6V_\gamma^\uparrow}{1-\gamma} \log \frac{6}{\delta}. \tag{9}$$

Next, we will bound the sum of variances $L$. First, we look at each individual variance.

$$
\begin{aligned}
L_t := \mathrm{Var}(Y^t(s_{t+1}))(s_t, a_t) &= P\left(Y^t(s_{t+1})\right)^2(s_t, a_t) - \left(PY^t(s_{t+1})(s_t, a_t)\right)^2 \\
&= P\left(Y^t(s_{t+1})\right)^2(s_t, a_t) - (Y^t(s_{t+1}))^2 + (Y^t(s_{t+1}))^2 - \left(PY^t(s_{t+1})(s_t, a_t)\right)^2 \\
&= P\left(Y^t(s_{t+1})\right)^2(s_t, a_t) - (Y^t(s_{t+1}))^2 \\
&\quad + \left(Y^t(s_{t+1}) + PY^t(s_{t+1})(s_t, a_t)\right) \cdot \left(Y^t(s_{t+1}) - PY^t(s_{t+1})(s_t, a_t)\right) \\
&\leq \underbrace{P\left(Y^t(s_{t+1})\right)^2(s_t, a_t) - (Y^t(s_{t+1}))^2}_{:=Z_t} + \frac{2V_\gamma^\uparrow}{1-\gamma}\left(Y^t(s_{t+1}) - PY^t(s_{t+1})(s_t, a_t)\right).
\end{aligned}
$$

Akin to the previous argument, the second term is a martingale difference sequence, therefore, we obtain:

$$\sum_{t=1}^{T} Y^t(s_{t+1}) - PY^t(s_{t+1})(s_t, a_t) \leq \frac{1-\gamma}{8V_\gamma^\uparrow} \sum_{t=1}^{T} \mathrm{Var}(Y^t(s_{t+1}))(s_t, a_t) + \frac{6V_\gamma^\uparrow}{1-\gamma} \log \frac{6}{\delta}.$$

Based on this, we can simplify the bounding on $L$:

$$
\begin{aligned}
L &= \sum_{t=1}^{T} L_t \\
&\leq \sum_{t=1}^{T} Z_t + \frac{2V_\gamma^\uparrow}{1-\gamma} \sum_{t=1}^{T} \left(Y^t(s_{t+1}) - PY^t(s_{t+1})(s_t, a_t)\right) \\
&\leq \sum_{t=1}^{T} Z_t + \frac{2V_\gamma^\uparrow}{1-\gamma} \left(\frac{1-\gamma}{8V_\gamma^\uparrow} \sum_{t=1}^{T} \mathrm{Var}(Y^t(s_{t+1}))(s_t, a_t) + \frac{6V_\gamma^\uparrow}{1-\gamma} \log \frac{6}{\delta}\right) \\
&= \sum_{t=1}^{T} Z_t + \frac{1}{4} \sum_{t=1}^{T} \mathrm{Var}(Y^t(s_{t+1}))(s_t, a_t) + \frac{12V_\gamma^{\uparrow 2}}{(1-\gamma)^2} \log \frac{6}{\delta} \\
&= \sum_{t=1}^{T} Z_t + \frac{1}{4} L + \frac{12V_\gamma^{\uparrow 2}}{(1-\gamma)^2} \log \frac{6}{\delta} \\
&\leq \sum_{t=1}^{T} Z_t + \frac{1}{4} L + \frac{12V_\gamma^{\uparrow 2}}{(1-\gamma)^2} \log \frac{6}{\delta}. \tag{10}
\end{aligned}
$$

It is not difficult to check that $\{Z_t\}_{t=1}^{\infty}$ is a martingale difference sequence, with $E[Z_t|\mathcal{F}_t] = 0$, $E[(Z_t)^2|\mathcal{F}_t] = \mathrm{Var}\left((Y^t(s_{t+1}))^2\right)(s_t, a_t)$ and bounded as $|Z_t| \leq \frac{V_\gamma^{\uparrow 2}}{(1-\gamma)^2}$. Moreover, by applying Lemma 9 in (Lee & Oh, 2025) to the second-order moment, we have:

$$\mathrm{Var}\left((Y^t(s_{t+1}))^2\right)(s_t, a_t) \leq \frac{4V_\gamma^{\uparrow 2}}{(1-\gamma)^2} \mathrm{Var}\left(Y^t(s_{t+1})\right)(s_t, a_t).$$

Combining this with Lemma 18 with $\lambda = \frac{1}{12}$, we get the following.

$$\sum_{t=1}^{T} Z_t \leq \frac{1}{4} \sum_{t=1}^{T} \text{Var}\left(Y^t(s_{t+1})\right)(s_t, a_t) + \frac{12 V_\gamma^{\uparrow 2}}{(1-\gamma)^2} \log \frac{6}{\delta}$$

$$= \frac{1}{4} L + \frac{12 V_\gamma^{\uparrow 2}}{(1-\gamma)^2} \log \frac{6}{\delta}$$

Substituting this into Eq. 10, we obtain:

$$L \leq \sum_{t=1}^{T} Z_t + \frac{1}{4} L + \frac{12 V_\gamma^{\uparrow 2}}{(1-\gamma)^2} \log \frac{6}{\delta}$$

$$\leq \frac{1}{4} L + \frac{12 V_\gamma^{\uparrow 2}}{(1-\gamma)^2} \log \frac{6}{\delta} + \frac{1}{4} L + \frac{12 V_\gamma^{\uparrow 2}}{(1-\gamma)^2} \log \frac{6}{\delta}$$

$$= \frac{1}{2} L + \frac{24 V_\gamma^{\uparrow 2}}{(1-\gamma)^2} \log \frac{6}{\delta},$$

which has a recursive structure, leading to:

$$L \leq \frac{48 V_\gamma^{\uparrow 2}}{(1-\gamma)^2} \log \frac{6}{\delta}.$$

Substituting this into 9, we have:

$$I_2 \leq \frac{(1-\gamma)}{8 V_\gamma^{\uparrow}} L + \frac{6 V_\gamma^{\uparrow}}{1-\gamma} \log \frac{6}{\delta}$$

$$\leq \frac{(1-\gamma)}{8 V_\gamma^{\uparrow}} \frac{48 V_\gamma^{\uparrow 2}}{(1-\gamma)^2} \log \frac{6}{\delta} + \frac{6 V_\gamma^{\uparrow}}{1-\gamma} \log \frac{6}{\delta}$$

$$\leq \frac{12 V_\gamma^{\uparrow}}{1-\gamma} \log \frac{6}{\delta}.$$

Finally, we conclude that

$$\sum_{t=1}^{T} J_\gamma^t(s_t) \leq I_1 + I_2 + I_3$$

$$\leq I_1 + \frac{12 V_\gamma^{\uparrow}}{1-\gamma} \log \frac{6}{\delta} + \left( V_\gamma^{\uparrow} SA \log_2 2T + \frac{V_\gamma^{\uparrow}}{1-\gamma} \right)$$

$$\leq I_1 + \frac{13 V_\gamma^{\uparrow}}{1-\gamma} \log \frac{6}{\delta} + V_\gamma^{\uparrow} SA \log_2 2T$$

$$\leq I_1 + \frac{13 V_\gamma^{\uparrow}}{1-\gamma} \log \frac{6}{\delta} + \frac{V_\gamma^{\uparrow}}{1-\gamma} SA \log_2 2T$$

$$\leq I_1 + \frac{13 V_\gamma^{\uparrow}}{1-\gamma} SA \log \frac{6}{\delta} + \frac{2 V_\gamma^{\uparrow}}{1-\gamma} SA \log 2T$$

$$\leq I_1 + \frac{13 V_\gamma^{\uparrow}}{1-\gamma} SA \log \frac{6}{\delta} + \frac{13 V_\gamma^{\uparrow}}{1-\gamma} SA \log 2T$$

$$= I_1 + \frac{13 V_\gamma^{\uparrow}}{1-\gamma} SA \log \frac{12T}{\delta}.$$

which completes the proof. □

**Lemma 21.** *For infinite-horizon discounted MDPs, under high-probability event* $\mathbf{A}_5^\gamma \cap \mathbf{A}_6^\gamma$, *it holds that*

$$\sum_{t=1}^{T} J_\gamma^t(s_t) \leq \frac{2 \mathcal{Y}^{(T)}}{1-\gamma} SA \log\left(1 + \frac{T}{SA}\right) + \frac{13 V_\gamma^{\uparrow}}{1-\gamma} SA \log \frac{12T}{\delta},$$

*for all* $T \in \mathbb{N}$.

*Proof.* Based on Lemma 20, we only need to bound $I_1$, which is a series of discounted sum of $\beta^t$. By the definition of the stopping time $\nu_t$, we know that for any $t$ that satisfies $t - \tau \le \nu_t - 1$, we have $n^t(s_t, a_t) \le 2N^\tau(s_t, a_t)$ looking back at a previous anchor $\tau$. Moreover, we infer that $n^t(s_t, a_t) \ge 2$ must hold, otherwise it cannot satisfy the condition. We denote the set $\mathcal{I}(s, a) \subseteq \{1, 2, \dots, T\}$ as the time steps at which the pair $(s, a)$ is encountered.

$$
\begin{aligned}
I_1 &= \sum_{t=1}^{T} \sum_{l=0}^{\nu_t - 1} \gamma^l \beta^t(s_{t+l}, a_{t+l}) \\
&= \sum_{t=1}^{T} \sum_{l=0}^{T-t} \mathbf{1}(l \le \nu_t - 1) \gamma^l \beta^t(s_{t+l}, a_{t+l}) \\
&= \sum_{\tau=1}^{T} \sum_{t=1}^{\tau} \mathbf{1}(\tau - t \le \nu_t - 1) \gamma^{\tau - t} \beta^t(s_\tau, a_\tau) \\
&= \sum_{t=1}^{T} \sum_{\tau=1}^{t} \mathbf{1}(t - \tau \le \nu_\tau - 1) \gamma^{t - \tau} \beta^\tau(s_t, a_t) \\
&= \sum_{t=1}^{T} \sum_{\tau=1}^{t} \mathbf{1}(t - \tau \le \nu_\tau - 1) \gamma^{t - \tau} \frac{\mathcal{Y}^\tau}{N^\tau(s_t, a_t)} \\
&\le \mathcal{Y}^{(T)} \sum_{t=1}^{T} \sum_{\tau=1}^{t} \mathbf{1}(t - \tau \le \nu_\tau - 1) \gamma^{t - \tau} \frac{1}{N^\tau(s_t, a_t)} \\
&\le \mathcal{Y}^{(T)} \sum_{t=1}^{T} \sum_{\tau=1}^{t} \mathbf{1}(t - \tau \le \nu_\tau - 1) \gamma^{t - \tau} \frac{2}{n^t(s_t, a_t)} \\
&\overset{\text{(a)}}{=} \mathcal{Y}^{(T)} \sum_{t=1}^{T} \sum_{\tau=1}^{t} \mathbf{1}(t - \tau \le \nu_\tau - 1) \mathbf{1}(n^t(s_t, a_t) \ge 2) \gamma^{t - \tau} \frac{2}{n^t(s_t, a_t)} \\
&\le 2\mathcal{Y}^{(T)} \sum_{t=1}^{T} \mathbf{1}(n^t(s_t, a_t) \ge 2) \frac{1}{n^t(s_t, a_t)} \sum_{\tau=1}^{t} \mathbf{1}(t - \tau \le \nu_\tau - 1) \gamma^{t - \tau} \\
&\le \frac{2\mathcal{Y}^{(T)}}{1 - \gamma} \sum_{(s,a) \in \mathcal{S} \times \mathcal{A}} \sum_{t \in \mathcal{I}(s,a)} \mathbf{1}(n^t(s, a) \ge 2) \frac{1}{n^t(s, a)} \\
&\le \frac{2\mathcal{Y}^{(T)}}{1 - \gamma} \sum_{(s,a) \in \mathcal{S} \times \mathcal{A}} \sum_{n=2}^{N^{T+1}(s,a)} \frac{1}{n} \\
&\le \frac{2\mathcal{Y}^{(T)}}{1 - \gamma} \sum_{(s,a) \in \mathcal{S} \times \mathcal{A}} \log\left(1 + N^{T+1}(s, a)\right) \\
&\le \frac{2\mathcal{Y}^{(T)}}{1 - \gamma} SA \log\left(1 + \frac{T}{SA}\right),
\end{aligned}
$$

where (a) holds because we can distinguish two cases:

- If $t - \tau > \nu_\tau - 1$, then the indicator $\mathbf{1}(t - \tau \le \nu_\tau - 1)$ is zero, so the product vanishes regardless of the other indicator;

- If $t - \tau \le \nu_\tau - 1$, then, as shown earlier, we have $n^t(s_t, a_t) \ge 2$.

$\square$

### G.6 LOWER BOUND OF EPISTEMIC RESISTANCE

*Proof.*

$$\sum_{t=1}^{T} P_U^t(s_t, a_t) = 1 + \frac{1}{\mathcal{E}_{\max}} \sum_{t=2}^{T} \frac{1}{\sqrt{N^t(s_t, a_t)}}$$

$$\geq 1 + \frac{1}{\mathcal{E}_{\max}} \sum_{t=2}^{T} \frac{1}{\sqrt{t-1}}$$

$$= 1 + \frac{1}{\mathcal{E}_{\max}} \sum_{t=1}^{T-1} \frac{1}{\sqrt{t}}$$

$$\geq 1 + \frac{1}{\mathcal{E}_{\max}} \sum_{t=1}^{T-1} \int_{t}^{t+1} \frac{1}{\sqrt{x}} \, dx$$

$$= 1 + \frac{1}{\mathcal{E}_{\max}} \int_{1}^{T} \frac{1}{\sqrt{x}} \, dx$$

$$= 1 + \frac{1}{\mathcal{E}_{\max}} \left( 2\sqrt{T} - 2 \right).$$

Note, this also holds for $P_U^{t,\star}(s_t, \pi^\star(s_t))$. Therefore, multiplying with $\frac{23}{7} \lambda V_\gamma^\uparrow$ completes the proof. $\qquad\square$

### G.7 REGRET ANALYSIS

#### G.7.1 PROOF OF THEOREM 2

Prior to deriving the regret, we state the following lemma.

**Lemma 22.** *It holds that*

$$\ell_{4,T} \leq \ell_1 + \ell_{2,T},$$

*for all $T \in \mathbb{N}$.*

*Proof.* Expand $\ell_{4,T}$ and relate it to $\ell_{2,T}$, we get:

$$\ell_{4,T} = \log \frac{12T}{\delta}$$

$$< \log \frac{12(SA + T)}{\delta}$$

$$= \log \frac{12SA(1 + \frac{T}{SA})}{\delta}$$

$$= \log \left( \frac{12SA}{\delta} \right) + \log \left( 1 + \frac{T}{SA} \right)$$

$$\leq \ell_1 + \ell_{2,T}.$$

$\qquad\square$

*Proof.* From Theorem 1, we have:

$$\text{Regret}(T) \leq \frac{9V_\gamma^\uparrow}{2} \sum_{t=1}^{T} \lambda_t - V_\gamma^\uparrow \sum_{t=1}^{T} \mathfrak{R}^t(s) \lambda_t + 2 \sum_{t=1}^{T} J_\gamma^t(s) + \sum_{t=1}^{T} \mathcal{O} \left( \Phi_t \left( 1 + \frac{\Phi_t}{V_\gamma^\uparrow} \right) \right)$$

Choose $\lambda_t = \min\{1, 3\sqrt{\frac{SA\ell_1\ell_{2,T}}{T(1-\gamma)}}\}$, $\forall t \in [T]$ and denote $\Psi(T) := \frac{2\sum_{t=1}^{T}\mathfrak{R}^t(s)}{9T}$, we have

$$\frac{9V_\gamma^\uparrow}{2}\sum_{t=1}^{T}\lambda_t - V_\gamma^\uparrow\sum_{t=1}^{T}\mathfrak{R}^t(s)\lambda_t = \frac{9V_\gamma^\uparrow}{2}\left(1 - \frac{2\sum_{t=1}^{T}\mathfrak{R}^t(s)}{9T}\right)T\min\{1, 3\sqrt{\frac{SA\ell_1\ell_{2,T}}{T(1-\gamma)}}\}$$

$$\leq 14\frac{R_{\max}}{1-\gamma}\left(1-\Psi(T)\right)T\sqrt{\frac{SA\ell_1\ell_{2,T}}{T(1-\gamma)}}$$

$$= 14\left(1-\Psi(T)\right)\frac{R_{\max}}{(1-\gamma)^{1.5}}\sqrt{SAT\ell_1\ell_{2,T}}.$$

From Lemma 21, we know that

$$2\sum_{t=1}^{T}J_\gamma^t(s_1^t) \leq \frac{4\mathcal{V}^{(T)}}{1-\gamma}SA\log\left(1+\frac{T}{SA}\right) + \frac{26V_\gamma^\uparrow}{1-\gamma}SA\log\frac{12T}{\delta}$$

$$\leq \frac{48V_\gamma^\uparrow SA\ell_{1,T}\ell_{2,T}}{(1-\gamma)\lambda_T} + \frac{120V_\gamma^\uparrow}{1-\gamma}S^2A\ell_{2,T}\ell_{3,T} + \frac{26V_\gamma^\uparrow}{1-\gamma}SA\log\frac{12T}{\delta}.$$

$$\leq \frac{48V_\gamma^\uparrow SA\ell_{1,T}\ell_{2,T}}{(1-\gamma)}\max\left\{1, \frac{1}{3}\sqrt{\frac{T(1-\gamma)}{SA\ell_1\ell_{2,T}}}\right\} + \frac{120V_\gamma^\uparrow}{1-\gamma}S^2A\ell_{2,T}\ell_{3,T} + \frac{26V_\gamma^\uparrow}{1-\gamma}SA\log\frac{12T}{\delta}.$$

$$\leq \frac{48R_{\max}SA\ell_{1,T}\ell_{2,T}}{(1-\gamma)^2} + \frac{16R_{\max}}{(1-\gamma)^{1.5}}\sqrt{SAT\ell_1\ell_{2,T}} + \frac{120R_{\max}}{(1-\gamma)^2}S^2A\ell_{2,T}\ell_{3,T} + \frac{26R_{\max}}{(1-\gamma)^2}SA\log\frac{12T}{\delta}.$$

$$\overset{(a)}{\leq} \frac{16R_{\max}}{(1-\gamma)^{1.5}}\sqrt{SAT\ell_1\ell_{2,T}} + \left(\frac{120R_{\max}}{(1-\gamma)^2}S^2A\ell_{2,T}\ell_{3,T} + \frac{48R_{\max}SA\ell_{1,T}\ell_{2,T}}{(1-\gamma)^2} + \frac{26R_{\max}}{(1-\gamma)^2}SA(\ell_{1,T}+\ell_{2,T})\right)$$

$$\overset{(b)}{\leq} \frac{16R_{\max}}{(1-\gamma)^{1.5}}\sqrt{SAT\ell_1\ell_{2,T}} + \left(\frac{120R_{\max}}{(1-\gamma)^2}S^2A\ell_{2,T}\ell'_{1,T} + \frac{48R_{\max}SA\ell'_{1,T}\ell_{2,T}}{(1-\gamma)^2} + \frac{26R_{\max}}{(1-\gamma)^2}SA(\ell'_{1,T}+\ell_{2,T})\right)$$

$$\leq \frac{16R_{\max}}{(1-\gamma)^{1.5}}\sqrt{SAT\ell_1\ell_{2,T}} + \left(\frac{120R_{\max}}{(1-\gamma)^2}S^2A\ell_{2,T}\ell'_{1,T} + \frac{48R_{\max}SA\ell'_{1,T}(1+\ell_{2,T})}{(1-\gamma)^2} + \frac{26R_{\max}}{(1-\gamma)^2}SA\ell_{2,T}\right)$$

$$\leq \frac{16R_{\max}}{(1-\gamma)^{1.5}}\sqrt{SAT\ell_1\ell_{2,T}} + \left(\frac{120R_{\max}}{(1-\gamma)^2}S^2A\ell_{2,T}(1+\ell'_{1,T}) + \frac{48R_{\max}SA\ell'_{1,T}(1+\ell_{2,T})}{(1-\gamma)^2}\right)$$

$$\leq \frac{16R_{\max}}{(1-\gamma)^{1.5}}\sqrt{SAT\ell_1\ell_{2,T}} + \frac{168R_{\max}}{(1-\gamma)^2}S^2A(1+\ell'_{1,T})(1+\ell_{2,T}),$$

where (a) uses the Lemma 22, and (b) $\ell'_{1,T}$ is denoted as $\log\frac{24SA(1+\log T)}{\delta}$, therein $\ell'_{1,T} \geq \ell_1$ and $\ell'_{1,T} \geq \ell_{3,T}$.

Combining these two together, we get:

$$\text{Regret}(T) \leq (30 - 14\Psi(T))R_{\max}\frac{\sqrt{SAT\ell_1\ell_{2,T}}}{(1-\gamma)^{1.5}} + 168R_{\max}\frac{S^2A}{(1-\gamma)^2}(1+\ell'_{1,T})(1+\ell_{2,T}) + \sum_{t=1}^{T}\mathcal{O}\left(\Phi_t\left(1+\frac{\Phi_t}{V_\gamma^\uparrow}\right)\right),$$

where only the last part left to resolve.

Given $\Phi_t = R_{\max}\lambda_t$, we have one additional source of $\mathcal{O}(\lambda T)$, which will be merged into the leading term. In addition, note that

$$\sum_{t=1}^{T}\mathcal{O}(\frac{\Phi_t^2}{V_\gamma^\uparrow}) = \sum_{t=1}^{T}\widetilde{\mathcal{O}}\left(\frac{R_{\max}^2SA}{TV_\gamma^\uparrow}\right)$$

$$= \widetilde{\mathcal{O}}\left(\frac{R_{\max}^2SA}{V_\gamma^\uparrow}\right)$$

$$\leq \widetilde{\mathcal{O}}(R_{\max}SA),$$

which only increases the non-leading term by some constants. So overall, we have:

$$\text{Regret}(T) = \widetilde{\mathcal{O}}\left(\frac{\sqrt{SAT}}{(1-\gamma)^{1.5}} + \frac{S^2 A}{(1-\gamma)^2}\right).$$

$\square$

### G.7.2 STATE-ACTION DEPENDENT $\lambda_t(s,a)$

**Definition 8.** Let $\mathcal{G}$ be defined by

$$\mathcal{G} = \sum_{(s,a)\in\mathcal{S}\times\mathcal{A}} \left(\sqrt{1 - \frac{46}{63}\bar{P}_U(s,a)}\right),$$

where

$$\bar{P}_U^\tau(s,a) := \min\{P_U^\tau(s,a), P_U^{\tau,\star}(s)\}$$
$$\bar{P}_U(s,a) := \min_{2\leq n\leq N^{T+1}(s,a)} \min_{1\leq\tau\leq t_{(s,a)}(n)} \bar{P}_U^\tau(s,a)$$

Notably, we have the property of $\mathcal{G}$ that $\frac{17}{63}SA \leq \mathcal{G} \leq SA$. The maximum is attained only if $\bar{P}_U(s,a) \equiv 0, \forall(s,a) \in \mathcal{S}\times\mathcal{A}$.

If the epistemic uncertainty is non-increasing, then $\bar{P}_U(s,a)$ is corresponding to exactly the epistemic uncertainty at the end of learning, that is, $\bar{P}_U^{T+1}(s,a)$, reflecting the systematic uncertainty of a particular state-action.

**Lemma 23.** *Denote* $\rho^t(s,a) := \frac{\sqrt{\frac{9}{2}-\Re^t(s,a)}\ell_{1,t}}{N^t(s,a)}$, *it holds that*

$$\sum_{t=1}^{T}\sum_{l=0}^{\nu_t-1} \gamma^l \rho^t(s_{t+l}, a_{t+l}) \leq \frac{3\sqrt{2}\ell_{1,T}}{(1-\gamma)}\mathcal{G}\log\left(1 + \frac{T}{\mathcal{G}}\right).$$

*Proof.* Denote $I := \sum_{t=1}^{T} \sum_{l=0}^{\nu_t - 1} \gamma^l \rho^t(s_{t+l}, a_{t+l})$, we have

$$I = \sum_{t=1}^{T} \sum_{\tau=1}^{t} \mathbf{1}(t - \tau \le \nu_\tau - 1) \gamma^{t-\tau} \frac{\ell_{1,\tau}}{N^\tau(s_t, a_t)} \left( \sqrt{\frac{9}{2} - \mathfrak{R}^\tau(s_t, a_t)} \right)$$

$$\le \sum_{t=1}^{T} \sum_{\tau=1}^{t} \mathbf{1}(t - \tau \le \nu_\tau - 1) \gamma^{t-\tau} \frac{2\ell_{1,\tau}}{n^t(s_t, a_t)} \left( \sqrt{\frac{9}{2} - \mathfrak{R}^\tau(s_t, a_t)} \right)$$

$$\overset{(a)}{\le} 2\ell_{1,T} \sum_{t=1}^{T} \sum_{\tau=1}^{t} \mathbf{1}(t - \tau \le \nu_\tau - 1) \gamma^{t-\tau} \frac{1}{n^t(s_t, a_t)} \left( \sqrt{\frac{9}{2} - \mathfrak{R}^\tau(s_t, a_t)} \right)$$

$$\overset{(b)}{\le} 3\sqrt{2}\ell_{1,T} \sum_{t=1}^{T} \sum_{\tau=1}^{t} \mathbf{1}(t - \tau \le \nu_\tau - 1) \gamma^{t-\tau} \frac{1}{n^t(s_t, a_t)} \left( \sqrt{1 - \frac{46}{63} \bar{P}_U^\tau(s_t, a_t)} \right)$$

$$= 3\sqrt{2}\ell_{1,T} \sum_{t=1}^{T} \sum_{\tau=1}^{t} \mathbf{1}(t - \tau \le \nu_\tau - 1)\mathbf{1}(n^t(s_t, a_t) \ge 2) \gamma^{t-\tau} \frac{1}{n^t(s_t, a_t)} \left( \sqrt{1 - \frac{46}{63} \bar{P}_U^\tau(s_t, a_t)} \right)$$

$$\le 3\sqrt{2}\ell_{1,T} \sum_{t=1}^{T} \mathbf{1}(n^t(s_t, a_t) \ge 2) \frac{1}{n^t(s_t, a_t)} \sum_{\tau=1}^{t} \mathbf{1}(t - \tau \le \nu_\tau - 1)\gamma^{t-\tau} \left( \sqrt{1 - \frac{46}{63} \bar{P}_U^\tau(s_t, a_t)} \right)$$

$$\le 3\sqrt{2}\ell_{1,T} \sum_{t=1}^{T} \mathbf{1}(n^t(s_t, a_t) \ge 2) \frac{1}{n^t(s_t, a_t)} \left( \sqrt{1 - \frac{46}{63} \min_{1 \le \tau \le t} \bar{P}_U^\tau(s_t, a_t)} \right) \sum_{\tau=1}^{t} \mathbf{1}(t - \tau \le \nu_\tau - 1)\gamma^{t-\tau}$$

$$\le \frac{3\sqrt{2}\ell_{1,T}}{(1-\gamma)} \sum_{t=1}^{T} \mathbf{1}(n^t(s_t, a_t) \ge 2) \frac{1}{n^t(s_t, a_t)} \left( \sqrt{1 - \frac{46}{63} \min_{1 \le \tau \le t} \bar{P}_U^\tau(s_t, a_t)} \right)$$

$$\le \frac{3\sqrt{2}\ell_{1,T}}{(1-\gamma)} \sum_{(s,a)\in\mathcal{S}\times\mathcal{A}} \sum_{t\in\mathcal{I}(s,a)} \mathbf{1}(n^t(s, a) \ge 2) \frac{\left( \sqrt{1 - \frac{46}{63} \min_{1 \le \tau \le t} \bar{P}_U^\tau(s, a)} \right)}{n^t(s, a)}$$

$$\le \frac{3\sqrt{2}\ell_{1,T}}{(1-\gamma)} \sum_{(s,a)\in\mathcal{S}\times\mathcal{A}} \sum_{n=2}^{N^{T+1}(s,a)} \frac{\left( \sqrt{1 - \frac{46}{63} \min_{1 \le \tau \le t_{(s,a)}(n)} \bar{P}_U^\tau(s, a)} \right)}{n}$$

$$\le \frac{3\sqrt{2}\ell_{1,T}}{(1-\gamma)} \sum_{(s,a)\in\mathcal{S}\times\mathcal{A}} \left( \sqrt{1 - \frac{46}{63} \min_{2 \le n \le N^{T+1}(s,a)} \min_{1 \le \tau \le t_{(s,a)}(n)} \bar{P}_U^\tau(s, a)} \right) \sum_{n=2}^{N^{T+1}(s,a)} \frac{1}{n}$$

$$= \frac{3\sqrt{2}\ell_{1,T}}{(1-\gamma)} \sum_{(s,a)\in\mathcal{S}\times\mathcal{A}} \left( \sqrt{1 - \frac{46}{63} \bar{P}_U(s, a)} \right) \log\left(1 + N^{T+1}(s, a)\right)$$

$$\le \frac{3\sqrt{2}\ell_{1,T}}{(1-\gamma)} \mathcal{G} \log \left( \sum_{(s,a)\in\mathcal{S}\times\mathcal{A}} \frac{\left( \sqrt{1 - \frac{46}{63} \bar{P}_U(s, a)} \right)\left(1 + N^{T+1}(s, a)\right)}{\mathcal{G}} \right)$$

$$= \frac{3\sqrt{2}\ell_{1,T}}{(1-\gamma)} \mathcal{G} \log \left( 1 + \sum_{(s,a)\in\mathcal{S}\times\mathcal{A}} \frac{\left( \sqrt{1 - \frac{46}{63} \bar{P}_U(s, a)} \right)\left(N^{T+1}(s, a)\right)}{\mathcal{G}} \right)$$

$$\le \frac{3\sqrt{2}\ell_{1,T}}{(1-\gamma)} \mathcal{G} \log \left( 1 + \sum_{(s,a)\in\mathcal{S}\times\mathcal{A}} \frac{N^{T+1}(s, a)}{\mathcal{G}} \right)$$

$$\overset{(c)}{\le} \frac{3\sqrt{2}\ell_{1,T}}{(1-\gamma)} \mathcal{G} \log \left( 1 + \frac{T}{\mathcal{G}} \right),$$

where we have used the following facts:

(a) Monotonicity of $\ell_{1,\tau}$

(b) $\mathfrak{R}^\tau(s_t, a_t) \geq \frac{23}{7} \bar{P}_U^\tau(s_t, a_t)$

(c) Jensen's inequality

$\square$

**Lemma 24.** *For any $T \in \mathbb{N}$ and $\mathbf{x} \in [a,b]^N$, $0 < a < b$, define function $G(\mathbf{x}) = \sum\limits_{n=1}^{N} \sqrt{x_n}$ and $f(\mathbf{x}) = G(\mathbf{x}) \log\left(1 + \frac{T}{G(\mathbf{x})}\right)$, we have that*

$$f(\mathbf{1}a) \leq f(\mathbf{x}) \leq f(\mathbf{1}b).$$

*Proof.* Using the elementary fact that $g(u) = u \log\left(1 + \frac{T}{u}\right)$, $u > 0$ is nondecreasing on $(0, \infty)$ completes the proof. $\square$

*Proof.* From Theorem 1, we have:

$$\text{Regret}(T) \leq \frac{9V_\gamma^\uparrow}{2} \sum_{t=1}^{T} \lambda_t - V_\gamma^\uparrow \sum_{t=1}^{T} \mathfrak{R}^t(s)\lambda_t + 2\sum_{t=1}^{T} J_\gamma^t(s) + \sum_{t=1}^{T} \mathcal{O}\left(\Phi_t\left(1 + \frac{\Phi_t}{V_\gamma^\uparrow}\right)\right)$$

$$= V_\gamma^\uparrow \sum_{t=1}^{T} \left(\frac{9}{2} - \mathfrak{R}^t(s)\right)\lambda_t + 2\sum_{t=1}^{T} J_\gamma^t(s) + \sum_{t=1}^{T} \mathcal{O}\left(\Phi_t\left(1 + \frac{\Phi_t}{V_\gamma^\uparrow}\right)\right).$$

Choosing $\lambda_t = \min\left\{1, \frac{C}{\sqrt{\frac{9}{2} - \mathfrak{R}^t(s)}} \sqrt{\frac{SA\ell_{1,T}\ell_{2,T}}{T(1-\gamma)}}\right\}$, $\forall t \in [T]$, we have

$$V_\gamma^\uparrow \sum_{t=1}^{T} \left(\frac{9}{2} - \mathfrak{R}^t(s)\right)\lambda_t = V_\gamma^\uparrow \sum_{t=1}^{T} \left(\frac{9}{2} - \mathfrak{R}^t(s)\right) \min\left\{1, \frac{C}{\sqrt{\frac{9}{2} - \mathfrak{R}^t(s)}} \sqrt{\frac{SA\ell_{1,T}\ell_{2,T}}{T(1-\gamma)}}\right\}$$

$$\leq C\frac{R_{\max}}{1-\gamma} \sum_{t=1}^{T} \left(\sqrt{\frac{9}{2} - \mathfrak{R}^t(s)}\right) \sqrt{\frac{SA\ell_{1,T}\ell_{2,T}}{T(1-\gamma)}}$$

$$\leq C\frac{R_{\max}}{1-\gamma} \sqrt{T\left(\frac{9}{2}T - \sum_{t=1}^{T} \mathfrak{R}^t(s)\right)} \sqrt{\frac{SA\ell_{1,T}\ell_{2,T}}{T(1-\gamma)}}$$

$$\leq \underbrace{\frac{CR_{\max}}{(1-\gamma)^{1.5}} \sqrt{\left(\frac{9}{2}T - \sum_{t=1}^{T} \mathfrak{R}^t(s)\right) SA\ell_{1,T}\ell_{2,T}}}_{:=\mathcal{J}_1\left(\sum\limits_{t=1}^{T} \mathfrak{R}^t(s)\right)}.$$

Given

$$\mathcal{Y}^t = \frac{12V_\gamma^\uparrow \ell_{1,t}}{\lambda_t} + 30V_\gamma^\uparrow S\ell_{3,t}$$

$$= 12V_\gamma^\uparrow \ell_{1,t} \max\left\{1, \frac{\sqrt{\frac{9}{2} - \mathfrak{R}^t(s)}}{C} \sqrt{\frac{T(1-\gamma)}{SA\ell_{1,T}\ell_{2,T}}}\right\} + 30V_\gamma^\uparrow S\ell_{3,t}$$

$$= \underbrace{12V_\gamma^\uparrow \ell_{1,t}}_{\mathcal{Y}_1^t} + \underbrace{\frac{12}{C}V_\gamma^\uparrow \ell_{1,t} \sqrt{\frac{9}{2} - \mathfrak{R}^t(s)} \sqrt{\frac{T(1-\gamma)}{SA\ell_{1,T}\ell_{2,T}}}}_{\mathcal{Y}_2^t} + \underbrace{30V_\gamma^\uparrow S\ell_{3,t}}_{\mathcal{Y}_3^t}.$$

From Lemmas 20 and 23, we get

$$
\begin{aligned}
2\sum_{t=1}^{T} J_\gamma^t(s_1^t) &\le 2\sum_{t=1}^{T}\sum_{l=0}^{\nu_t-1}\gamma^l\beta^t(s_{t+l},a_{t+l}) + \frac{26V_\gamma^\uparrow}{1-\gamma}SA\log\frac{12T}{\delta}\\
&\le 2\sum_{t=1}^{T}\sum_{\tau=1}^{t}\mathbf{1}(t-\tau\le\nu_\tau-1)\gamma^{t-\tau}\frac{\mathcal{Y}^\tau}{N^\tau(s_t,a_t)} + \frac{26V_\gamma^\uparrow}{1-\gamma}SA\log\frac{12T}{\delta}\\
&\le 2\sum_{t=1}^{T}\sum_{\tau=1}^{t}\mathbf{1}(t-\tau\le\nu_\tau-1)\gamma^{t-\tau}\frac{1}{N^\tau(s_t,a_t)}\left(\mathcal{Y}_1^\tau+\mathcal{Y}_2^\tau+\mathcal{Y}_3^\tau\right) + \frac{26V_\gamma^\uparrow}{1-\gamma}SA\log\frac{12T}{\delta}\\
&\le \frac{48V_\gamma^\uparrow SA\ell_{1,T}\ell_{2,T}}{(1-\gamma)\lambda_T}\\
&\quad + \underbrace{\frac{72\sqrt{2}V_\gamma^\uparrow\ell_{1,T}}{C(1-\gamma)}\sqrt{\frac{T(1-\gamma)}{SA\ell_{1,T}\ell_{2,T}}}\left(\mathcal{G}\ell_{2,T}'\right)}_{:=\mathcal{J}_2(\mathcal{G})}\\
&\quad + \frac{120V_\gamma^\uparrow}{1-\gamma}S^2A\ell_{2,T}\ell_{3,T}\\
&\quad + \frac{26V_\gamma^\uparrow}{1-\gamma}SA\log\frac{12T}{\delta}\\
&\le \frac{72\sqrt{2}R_{\max}}{C(1-\gamma)^{1.5}}\sqrt{SAT\ell_{1,T}\ell_{2,T}} + \frac{168R_{\max}}{(1-\gamma)^2}S^2A\ell_{1,T}(1+\ell_{2,T})
\end{aligned}
$$

Combining everything together, we get:

$$
\text{Regret}(T) \le \mathcal{J}_1\left(\sum_{t=1}^{T}\mathfrak{R}^t(s)\right) + \mathcal{J}_2(\mathcal{G}) + \frac{168R_{\max}}{(1-\gamma)^2}S^2A\ell_{1,T}(1+\ell_{2,T}) + \sum_{t=1}^{T}\mathcal{O}\left(\Phi_t\left(1+\frac{\Phi_t}{V_\gamma^\uparrow}\right)\right).
$$

So, depending on the contribution of $\sum_{t=1}^{T}\mathfrak{R}^t(s)$ and $\mathcal{G}$, we can get different bounds. In what will follow, we choose $C = 3\sqrt{\frac{9}{2}}$.

**Disregarding** $\mathcal{G}$   Even ignoring the first part, we can obtain a tighter bound where the leading term is offset by the sum of epistemic resistance $\sum_{t=1}^{T}\mathfrak{R}^t(s)$ as follows:

$$
\begin{aligned}
\text{Regret}(T) &\le \left(16+14\sqrt{\left(1-\frac{2\sum_{t=1}^{T}\mathfrak{R}^t(s)}{9T}\right)}\right)\frac{R_{\max}}{(1-\gamma)^{1.5}}\sqrt{SAT\ell_{1,T}\ell_{2,T}} + \frac{168R_{\max}}{(1-\gamma)^2}S^2A\ell_{1,T}(1+\ell_{2,T})\\
&\quad + \sum_{t=1}^{T}\mathcal{O}\left(\Phi_t\left(1+\frac{\Phi_t}{V_\gamma^\uparrow}\right)\right),
\end{aligned}
$$

**Considering Both**   Let $N = SA, a = \frac{17}{63}, b = 1$, by Lemma 24, we know that

$$
\ell_{2,T}'^\star =: \sqrt{\frac{17}{63}}SA\log\left(1+\frac{T}{\sqrt{\frac{17}{63}}SA}\right) \le \mathcal{G}\ell_{2,T}' \le SA\log\left(1+\frac{T}{SA}\right),
$$

with this, at best, we can achieve:

$$
\text{Regret}^\star(T) \le \sqrt{\frac{17}{63}} \left( 16 \sqrt{\frac{{\ell'_{2,T}}^{\star 2}}{\ell_{2,T}}} + 14\sqrt{\ell_{2,T}} \right) \frac{R_{\max}}{(1-\gamma)^{1.5}} \sqrt{SAT\ell_{1,T}} + \frac{168 R_{\max}}{(1-\gamma)^2} S^2 A\ell_{1,T}(1 + \ell_{2,T})
$$
$$
+ \sum_{t=1}^{T} \mathcal{O}\left( \Phi_t \left( 1 + \frac{\Phi_t}{V_\gamma^\uparrow} \right) \right).
$$

If the ratio $\frac{T}{SA}$ is large, then $\ell'^\star_{2,T} \simeq \ell_{2,T}$. Therefore, the overall reduction is by a factor of $\sqrt{\frac{17}{63}} \approx 0.519$. In this case, we can improve the constant in the leading term by roughly one-half.

Lastly, the treatment of the part of $\Phi_t$ is similar to that in the uniform case. Therefore, we ultimately have

$$
\text{Regret}(T) = \widetilde{\mathcal{O}}\left( \frac{\sqrt{SAT}}{(1-\gamma)^{1.5}} + \frac{S^2 A}{(1-\gamma)^2} \right).
$$

$\square$

### G.8 SAMPLE COMPLEXITY

For infinite-horizon discounted MDPs, the sample complexity of an algorithm is defined as the number of non-$\epsilon$-optimal steps such that $V^{\pi_t}(s_t) \le V^\star(s_t) - \epsilon$ taken over the course of learning (Kakade, 2003; Strehl & Littman, 2008). If this sample complexity can be bounded by a polynomial function $f(S, A, \frac{1}{\epsilon}, \frac{1}{\delta}, \frac{1}{1-\gamma})$, then the algorithm is PAC-MDP. We are interested in proving PAC-MDP for the full range $\epsilon \in (0, V_\gamma^\uparrow]$.

From Theorem 1, we know that the per-step regret can be bounded as follows:

$$
V^\star(s_t) - V^{\pi_t}(s_t) \le \left( \frac{9}{2} - \mathfrak{R}^t(s_t) \right) \lambda_t V_\gamma^\uparrow + 2J_\gamma^t(s_t) + \Phi_t \left( 1 + \left(3 - 2P_U^{t,\star}(s)\right)\lambda_t + \frac{\Phi_t}{V_\gamma^\uparrow} \right)
$$
$$
\le \underbrace{\left( \frac{9}{2} - \mathfrak{R}^t(s_t) \right) \lambda_t V_\gamma^\uparrow}_{:=L_{1,t}} + \underbrace{2J_\gamma^t(s_t)}_{:=L_{2,t}} + \underbrace{\Phi_t\left( 4 + \frac{\Phi_t}{V_\gamma^\uparrow} \right)}_{:=L_{3,t}}
$$

For $L_{1,t}$, we can choose $\lambda_t = \frac{\epsilon}{18V_\gamma^\uparrow}$, so that we have $L_{1,t} \le \frac{\epsilon}{4}$. In addition, note that $\Phi_t = R_{\max}\lambda_t$ and $\lambda_t^2 = \frac{\epsilon^2}{18^2 V_\gamma^{\uparrow 2}} \le \frac{\epsilon}{18^2 V_\gamma^\uparrow}$, substituting it into $L_{3,t}$, we have:

$$
\Phi_t = R_{\max}\lambda_t = R_{\max}\frac{\epsilon}{18V_\gamma^\uparrow} = \frac{\epsilon(1-\gamma)}{18} \le \frac{\epsilon}{18}
$$
$$
\frac{\Phi_t^2}{V_\gamma^\uparrow} = \frac{R_{\max}^2\lambda_t^2}{V_\gamma^\uparrow} \le \frac{1}{18^2}\frac{R_{\max}^2\epsilon}{V_\gamma^{\uparrow 2}} = \frac{\epsilon(1-\gamma)^2}{18^2} \le \frac{\epsilon}{18^2}.
$$

Therefore, we obtain $L_{3,t} \le \left( \frac{4}{18} + \frac{1}{18^2} \right)\epsilon \le \frac{\epsilon}{4}$.

If we can prove that $L_{2,t} \le \frac{\epsilon}{2}$, or equivalently, $J_\gamma^t(s_t) \le \frac{\epsilon}{4}$, then the time step $t$ can be said to be optimal. To achieve this, we introduce a set of new notations that explicitly connect the number of non-optimal steps with $J_\gamma^t(s_t)$.

We define the set of non-optimal steps within $T$ total steps as $\Gamma_T := \{t \in [T] : J_\gamma^t(s_t) > \frac{\epsilon}{4}\}$, and its cardinality $|\Gamma_T|$. Then we want to prove that $|\Gamma_T|$ is polynomially bounded for all $T \in \mathbb{N}$.

For analyzing non-$\epsilon$-optimal steps, it is useful to overload the definition of visits so that it only includes those occurring in $\Gamma_T$.

$$n^t(s,a) := \sum_{t \in \Gamma_t} \mathbf{1}((s_t, a_t) = (s, a))$$

$$N^t(s,a) := \sum_{t \in \Gamma_{t-1}} \mathbf{1}((s_t, a_t) = (s, a))$$

$$\nu_t := \begin{cases} \min\{\tau \in [t, T] : n^\tau(s_\tau, a_\tau) > 2N^t(s_\tau, a_\tau)\}, & \text{if } \tau \text{ exists.} \\ T + 1, & \text{otherwise.} \end{cases}$$

Next, we bound the sum of $J_\gamma^t(s_t)$ but only for the steps in $\Gamma_T$.

**Lemma 25.** *For infinite-horizon discounted MDPs, under high-probability event $\mathbf{A}_7^\gamma \cap \mathbf{A}_8^\gamma$, it holds that*

$$\sum_{t \in \Gamma_T} J_\gamma^t(s_t) \leq \frac{2\mathcal{Y}^{(|\Gamma_T|)}}{1 - \gamma} SA \log\left(1 + \frac{|\Gamma_T|}{SA}\right) + \frac{13V_\gamma^\uparrow}{1 - \gamma} SA \log \frac{12|\Gamma_T|}{\delta},$$

*for all $T \in \mathbb{N}$.*

*Proof.* The proof follows the same procedure as in Lemmas 19–21, except adding the indicator function $\mathbf{1}(t \in \Gamma_T)$ to each time step. □

Based on the above result and Lemma 22, we can bound $|\Gamma_T|$ using the fact that $J_\gamma^t(s_t) > \frac{\epsilon}{4}$.

**Definition 9.** Let $W(T)$ be defined by

$$W(T) := \frac{1780R_{\max}^2 SA\ell_{1,T}\ell_{2,T}}{\epsilon^2(1-\gamma)^3} + \frac{240R_{\max}S^2 A\ell_{2,T}\ell_{3,T}}{\epsilon(1-\gamma)^2} + \frac{52R_{\max}SA\ell_{1,T}}{\epsilon(1-\gamma)^2}.$$

**Lemma 26.** *For infinite-horizon discounted MDPs, under high-probability event $\mathbf{D}$, it holds that*

$$|\Gamma_T| \leq W(|\Gamma_T|),$$

*for all $T \in \mathbb{N}$.*

*Proof.* From Lemmas 25 and 22, we get

$$|\Gamma_T| \leq \frac{8\mathcal{Y}^{(T)}SA\ell_{2,T}}{\epsilon(1-\gamma)} + \frac{52V_\gamma^\uparrow SA(\ell_{1,T} + \ell_{2,T})}{\epsilon(1-\gamma)}.$$

Substituting the definition of $\mathcal{Y}^{(T)}$ into the above, we have

$$|\Gamma_T| \leq \frac{1728R_{\max}^2 SA\ell_{1,|\Gamma_T|}\ell_{2,|\Gamma_T|}}{\epsilon^2(1-\gamma)^3} + \frac{240R_{\max}S^2 A\ell_{2,|\Gamma_T|}\ell_{3,|\Gamma_T|}}{\epsilon(1-\gamma)^2} + \frac{52R_{\max}SA(\ell_{1,|\Gamma_T|} + \ell_{2,|\Gamma_T|})}{\epsilon(1-\gamma)^2}$$

$$\overset{(a)}{\leq} \frac{1780R_{\max}^2 SA\ell_{1,|\Gamma_T|}\ell_{2,|\Gamma_T|}}{\epsilon^2(1-\gamma)^3} + \frac{240R_{\max}S^2 A\ell_{2,|\Gamma_T|}\ell_{3,|\Gamma_T|}}{\epsilon(1-\gamma)^2} + \frac{52R_{\max}SA\ell_{1,|\Gamma_T|}}{\epsilon(1-\gamma)^2},$$

where (a) uses the facts that $\frac{V_\gamma^\uparrow}{\epsilon} \geq 1$ and $\ell_{1,|\Gamma_T|} \geq 1$, therefore concludes the proof. □

**Proposition 3.** *For infinite-horizon discounted MDPs, let $T_0$ be defined as*

$$T_0 := \left\lfloor \frac{3670R_{max}^2 SA\ell_1\ell_{5,\epsilon}}{\epsilon^2(1-\gamma)^3} + \frac{480R_{max}S^2 A(2\ell_1 + \ell_{6,\epsilon})\ell_{5,\epsilon}}{\epsilon(1-\gamma)^2} \right\rfloor.$$

*Then the sample complexity of* `EUBRL` *is at most $T_0$ with probability at least $1 - \delta$.*

Before proving this result, we need to bound the the other way around i.e. $W(T_0) < T_0$.

**Lemma 27.** *It holds that*

$$W(T_0) < T_0.$$

*Proof.* Denote $B := \frac{R_{\max}^2 \ell_1}{\epsilon^2 (1-\gamma)^3} + \frac{R_{\max} S(2\ell_1 + \ell_{6,\epsilon})}{\epsilon(1-\gamma)^2}$, therefore we have $T_0 \leq 3670 BSA\ell_{5,\epsilon}$ where $\ell_{5,\epsilon} = \log\left(1 + 140B\right) \leq 5B$. Then we have:

$$\ell_{2,T_0} = \log\left(1 + \frac{T_0}{SA}\right)$$
$$\leq 2\ell_{5,\epsilon}.$$

Moreover, we have:

$$\ell_1 \leq \frac{4SA}{\delta}$$
$$\ell_{6,\epsilon} \leq \frac{V_\gamma^\uparrow}{\epsilon(1-\gamma)}.$$

Therefore, we get:

$$2\ell_1 + \ell_{6,\epsilon} \leq \frac{8SA}{\delta} + \frac{V_\gamma^\uparrow}{\epsilon(1-\gamma)}$$
$$\leq \frac{9V_\gamma^\uparrow SA}{\delta\epsilon(1-\gamma)}.$$

We use this to bound $B$ as follows:

$$B = \frac{R_{\max}^2 \ell_1}{\epsilon^2(1-\gamma)^3} + \frac{R_{\max} S(2\ell_1 + \ell_{6,\epsilon})}{\epsilon(1-\gamma)^2}$$
$$\leq \frac{4R_{\max}^2 SA}{\delta\epsilon^2(1-\gamma)^3} + \frac{9R_{\max}^2 S^2 A}{\delta\epsilon^2(1-\gamma)^4}$$
$$= \frac{13R_{\max}^2 S^2 A}{\delta\epsilon^2(1-\gamma)^4}.$$

With this, we now bound $\log T_0$, which is a part of $\ell_{3,T_0}$.

$$\log T_0 \leq \log 18350 B^2 SA$$
$$\leq \log 18350 \frac{169 R_{\max}^4 S^4 A^2}{\delta^2 \epsilon^4 (1-\gamma)^8} SA$$
$$= \log \frac{18350 \times 169}{e^4} e^4 \frac{R_{\max}^4 S^4 A^2}{\delta^2 \epsilon^4 (1-\gamma)^8} SA$$
$$\leq \underbrace{\log \frac{56800 S^5 A^3}{\delta^2}}_{:=L_1} + \log \frac{V_\gamma^{\uparrow 2} e^4}{\epsilon^4 (1-\gamma)^4}.$$

We now bound $L_1$.

$$L_1 = \log \frac{56800 S^5 A^3}{\delta^2}$$
$$\leq \log \frac{56800 S^5 A^5}{\delta^5}$$
$$\leq \log \frac{9^5 S^5 A^5}{\delta^5}$$
$$= 5 \log \frac{9SA}{\delta}$$
$$\leq 11 \frac{SA}{\delta}.$$

Therefore

$$\log T_0 \leq 11\frac{SA}{\delta} + 4\log \frac{V_\gamma^\uparrow e}{\epsilon(1-\gamma)}$$
$$\leq \frac{15SA}{\delta} \log \frac{V_\gamma^\uparrow e}{\epsilon(1-\gamma)}.$$

Then, substitute this into $\ell_{3,T_0}$, we get:

$$
\begin{aligned}
\ell_{3,T_0} &= \log \frac{12SA(1 + \log T_0)}{\delta} \\
&= \log \frac{12SA}{\delta} + \log\left(1 + \log T_0\right) \\
&\leq \ell_1 + \log\left(1 + \frac{15SA}{\delta}\log\frac{V_\gamma^\uparrow e}{\epsilon(1-\gamma)}\right) \\
&\overset{(a)}{\leq} \ell_1 + \log\left(\frac{16SA}{\delta}\log\frac{V_\gamma^\uparrow e}{\epsilon(1-\gamma)}\right) \\
&\leq \ell_1 + \log\left(\frac{16SA}{\delta}\right) + \log\log\frac{V_\gamma^\uparrow e}{\epsilon(1-\gamma)} \\
&\leq 2\ell_1 + \ell_{6,\epsilon},
\end{aligned}
$$

where for (a) we have used the facts that $\frac{SA}{\delta} \geq 1$ and $\log\frac{V_\gamma^\uparrow e}{\epsilon(1-\gamma)} \geq 1$.

Now, we prove $W(T_0) < T_0$. Since $B \geq 1$, therefore $\ell_{5,\epsilon} \geq \log 141 > 1$. This leads to $T_0 \geq 3670SA$, henceforce $\ell_{2,T_0} \geq 1$. Along with $\frac{V_\gamma^\uparrow}{\epsilon} \geq 1$, we have:

$$
\begin{aligned}
W(T_0) &= \frac{1780R_{\max}^2 SA\ell_{1,T_0}\ell_{2,T_0}}{\epsilon^2(1-\gamma)^3} + \frac{240R_{\max}S^2A\ell_{2,T_0}\ell_{3,T_0}}{\epsilon(1-\gamma)^2} + \frac{52R_{\max}SA\ell_{1,T_0}}{\epsilon(1-\gamma)^2} \\
&\leq \frac{1780R_{\max}^2 SA\ell_{1,T_0}\ell_{2,T_0}}{\epsilon^2(1-\gamma)^3} + \frac{240R_{\max}S^2A\ell_{2,T_0}\ell_{3,T_0}}{\epsilon(1-\gamma)^2} + \frac{52R_{\max}^2 SA\ell_{1,T_0}\ell_{2,T_0}}{\epsilon^2(1-\gamma)^3} \\
&= \frac{1832R_{\max}^2 SA\ell_{1,T_0}\ell_{2,T_0}}{\epsilon^2(1-\gamma)^3} + \frac{240R_{\max}S^2A\ell_{2,T_0}\ell_{3,T_0}}{\epsilon(1-\gamma)^2} \\
&:= W'(T_0)
\end{aligned}
$$

Substituting the bounds on logarithmic terms, we obtain:

$$
\begin{aligned}
W(T_0) &\leq W'(T_0) \\
&\leq \frac{3664R_{\max}^2 SA\ell_1\ell_{5,\epsilon}}{\epsilon^2(1-\gamma)^3} + \frac{480R_{\max}S^2A\ell_{5,\epsilon}(2\ell_1 + \ell_{6,\epsilon})}{\epsilon(1-\gamma)^2} \\
&\leq \frac{3666R_{\max}^2 SA\ell_1\ell_{5,\epsilon}}{\epsilon^2(1-\gamma)^3} + \frac{480R_{\max}S^2A\ell_{5,\epsilon}(2\ell_1 + \ell_{6,\epsilon})}{\epsilon(1-\gamma)^2} - 2 \\
&\leq \frac{3670R_{\max}^2 SA\ell_1\ell_{5,\epsilon}}{\epsilon^2(1-\gamma)^3} + \frac{480R_{\max}S^2A\ell_{5,\epsilon}(2\ell_1 + \ell_{6,\epsilon})}{\epsilon(1-\gamma)^2} - 2 \\
&\leq \left\lfloor \frac{3670R_{\max}^2 SA\ell_1\ell_{5,\epsilon}}{\epsilon^2(1-\gamma)^3} + \frac{480R_{\max}S^2A\ell_{5,\epsilon}(2\ell_1 + \ell_{6,\epsilon})}{\epsilon(1-\gamma)^2} \right\rfloor + 1 - 2 \\
&\leq \left\lfloor \frac{3670R_{\max}^2 SA\ell_1\ell_{5,\epsilon}}{\epsilon^2(1-\gamma)^3} + \frac{480R_{\max}S^2A\ell_{5,\epsilon}(2\ell_1 + \ell_{6,\epsilon})}{\epsilon(1-\gamma)^2} \right\rfloor - 1 \\
&= T_0 - 1 \\
&< T_0.
\end{aligned}
$$

$\square$

Now, we formally prove Proposition 3.

*Proof.* From Lemmas 26 and 27, we know that $|\Gamma_T| \leq W(|\Gamma_T|)$ and $W(T_0) < T_0$. It implies that $|\Gamma_T| \neq T_0$ for all $T \in \mathbb{N}$. Since $|\Gamma_T|$ increases by at most 1 starting from $|\Gamma_0| = 0$, that is, $|\Gamma_{T+1}| \leq |\Gamma_T| + 1$ for all $T \in \mathbb{N}$, we conclude that $|\Gamma_T| < T_0$ for all $T \in \mathbb{N}$. Otherwise, there exists $T'$ such that $|\Gamma_{T'}| > T_0$. Assume $T'$ is the minimal such index. Then it follows that $|\Gamma_{T'-1}| = T_0$, which leads to a contradiction. $\square$

# H  POSTERIOR PREDICTIVE AND EPISTEMIC UNCERTAINTY

In this section, we will give backgrounds necessary to relate the Bayes estimator to the MLE estimator.

## H.1  POSTERIOR PREDICTIVE

### H.1.1  TRANSITION

**Lemma 28.** *Let $b_0 := \mathrm{Dir}(\boldsymbol{\alpha})$ be a Dirichlet prior over transition for a fixed $(s, a) \in \mathcal{S} \times \mathcal{A}$, and define $\alpha_0 := \mathbf{1}^\top \boldsymbol{\alpha}$ as the sum of prior parameters. Let $n$ denote the total number of visits to $(s, a)$. Then, the following decomposition holds:*

$$P_b - P = \frac{n}{n + \alpha_0} \left( \hat{P} - P \right) + \frac{\alpha_0}{n + \alpha_0} \left( P_{b_0} - P \right),$$

*for any posterior $b$ and $n \in \mathbb{N}$.*

*Proof.* Note $P_{b_0} = \frac{\boldsymbol{\alpha}}{\alpha_0}$, we get:

$$
\begin{aligned}
P_b - P &= \frac{\mathbf{n} + \boldsymbol{\alpha}}{n + \alpha_0} - P \\
&= \frac{n\hat{P} + \alpha_0 P_{b_0}}{n + \alpha_0} - \left( \frac{n}{n + \alpha_0} + \frac{\alpha_0}{n + \alpha_0} \right) P \\
&= \frac{n}{n + \alpha_0} \left( \hat{P} - P \right) + \frac{\alpha_0}{n + \alpha_0} \left( P_{b_0} - P \right).
\end{aligned}
$$

$\square$

### H.1.2  REWARD

**Lemma 29.** *Let $b_0 := \mathcal{N}(\mu_0, \frac{1}{\tau_0})$ be a Normal prior over mean of reward for a fixed $(s, a) \in \mathcal{S} \times \mathcal{A}$, and $\tau$ the precision of the data distribution, which is assumed to be known. Let $n$ denote the total number of visits to $(s, a)$. Then, the following decomposition holds:*

$$r_b(s, a) - r(s, a) = \frac{\tau_0}{\tau_0 + n\tau} (\mu_0 - r(s, a)) + \frac{n\tau}{\tau_0 + n\tau} (\hat{r}(s, a) - r(s, a)),$$

*for any posterior $b$ and $n \in \mathbb{N}$.*

*Proof.* By definition, we have the posterior predictive mean of the reward:

$$r_b(s, a) = \frac{\tau_0 \mu_0 + \tau \sum_{i=1}^{n} r_i}{\tau_0 + n\tau}.$$

The difference to the ground truth reward is:

$$
\begin{aligned}
r_b(s, a) - r(s, a) &= \frac{\tau_0 \mu_0 + \tau \sum_{i=1}^{n} r_i}{\tau_0 + n\tau} - r(s, a) \\
&= \frac{(\tau_0 \mu_0 + n\tau \hat{r}(s, a)) - (\tau_0 + n\tau) r(s, a)}{\tau_0 + n\tau} \\
&= \frac{\tau_0 (\mu_0 - r(s, a)) + n\tau (\hat{r}(s, a) - r(s, a))}{\tau_0 + n\tau} \\
&= \frac{\tau_0}{\tau_0 + n\tau} (\mu_0 - r(s, a)) + \frac{n\tau}{\tau_0 + n\tau} (\hat{r}(s, a) - r(s, a)).
\end{aligned}
$$

$\square$

**Corollary 4.** *Let* $b_0 := \mathcal{NG}(\mu_0, \lambda_0, \alpha_0, \beta_0)$ *be a Normal-Gamma prior over reward for a fixed* $(s, a) \in \mathcal{S} \times \mathcal{A}$. *Let* $n$ *denote the total number of visits to* $(s, a)$. *Then, the following decomposition holds:*

$$r_b(s, a) - r(s, a) = \frac{\lambda_0}{\lambda_0 + n}(\mu_0 - r(s, a)) + \frac{n}{\lambda_0 + n}(\hat{r}(s, a) - r(s, a)),$$

*for any posterior* $b$ *and* $n \in \mathbb{N}$.

## H.2 EPISTEMIC UNCERTAINTY

The definition of variance-based epistemic uncertainty for both transition and reward is:

$$\mathcal{E}_T(s, a) := \mathrm{Var}_{\mathbf{w} \sim b}\left(\mathbb{E}[s'|s, a, \mathbf{w}]\right)$$
$$\mathcal{E}_R(s, a) := \mathrm{Var}_{\mathbf{w} \sim b}\left(\mathbb{E}[r|s, a, \mathbf{w}]\right)$$

And we consider a generalized form of epistemic uncertainty to combine the two sources together:

$$\mathcal{E}'(s, a) := f(\mathcal{E}_T(s, a), \mathcal{E}_R(s, a)).$$

In this paper, we consider $f(x, y) = \eta(\sqrt{x} + \sqrt{y})$.

### H.2.1 BOUNDS FOR TRANSITION

Since it is meaningless to take expectation over categories for a categorical distribution, we instead choose some feature vector for each component. One of the sensible choices is the basis function $\mathbf{e}_i = (0, 0, \ldots, i, \ldots, 0)$, which leads to the following formulation.

**Definition 10.** The variance-based epistemic uncertainty of Dirichlet-Multinomial model is defined as follows:

$$\mathcal{E}_T(s, a) = \sum_{k=1}^{S} \frac{(\alpha_k + n_k)(\alpha_0 + n - \alpha_k - n_k)}{(\alpha_0 + n)^2(\alpha_0 + n + 1)}.$$

**Lemma 30.** *For Dirichlet prior, the epistemic uncertainty in transition follows that*

$$\mathcal{E}_T(s, a) = \mathcal{O}\left(\frac{1}{n}\right) \quad and \quad \mathcal{E}_T(s, a) = \Omega\left(\frac{1}{n^2}\right),$$

*for any* $(s, a) \in \mathcal{S} \times \mathcal{A}$ *and* $n \in \mathbb{N}$.

*Proof.* Let $T := \alpha_0 + n$, then we have:

$$\mathcal{E}_T(s, a) = \frac{T^2 - \sum\limits_{k=1}^{S}(\alpha_k + n_k)^2}{T^2(T + 1)}.$$

We will derive its upper and lower bound. We start with the upper bound.

Note $\sum\limits_{k=1}^{S}(\alpha_k + n_k)^2 \geq 0$, therefore we have:

$$\begin{aligned}
\mathcal{E}_T(s, a) &\leq \frac{T^2}{T^2(T + 1)} \\
&= \frac{1}{(T + 1)} \\
&= \frac{1}{n + \alpha_0 + 1} \\
&\leq \frac{1}{n}.
\end{aligned}$$

So $\mathcal{E}_T(s, a) = \mathcal{O}(\frac{1}{n})$ with constant $C_2 = 1$

Now, we focus on the lower bound. Consider the worse case, where we have only one state being visited, denote its index as $j$, we have

$$T^2 - \sum_{k=1}^{S}(\alpha_k + n_k)^2 = (\alpha_0 + n)^2 - (n + \alpha_j)^2 + \sum_{k \neq j} \alpha_j^2$$

$$= (n^2 + 2\alpha_0 n + \alpha_0^2) - (n^2 + 2\alpha_j n + \sum_{j=1}^{S} \alpha_j^2)$$

$$= (2\alpha_0 - 2\alpha_j)n + (\alpha_0^2 - \sum_{j=1}^{S} \alpha_j^2)$$

$$\geq (2\alpha_0 - 2\alpha_j)n.$$

Therefore

$$\frac{\mathcal{E}_T(s,a)}{\frac{1}{n^2}} \geq \frac{(2\alpha_0 - 2\alpha_j)n}{\frac{T^2}{n}(T+1)}$$

$$\geq \frac{(2\alpha_0 - 2\alpha_j)}{\frac{T^2}{n}(\frac{T+1}{n})}$$

$$= \frac{(2\alpha_0 - 2\alpha_j)}{(1 + \frac{\alpha_0}{n})^2(1 + \frac{\alpha_0+1}{n})}$$

$$\geq \frac{(2\alpha_0 - 2\alpha_j)}{(1 + \alpha_0)^2(2 + \alpha_0)}.$$

So $\mathcal{E}_T(s,a) = \Omega(\frac{1}{n^2})$ with constant $C_1 = \frac{(2\alpha_0 - 2\alpha_j)}{(1+\alpha_0)^2(2+\alpha_0)}$. This corresponds to the case where the transition is deterministic or near-deterministic. □

### H.2.2 BOUNDS FOR REWARD

**Definition 11** (Normal-Normal). The variance-based epistemic uncertainty of Normal-Normal model is defined as follows:

$$\mathcal{E}_R(s,a) = \frac{1}{\tau_0 + \tau n}.$$

**Definition 12** (Normal-Gamma). The variance-based epistemic uncertainty of Normal-Gamma model is defined as follows:

$$\mathcal{E}_R(s,a) = \frac{\beta}{\lambda(\alpha - 1)},$$

where

$$\lambda = \lambda_0 + n$$
$$\alpha = \alpha_0 + \frac{n}{2}$$
$$\beta = \beta_0 + \frac{1}{2}\left(n\hat{\sigma}^2 + \frac{\lambda_0 n(\bar{x} - \mu_0)^2}{\lambda_0 + n}\right).$$

**Lemma 31.** *For Normal prior, the epistemic uncertainty in reward follows that*

$$\mathcal{E}_R(s,a) = \Theta\left(\frac{1}{n}\right)$$

*for any $(s,a) \in \mathcal{S} \times \mathcal{A}$ and $n \in \mathbb{N}$.*

*Proof.* Note by choosing $C_1 = \frac{1}{\tau_0 + \tau}$ for lower bound and $C_2 = \frac{1}{\tau}$ for upper bound concludes. □

**Lemma 32.** *For Normal-Gamma prior, the epistemic uncertainty in reward follows that*

$$\mathcal{E}_R(s,a) = \mathcal{O}\left(\frac{1}{n}\right) \quad and \quad \mathcal{E}_T(s,a) = \Omega\left(\frac{1}{n^2}\right),$$

*for any $(s,a) \in \mathcal{S} \times \mathcal{A}$ and $n \in \mathbb{N}$.*

*Proof.* The upper bound is trivial. For the lower bound, consider the deterministic case, leading to sample variance being zero. Therefore the numerator is $\Theta(1)$ whereas the denominator $\mathcal{O}(n^2)$. $\square$

## I   FROM FREQUENTIST TO BAYESIAN

### I.1   PROPERTIES OF PRIORS

**Definition 13** (Decomposable). A prior $b_0$ parameterized by $\boldsymbol{\theta}$ is said to be *decomposable* if there exist functions $f(n, \boldsymbol{\theta})$, $g(n, \boldsymbol{\theta})$ for transitions and $h(n, \boldsymbol{\theta})$, $s(n, \boldsymbol{\theta})$ for rewards such that

$$P_b - P = f(n, \boldsymbol{\theta})(\hat{P} - P) + g(n, \boldsymbol{\theta})(P_{b_0} - P),$$
$$r_b - r = h(n, \boldsymbol{\theta})(\hat{r} - r) + s(n, \boldsymbol{\theta})(r_{b_0} - r),$$

with the constraints

$$f(n, \boldsymbol{\theta}) \leq 1, \quad h(n, \boldsymbol{\theta}) \leq 1 \quad \forall n \in \mathbb{N}; \quad g(n, \boldsymbol{\theta}) = \mathcal{O}\left(\frac{S}{n}\right), \quad s(n, \boldsymbol{\theta}) = \mathcal{O}\left(\frac{1}{n}\right),$$

for some positive multiplicative constant $C_g(\boldsymbol{\theta})$ and $C_s(\boldsymbol{\theta})$.

Note, when indexed by a particular $(s,a)$, all the quantities above can depend on it.

**Definition 14** (Weakly Informative). A prior $b_0$ parameterized by $\boldsymbol{\theta}$ is said to be *weakly informative* if

$$|r_b - \hat{r}| = \mathcal{O}\left(\frac{1}{n}\right) \quad and \quad \|P_b - \hat{P}\|_1 = \mathcal{O}\left(\frac{S}{n}\right).$$

**Definition 15** (Uniform). A prior $b_0$ parameterized by $\boldsymbol{\theta}$ is said to be *uniform* if there exist positive constants $C_g$ and $C_s$ such that

$$C_g(\boldsymbol{\theta})(s,a) \leq C_g \quad and \quad C_s(\boldsymbol{\theta})(s,a) \leq C_s$$

for any $(s,a) \in \mathcal{S} \times \mathcal{A}$.

**Definition 16** (Bounded). A prior $b_0$ parameterized by $\boldsymbol{\theta}$ is said to be *bounded* if there exists $\bar{R} \geq 0$ such that $|r_{b_0}(s,a)| \leq \bar{R}$ for any $(s,a) \in \mathcal{S} \times \mathcal{A}$.

**Definition 17** (Restatement of Definition 1). Let $\mathfrak{C}$ be defined by the class of *decomposable* or *weakly informative* priors whose rate of epistemic uncertainty is $\Theta\left(\frac{1}{\sqrt{n}}\right)$.

**Theorem 10** (Restatement of Theorem 4). *Let $\mathcal{M} = (\mathcal{S}, \mathcal{A}, P, r, \gamma)$ be any MDP. For any prior $b_0 \in \mathfrak{C}$, there exists an instance of* EUBRL *such that, when executed on $\mathcal{M}$, it achieves, with probability at least $1 - \delta$, a prior-dependent bound on regret, or alternatively, on sample complexity, depending on the choice of $\eta$. If, furthermore, $b$ is assumed to be uniform and bounded, these bounds are nearly minimax-optimal.*

*Proof.* Note, by either weak informativeness or decomposability, the additional complexity is at most $\mathcal{O}\left(\frac{S}{n}\right)$ for transitions and $\mathcal{O}\left(\frac{1}{n}\right)$ for reward. This applies to the events, e.g. $\mathbf{A}_{1:4}^{\gamma}$, which involve bounding the distance between the posterior predictive and the ground truth. Without loss of generality, we assume $b$ is weakly informative. We bound $\left|(\hat{P}^t - P)V^\star(s,a)\right|$ as follows:

$$|(P_{b_t} - P)V^\star(s,a)| = \left| f(N^t(s,a), \boldsymbol{\theta})(\hat{P}^t - P)V^\star(s,a) + g(N^t(s,a), \boldsymbol{\theta})\left(P_{b_0} - P\right)V^\star(s,a)\right|$$

$$\leq \underbrace{\left|(\hat{P}^t - P)V^\star(s,a)\right|}_{\text{Frequentist Bound}} + \underbrace{\left(C_g(\boldsymbol{\theta})(s,a)\|P_{b_0} - P\|_1 V_\gamma^\uparrow\right)\frac{S}{N^t(s,a)}}_{\text{Prior Bias}},$$

where the first term is simply the original bound derived in the analysis of MLE estimators, while the second term captures the complexity arising from prior misspecification. If the prior is correctly specified, there is no additional overhead; otherwise, this term must be accounted for in the final bound.

Similarly, we have the decomposition for the reward

$$\left|r_{b_t}(s,a) - r(s,a)\right| \leq \left|\hat{r}^t(s,a) - r(s,a)\right| + \left(C_s(\boldsymbol{\theta})(s,a) \left|r_{b_0}(s,a) - r(s,a)\right|\right) \frac{1}{N^t(s,a)}.$$

By merging all the quantities of the same order of $\frac{1}{N^t}$, we can overload the definition of $\Upsilon^t$, $\mathcal{Y}^t$, and $\beta^t$, respectively. For brevity, we drop the dependency on $(s,a)$ for each term.

$$\texttt{Quasi-optimism} \quad \Upsilon^t \leftarrow \Upsilon^t + \left(C_g(\boldsymbol{\theta}) \left\|P_{b_0} - P\right\|_1 V_\gamma^\uparrow\right) \frac{S}{N^t} + \left(C_s(\boldsymbol{\theta}) \left|r_{b_0} - r\right|\right) \frac{1}{N^t}$$

$$\texttt{Accuracy} \quad \beta_1^t \leftarrow \beta_1^t + 2 \left(C_g(\boldsymbol{\theta}) \left\|P_{b_0} - P\right\|_1 V_\gamma^\uparrow\right) \frac{S}{N^t}$$

$$\beta^t \leftarrow P_U^t \eta^t \mathcal{E}^t + \beta_1^t + (1 - P_U^t(s)) \frac{V_\gamma^\uparrow \ell_1}{\lambda_t N^t} + (1 - P_U^t(s)) \left(C_s(\boldsymbol{\theta}) \left|r_{b_0} - r\right|\right) \frac{1}{N^t}$$

$$\texttt{Bounding } J_\gamma^t(s_t) \quad \mathcal{Y}^t \leftarrow \frac{12 V_\gamma^\uparrow \ell_1}{\lambda_t} + 30 V_\gamma^\uparrow S \ell_{3,t} + 3 \left(C_g(\boldsymbol{\theta}) \left\|P_{b_0} - P\right\|_1 V_\gamma^\uparrow S\right) + 2 \left(C_s(\boldsymbol{\theta}) \left|r_{b_0} - r\right|\right).$$

In addition, since the rate of the epistemic uncertainty is $\Theta\left(\frac{1}{\sqrt{N^t}}\right)$, a scaling factor $\eta$ can be chosen appropriately such that $P_U^t(s,a) \eta^t \mathcal{E}^t(s,a) - P_U^t(s,a) R_{\max} \geq \frac{\Upsilon^t}{N^t(s,a)}$, akin to that of the proof of Lemma 33, with which we are guaranteed the quasi-optimism to hold.

Since

$$\left\|P_{b_0}(\cdot|s,a) - P(\cdot|s,a)\right\|_1 \leq 2$$
$$\left|r_{b_0}(s,a) - r(s,a)\right| \leq \left|r_{b_0}(s,a)\right| + R_{\max},$$

we denote

$$\Lambda_T(\boldsymbol{\theta}) \coloneqq \max_{(s,a)\in\mathcal{S}\times\mathcal{A}} \left\{C_g(\boldsymbol{\theta})(s,a)\right\}$$

$$\Lambda_R(\boldsymbol{\theta}) \coloneqq \max_{(s,a)\in\mathcal{S}\times\mathcal{A}} \left\{C_s(\boldsymbol{\theta})(s,a) \left(\left|r_{b_0}(s,a)\right| + R_{\max}\right)\right\}.$$

Following the same procedure for analyzing regret and sample complexity, we obtain prior-dependent bounds as follows:

$$\textbf{Regret} \quad \widetilde{\mathcal{O}}\left(\frac{\sqrt{SAT}}{(1-\gamma)^{1.5}} + (1 + \Lambda_T(\boldsymbol{\theta})) \frac{S^2 A}{(1-\gamma)^2} + \Lambda_R(\boldsymbol{\theta}) \frac{SA}{1-\gamma}\right)$$

$$\textbf{Sample Complexity} \quad \widetilde{\mathcal{O}}\left(\left(\frac{SA}{\epsilon^2(1-\gamma)^3} + (1 + \Lambda_T(\boldsymbol{\theta}) + \Lambda_R(\boldsymbol{\theta})) \frac{S^2 A}{\epsilon(1-\gamma)^2}\right) \log\frac{1}{\delta}\right).$$

If the prior $b_0$ is furthermore assumed to be uniform and bounded, both $\Lambda_T(\boldsymbol{\theta})$ and $\Lambda_R(\boldsymbol{\theta})$ will reduce to constants that do not depend on the state-action pairs, thus leading to a bound similar to that in the frequentist case. $\qquad\square$

*Remark* 2. Since the epistemic uncertainty is additive across both reward and transition sources, it suffices for either source to satisfy an order of $\Theta\left(\frac{1}{\sqrt{n}}\right)$. The other source may decay faster.

In the following sections, we will instantiate specific priors.

## I.2   DIRICHLET AND NORMAL PRIORS

**Corollary 5** (Restatement of Corollary 1)**.** *Let $b_0$ denote the joint distribution consisting of a Dirichlet prior $\mathrm{Dir}(\alpha\mathbf{1}_{S\times 1})$ on the transition probability vector and a Normal prior $\mathcal{N}(\mu_0, \frac{1}{\tau_0})$ on the mean reward with known precision $\tau$ for all $(s,a) \in \mathcal{S} \times \mathcal{A}$. Then $b_0 \in \mathfrak{C}$ and is uniform and bounded, and hence achieves nearly minimax-optimality when used with* `EUBRL`*.*

*Proof.* By Lemma 31, we know that $\mathcal{E}'_R(s,a) = \Theta\left(\frac{1}{\sqrt{n}}\right)$. By Lemma 30, we know that $\mathcal{E}'_T(s,a) = \mathcal{O}\left(\frac{1}{\sqrt{n}}\right)$ and $\mathcal{E}'_T(s,a) = \Omega\left(\frac{1}{n}\right)$. By Remark 2, this makes the final epistemic uncertainty $\mathcal{E}'(s,a) = \Theta\left(\frac{1}{\sqrt{n}}\right)$. In addition, Lemmas 28 and 29 imply that the prior is *decomposable*. All together, we have $b_0 \in \mathfrak{C}$.

In addition, we can find $C_g = \alpha$ and $C_s = \frac{\tau_0}{\tau}$ as required by the uniformality in Definition 15. And note that $|r_{b_0}(s,a)| = |\mu_0|, \forall(s,a) \in \mathcal{S} \times \mathcal{A}$, therefore the boundedness in Definition 16 is satisfied as well. $\qquad\square$

### I.3   DIRICHLET AND NORMAL-GAMMA PRIORS

**Proposition 4** (Restatement of Proposition 1). *For a Normal-Gamma prior $\mathcal{NG}(\mu_0, \lambda_0, \alpha_0, \beta_0)$, there exists a parameterization and an MDP such that $\exists t \in \mathbb{N}$ for which quasi-optimism does not hold.*

This follows from the fact that the epistemic uncertainty under a Normal-Gamma prior depends on the sample variance, which multiplies the number of visits $n$ in the numerator (Definition 12). In deterministic or nearly deterministic MDPs, the sample variance can be zero, yielding a lower bound on the epistemic uncertainty:

$$\mathcal{E}_R(s,a) = \Omega\left(\frac{1}{n^2}\right),$$

which is insufficient to guarantee quasi-optimism, especially when a prior bias presents. Even the frequentist bound may vanish.

## J   HELPER LEMMAS

**Lemma 33.** *It holds that*

$$b^k(s,a) - P_U^k(s,a)R_{max} \geq \frac{\Upsilon^k}{N^k(s,a)},$$

*for any $(s,a) \in \mathcal{S} \times \mathcal{A}$ and $k \in \mathbb{N}$.*

*Proof.* For $N^k(s,a) \geq m$, the inequality trivially holds. For $N^k(s,a) < m$, note by choosing $\eta^k = \mathcal{E}_{max}\Upsilon^k + R_{max}\sqrt{m_k}$, we have:

$$
\begin{aligned}
(b^k(s,a) - P_U^k(s,a)R_{max}) - \frac{\Upsilon^k}{N^k(s,a)} &= \left(P_U^k(s,a)\eta^k\mathcal{E}^k(s,a) - P_U^k(s,a)R_{max}\right) - \frac{\Upsilon^k}{N^k(s,a)} \\
&= \left(\frac{\eta^k}{\mathcal{E}_{max}}\frac{1}{N^k(s,a)} - \frac{R_{max}}{\mathcal{E}_{max}}\frac{1}{\sqrt{N^k(s,a)}}\right) - \frac{\Upsilon^k}{N^k(s,a)} \\
&= \left(\left(\Upsilon^k + \frac{R_{max}}{\mathcal{E}_{max}}\sqrt{m_k}\right)\frac{1}{N^k(s,a)} - \frac{R_{max}}{\mathcal{E}_{max}}\frac{1}{\sqrt{N^k(s,a)}}\right) - \frac{\Upsilon^k}{N^k(s,a)} \\
&= \frac{R_{max}}{\mathcal{E}_{max}}\left(\frac{\sqrt{m_k}}{N^k(s,a)} - \frac{1}{\sqrt{N^k(s,a)}}\right) \\
&= \frac{R_{max}}{\mathcal{E}_{max}}\left(\frac{\sqrt{m_k} - \sqrt{N^k(s,a)}}{N^k(s,a)}\right) \\
&> 0,
\end{aligned}
$$

which is as desired. $\qquad\square$

The following Lemma is helpful in proving both the quasi-optimism and accuracy for discounted settings.

**Lemma 34.** *Let $C \geq 0$ be a constant and $\gamma \in (0,1)$. Let $V$ be a function such that $V : \mathcal{S} \to [0, C]$. For any $(s, a) \in \mathcal{S} \times \mathcal{A}$, the variance of $V$ under $P(\cdot|s,a)$ is bounded as follows:*

$$\gamma \mathrm{Var}(V)(s,a) \leq -\Delta_\gamma(V^2)(s,a) + (1+\gamma)C\max\{\Delta_\gamma(V)(s,a), 0\}.$$

*Equivalently, the following inequality holds:*

$$\gamma \mathrm{Var}(V)(s,a) - \gamma P(V)^2(s,a) \leq -(V(s))^2 + (1+\gamma)C\max\{\Delta_\gamma(V)(s,a), 0\}.$$

*Proof.* Adding and subtracting $(V(s))^2$ to $\gamma\mathrm{Var}(V)(s,a)$, we get

$$\begin{aligned}
\gamma\mathrm{Var}(V)(s,a) &= \gamma P(V)^2(s,a) - \gamma(PV(s,a))^2 \\
&= \gamma P(V)^2(s,a) - (V(s))^2 + (V(s))^2 - \gamma(PV(s,a))^2 \\
&\overset{(\mathbf{a})}{\leq} \gamma P(V)^2(s,a) - (V(s))^2 + (V(s))^2 - \gamma^2(PV(s,a))^2 \\
&= \gamma P(V)^2(s,a) - (V(s))^2 + (V(s) + \gamma PV(s,a))(V(s) - \gamma PV(s,a)) \\
&\overset{(\mathbf{b})}{\leq} \gamma P(V)^2(s,a) - (V(s))^2 + (1+\gamma)C(V(s) - \gamma PV(s,a)) \\
&= -\Delta_\gamma(V^2)(s,a) + (1+\gamma)C(V(s) - \gamma PV(s,a)) \\
&\leq -\Delta_\gamma(V^2)(s,a) + (1+\gamma)C\max\{\Delta_\gamma(V)(s,a), 0\},
\end{aligned}$$

where $(\mathbf{a})$ is due to the fact that $\gamma > \gamma^2$ and $(\mathbf{b})$ by the boundedness of value functions. $\qquad\square$

**Lemma 35.** *Let $V^k$ denote the value function of the approximate MDP under its derived policy $\pi_k$. Let $V^{\pi_k}$ denote the value function of the true MDP under the same policy. Then the difference between $V^k$ and $V^{\pi_k}$ is bounded as follows:*

$$\Delta_h\left(V^k - V^{\pi_k}\right)(s,a) \leq \left(\Delta_h(D^k)(s,a) + 2\beta^k(s,a)\right).$$

*Proof.* The proof is completed by applying the procedure of Lemma 13 in (Lee & Oh, 2025), except using $\widehat{V}_h^k(s) + S_k \geq 0$ from Lemmas 2–3 for variance decomposition, together with an adjustment of some constants. $\qquad\square$

## K  PRIOR MISSPECIFICATION

**Problem Setting**  Given a two-armed bandit:

$$a_1 : P(r|a_1) = \mathbf{Bern}(\mu_1) \tag{11}$$
$$a_2 : P(r|a_2) = \mathbf{Bern}(\mu_2) \tag{12}$$
$$\text{with } \mu_1 > \mu_2 \tag{13}$$

We use Beta distribution to model the belief over the parameter of the underlying Bernoulli distribution. We have independent prior $b(\mathbf{w}|a_i) = \mathbf{Beta}(\alpha_i, \beta_i)$ over each arm with parameters $\alpha_i > 0, \beta_i > 0, i \in \{1, 2\}$. Since Beta distribution is the conjugate prior of the Bernoulli distribution, after observing the number of success $S_i$ and failures $F_i$, we can get the posterior in a closed-from, i.e.

$$b(\mathbf{w}|a_i, S_i, F_i) = \mathbf{Beta}(\alpha_i + S_i, \beta_i + F_i) \tag{14}$$
$$= \mathbf{Beta}(\alpha_i', \beta_i'), i \in \{1, 2\}. \tag{15}$$

Then the EUBRL reward will be:

$$r_i^{\mathrm{EUBRL}} = (1 - P_U)\,\hat{r}_i + P_U\,\mathcal{E}_i, \text{ where} \tag{16}$$
$$\hat{r}_i = \mathbb{E}_{b(\mathbf{w}|a_i, S_i, F_i), P(r|a_i, \mathbf{w})}[r] \tag{17}$$
$$= \frac{\alpha_i + S_i}{(\alpha_i + S_i) + (\beta_i + F_i)} \tag{18}$$
$$= \frac{\alpha_i'}{\alpha_i' + \beta_i'} \tag{19}$$

The epistemic uncertainty can also be expressed in a closed form:

$$\mathcal{E}(a_i) = \text{Var}_{b(\mathbf{w}|a_i, S_i, F_i)} \left[ \mathbb{E}_{P(r|a_i, \mathbf{w})} [r] \right] \tag{20}$$

$$= \text{Var}_{b(\mathbf{w}|a_i, S_i, F_i)} [\mathbf{w}] \tag{21}$$

$$= \frac{\alpha_i' \beta_i'}{(\alpha_i' + \beta_i')^2 (\alpha_i' + \beta_i' + 1)} \tag{22}$$

If we assume that the parameters of the prior are equal, we can show that epistemic uncertainty is non-increasing. This result is formalized in the following lemma:

**Lemma 36.** *Given a Beta prior distribution* $\text{Beta}(\alpha, \beta)$ *with* $\alpha = \beta > 0$ *for the parameter of a Bernoulli distribution, the variance of the posterior distribution decreases monotonically with the number of observations.*

*Proof.* Let denote the $b_0$ as the Beta prior before observing any outcome from the Bernoulli distribution. It has a variance $\text{Var}(b_0) = \frac{1}{4(2\alpha+1)}$. After observing one sample from the Bernoulli distribution, whether it is success or failure, we will have an updated posterior $b_1$ with the variance:

$$\text{Var}(b_1) = \frac{\alpha}{2(2\alpha + 1)^2} \tag{23}$$

By examining the difference between the two, we have $\text{Var}(b_0) - \text{Var}(b_1) = \frac{1}{4(2\alpha+1)^2} > 0$. Therefore, the variance of the posterior is decreasing after observing one outcome. However, since this result will hold for the next posterior compared to the current posterior as well, we can conclude that the variance of the posterior is monotonically decreasing. $\qquad\square$

We will prove the following theorem:

**Theorem 11** (Prior Misspecification). *Let* $\eta = 1$*. There exists an MDP* $\mathcal{M}$*, a prior* $b_0$*, an accuracy level* $\epsilon_0 > 0$*, and a confidence level* $\delta_0 \in (0, 1]$ *such that, with probability greater than* $1 - \delta_0$*,*

$$V^{\pi_t}(s_t) < V^{\star}(s_t) - \epsilon_0 \tag{24}$$

*will hold for an unbounded number of time steps.*

*Proof.* Before any new observation, both $r_i^{\text{EUBRL}} = \mathcal{E}_{\max}$, therefore breaking the tie leads to a half probability to choose either arm. Consider choosing the second arm, it will lead to some reduction of the epistemic uncertainty because of the new observation.

We aim to force the agent to repeatedly select this arm, thereby preventing it from ever reaching the optimal one. To achieve this, we need to ensure that (to simplify notation, we will henceforth drop the dependency of the epistemic uncertainty on the action; $\mathcal{E}$ will refer to the epistemic uncertainty of the second arm whenever it is considered):

$$r_2^{\text{EUBRL}} - r_1^{\text{EUBRL}} = ((1 - P_U)\hat{r} + P_U \mathcal{E}) - \mathcal{E}_{\max} \tag{25}$$

$$= ((1 - P_U)\hat{r} + P_U \mathcal{E}) - ((1 - P_U)\mathcal{E}_{\max} + P_U \mathcal{E}_{\max}) \tag{26}$$

$$= (1 - P_U)(\hat{r} - \mathcal{E}_{\max}) + P_U (\mathcal{E} - \mathcal{E}_{\max}) \tag{27}$$

$$\geq 0. \tag{28}$$

Note, the second term in the penultimate line is a quadratic function; therefore, we can obtain its minimum as follows:

$$\min_{\mathcal{E}} \frac{1}{\mathcal{E}_{\max}} \left( \mathcal{E}^2 - \mathcal{E}_{\max} \mathcal{E} \right) \tag{29}$$

$$= -\frac{\mathcal{E}_{\max}}{4} \tag{30}$$

Therefore, as long as we ensure that Eq. 27 with substitution of this lower bound is non-negative, we can guarantee Eq. 28 to hold. That being said, we require the following condition to be satisfied:

$$(1 - P_U)(\hat{r} - \mathcal{E}_{\max}) - \frac{\mathcal{E}_{\max}}{4} \geq 0, \tag{31}$$

which is equivalent to:

$$\hat{r} \geq \frac{\mathcal{E}_{\max}}{4(1 - P_U)} + \mathcal{E}_{\max}. \tag{32}$$

By Lemma 36, we know that $P_U$ is decreasing. Therefore, it suffices to ensure that:

$$\hat{r} \geq \frac{\mathcal{E}_{\max}}{4(1 - P_{U,1})} + \mathcal{E}_{\max}, \tag{33}$$

where $P_{U,1}$ denotes the probability of uncertainty after observing the first outcome from the second arm.

Moreover, the right-hand side can be expressed as:

$$s(a) := \frac{\mathcal{E}_{\max}}{4(1 - P_{U,1})} + \mathcal{E}_{\max} \tag{34}$$

$$= \frac{1}{16} + \frac{1}{4(2\alpha + 1)}. \tag{35}$$

Since $\alpha \in (0, \infty)$, we can bound $s(a)$ within the interval $\left(\frac{1}{16}, \frac{5}{16}\right)$, which will be useful in our later analysis.

We now aim to show that, under certain priors, the probability of the agent sticking to the second arm is high. In other words, it suffices to show that the probability of not pulling the second arm is small. To that end, let us focus on the event $\hat{r} < s(a)$.

To proceed, we consider the following decomposition of the reward estimate:

$$\hat{r} = \frac{\alpha + S_n}{2\alpha + n} \tag{36}$$

$$= \frac{n}{2\alpha + n}\bar{r} + \frac{\alpha}{2\alpha + n}, \tag{37}$$

where $n$ is the total number of occurrences of the outcome from the second arm, and $S_n$ is the total number of successes among these $n$ occurrences.

Notably, we can factor out the empirical mean $\bar{r}$, resulting in a new inequality:

$$\bar{r} < \frac{s(a) - \frac{\alpha}{2\alpha + n}}{\frac{n}{2\alpha + n}} \tag{38}$$

$$= \frac{a(2s(a) - 1)}{n} + s \tag{39}$$

$$:= g(a, n) \tag{40}$$

Next, we apply Hoeffding's inequality to the expression above:

$$P(\bar{r} < g(a, n)) = P(\mu_2 - \bar{r} > \mu_2 - g(a, n)) \tag{41}$$

$$\leq \exp\left(-2n(\mu_2 - s(a))^2\right). \tag{42}$$

This provides an upper bound on the probability of not pulling the second arm over $n$ samples. By applying the union bound at each step, we can bound the probability that the second arm is not pulled at least once, and refer to this event as "Omission":

$$P(\text{Omission}) = P\left(\cup_{n=1}^{\infty}\left(\bar{r} < g(a, n)\right)\right) \tag{43}$$

$$\leq \sum_{n=1}^{\infty} P(\bar{r} < g(a, n)) \tag{44}$$

$$= \underbrace{\sum_{n=1}^{\lfloor a \rfloor} P(\bar{r} < g(a, n))}_{S_1} + \underbrace{\sum_{n=(\lfloor a \rfloor + 1)}^{\infty} P(\bar{r} < g(a, n))}_{S_2}, \tag{45}$$

where we split the sum into two parts based on the floor of $a$, which we will analyze individually.

**Bounding $S_2$**    We denote $k = \frac{\lfloor a \rfloor}{n}$. Since $n > \lfloor a \rfloor$, we know that $k \in [0, 1)$. Therefore, we can rewrite $g(a, n)$ as:

$$g(a, n) = k(2s - 1) + s, k \in [0, 1). \tag{46}$$

For every fixed $n$, we want to find both the lower and upper bound of $g(a, n)$. Since we know $s \in \left(\frac{1}{16}, \frac{5}{16}\right)$ and $g(a, n)$ is linear in $s$, we can solve for the range of $g(a, n)$ as $A_n = (\frac{1}{16} - \frac{7}{8}k, \frac{5}{16} - \frac{3}{8}k)$. In addition, since $k \in [0, 1)$, we can solve for a superset $A = (-\frac{13}{16}, \frac{5}{16})$ that contains every set $A_n, \forall n > \lfloor a \rfloor$. We then analyze the squared term $(\mu_2 - g(a, n))^2$. This is a quadratic function with axis of symmetry of $\mu_2$. There are two possible cases for the relationship between $\mu_2$ and $A$: either $\mu_2 \le \frac{5}{16}$ or $\mu_2 > \frac{5}{16}$. For the first case, the minimum of the quadratic function will be zero, which cancels out the effect of $n$ and results in the largest probability–an outcome we want to avoid. Therefore, we consider the second case, $\mu_2 > \frac{5}{16}$, where the minimum of the quadratic function occurs at $g = \frac{5}{16}$. We denote this minimum as $C := (\mu_2 - \frac{5}{16})^2$. Then we can bound the second term in the probability of omission as follows:

$$S_2 = \sum_{n=(\lfloor a \rfloor + 1)}^{\infty} P(\bar{r} < g(a, n)) \tag{47}$$

$$\le \sum_{n=(\lfloor a \rfloor + 1)}^{\infty} \exp(-2Cn) \tag{48}$$

$$= \exp(-2C\lfloor a \rfloor) \sum_{n=1}^{\infty} \exp(-2Cn) \tag{49}$$

$$= \exp(-2C\lfloor a \rfloor) \frac{\exp(-2C)}{1 - \exp(-2C)} \tag{50}$$

$$= \frac{\exp(-2C(\lfloor a \rfloor + 1))}{1 - \exp(-2C)} \tag{51}$$

$$\le \frac{\eta}{2}, \tag{52}$$

where $\eta \in (0, 1)$ is arbitrary confidence level.

We solve for the above and obtain $\lfloor a \rfloor \ge \frac{1}{2C} \log(\frac{2}{\eta(1 - \exp^{-2C})}) - 1 := a_1$. Next, we will bound the other term.

**Bounding $S_1$**    The goal is to isolate the parameter $a$ and make it dominant. We expand the exponent as:

$$2n(\mu_2 - g(a, n))^2 = 2 \left( \underbrace{n(\mu_2 - s)^2}_{I_1} + \underbrace{2((\mu_2 - s)(1 - 2s))a}_{I_2} + \underbrace{\frac{(2s-1)^2}{n}a^2}_{I_3} \right). \tag{53}$$

Since $\mu_2 > \frac{5}{16}$ and $s \in \left(\frac{1}{16}, \frac{5}{16}\right)$, therefore $I_2 > 0$. And the remaining two terms are also positive. Based on this observation, we provide a lower bound for the exponent as follows:

$$2n(\mu_2 - g(a, n))^2 \ge 2I_3 \ge \frac{\frac{9}{32}}{n}a^2. \tag{54}$$

Next, we use this result to bound $S_1$:

$$S_1 \le \sum_{n=1}^{\lfloor a \rfloor} \exp\left( -\frac{\frac{9}{32}}{n}\lfloor a \rfloor^2 \right) \tag{55}$$

$$\le \sum_{n=1}^{\lfloor a \rfloor} \exp\left( -\frac{9}{32}\lfloor a \rfloor \right) \tag{56}$$

$$= \lfloor a \rfloor \exp\left( -\frac{9}{32}\lfloor a \rfloor \right) \tag{57}$$

$$\le \frac{\eta}{2}, \tag{58}$$

which unfortunately has no closed-form solution. However, we can leverage the Lambart W function to obtain an analytical solution. Denote $u = -\frac{9}{32}\lfloor a \rfloor$, then Eq. 57 can be rewritten as $-\frac{32}{9}u\exp(u)$. We instead bound it as follows:

$$-\frac{32}{9}u\exp(u) \leq \frac{\eta}{2} \tag{59}$$

$$\Leftrightarrow u\exp(u) \geq -\frac{9}{64}\eta, \tag{60}$$

which matches to the Lambart W function. Since there are two branches $W_0(x)$ and $W_{-1}(x)$ of the Lambart W function when $x \in [-\frac{1}{e}, 0)$, and $W_{-1}(x) < W_0(x) < 0$. We can get $u \leq W_{-1}(-\frac{9}{64}\eta)$, therefore $\lfloor a \rfloor \geq -\frac{32}{9}W_{-1}(-\frac{9}{64}\eta) := a_2$.

Combining the two bounds together, as long as we choose $\lfloor a \rfloor > \max\{a_1, a_2\}$, the probability of omission will be bounded as follows:

$$P(\texttt{Omission}) \leq S_1 + S_2 \leq \frac{\eta}{2} + \frac{\eta}{2} = \eta. \tag{61}$$

Therefore, if we denote the event of sticking to the second arm as $\texttt{Sticky}$, its probability will be:

$$P(\texttt{Sticky}) = P\left(\cap_{n=1}^{\infty}\overline{(\bar{r} < g(a,n))}\right) \tag{62}$$

$$= 1 - P\left(\cup_{n=1}^{\infty}(\bar{r} < g(a,n))\right) \tag{63}$$

$$= 1 - P(\texttt{Omission}) \tag{64}$$

$$> 1 - \eta, \tag{65}$$

Therefore, we can conclude that with probability greater than $\delta_0 = \frac{1}{2} \cdot (1 - \eta)$, the second arm will be always pulled, leading to suboptimality. More formally, for any $\epsilon_0 < \mu_1 - \mu_2$, we have:

$$V^{\pi_t}(s_t) < V^{\star}(s_t) - \epsilon_0, \tag{66}$$

where $V^{\pi_t}(s_t) = u_2$ and $V^{\star}(s_t) = \mu_1$, which completes our proof. $\qquad\square$

