# OpenReview forum: "EUBRL: Epistemic Uncertainty Directed Bayesian Reinforcement Learning"
_ICLR.cc/2026/Conference — ICLR 2026 Poster_

### Official Review · Reviewer_kthm · 2025-10-20

**Soundness:** 3
**Presentation:** 1
**Contribution:** 3
**Rating:** 6
**Confidence:** 3

**Summary:**

The paper proposes a method for incorporating epistemic uncertainty into a standard Bayes-Adaptive-MDP-solving framework (interact, update belief-posterior, find the optimal policy with respect to the posterior, repeat). The paper includes a comprehensive theoretical analysis of the new method from the perspective of optimality.

**Strengths:**

Strong theoretical foundation.

Strong theoretical results.

Good sanity-check experiments.

Clear description of theoretical limitations.

**Weaknesses:**

Missing theoretical motivation:
  1. Why is CAI chosen to formulate the epistemic uncertainty?
  2. Why is epistemic uncertainty chosen to be formulated in CAI in this way?

Missing additional limitations - the method is most likely limited to tabular settings. Is that correct?

Presentation: multiple comments, described in more detail below.

**Questions:**

**Presentation:**
1. Algorithm 1 is very simple and very compact, which is excellent. However:
    1. I think it is important it is at least described in the main paper.
    2. ValueIteration(b) is never defined, described, or cited, I believe.
    3. BeliefUpdate(s,r) should also be described in more detail (at least, how the update is done from a practical perspective).
    4. It seems to me that algorithm 1 is not one of the contributions of the paper, but rather a rather-standard approach for posterior-belief/over-models based methods. Is that correct? If yes, I would rephrase the contributions. Rather than introducing a new algorith, the authors introduce a new method to incorporate uncertainty into the belief / posterior. I do not view this as weakening the contribution of the paper - merely making it easier for the reader to understand the novelty of the paper.
2. I don't think $\Epsilon_b$ is ever defined?
2. The "related work section" would work much better after preliminaries (Section 2) than where it currently is, in the end of the paper. It will also make introducing the baselines and understanding the approach much easier in my opinion.
3. I would cite at least one uncertainty survey for the Uncertainty Quantification paragraph, such as [1].
4. Section 3.1 should build on citations from the uncertaity quantification paragraphs in the related work (/preliminaries). Unless 3.1 is entirely novel (in which case - this should be emphasized stronger), the prior work it builds on should be cited.
5. I think lazy chain is incompletely explained. What is the complete action space? Are transitions det. or stoch.? Is there a difference between the left end and the right end? What does solving the problem constitute (ie what is the optimal policy), if the comulative return is zero?
6. I would also suggest adding more detailed descriptions of chain and deep sea.
7. Some \citep should be \cite (line 464).

**Experiments and baselines:**
1. I believe PSRL can be tuned to work much better if the "right" choice of prior over transition models (/rewards) is made (ie - transitions from any states are to at most two different states, rewards are det / from a normal). Since experiments are already run with EUBRL+ and there's a prior selection discussion, I think these results belong there. What do the authors think?

I'm open to increasing the score, especially since most changes I would like to see are textual. The most major comments from my perspective are the clarity that relates to Algorithm 1 and the missing theoretical motivation.

[1] Hüllermeier, Eyke, and Willem Waegeman. "Aleatoric and epistemic uncertainty in machine learning: An introduction to concepts and methods." Machine learning 110.3 (2021): 457-506.

---

> ### Author Response · Authors · 2025-11-21
>
> We would like to thank the reviewer for their thoughtful and constructive feedback. We have carefully considered the suggestions and address them in detail below. Since most concerns relate to the presentation, we will start with that first.
>
> ## **Presentation**
> > **Missing details of the algorithm**
>
> We thank the reviewer for pointing this out. To clarify, we have moved the algorithm to the main body and added a separate section explaining its procedure, including
> * All the components involved are explained.
> * The missing citation has been added (e.g., Value Iteration).
> * Belief update and epistemic uncertainty are elaborated on.
> * A comparison to previous works is made, highlighting that the key difference lies in our reward formulation.
>
> > **Typo** $\mathcal{E}_b$
>
> Thanks for pointing out this, we have revised it at **line 129**.
>
> > **Citations on uncertainty quantification**
>
> We thank the reviewer for pointing this out. Hüllermeier & Waegeman (2021) were cited in the first draft in the uncertainty quantification paragraph of the Related Work section. To ensure consistency, we have now added relevant citations in Section 3.1.
>
> > **Environment descriptions**
>
> For LazyChain, we additionally describe the optimal policy and per-step cost in the main text. Detailed descriptions of all environments are now provided in **Appendix B.1**.
>
> ## **Experiments**
> > **Improving PSRL Performance**
>
> That was also surprising to us, since PSRL has been shown to be an effective exploration strategy. However, it is worth noting that we ensured identical priors and modeling choices across Bayesian methods, differing only in their parameter values. Moreover, we used the same range of hyperparameters for tuning, thereby ensuring fairness (see details in **Appendix B.3**). We find that PSRL is less stable and fluctuates near convergence, potentially due to the high variance resulting from repeatedly sampling from the belief. Note that our success criterion is strict: the task must be successfully solved in consecutive episodes. If it fails in even one episode, it will not be considered fully solved. Additional evidence of the scaling issue of PSRL can be found in [1], which suggests that PSRL cannot reliably solve problems larger than size 10 on deterministic Deep Sea. Plots of cumulative regret for different problem sizes are shown in Figure 7.
>
> ## **Theoretical Motivation**
> Can you elaborate on what CAI refers to? I am not sure that I have defined it anywhere.
>
> ## **Limitation**
> > **Tabular setting**
>
> Yes, this work primarily focuses on the tabular setting, as indicated in the preliminaries, considering only finite state and action spaces. However, we believe our approach has the potential to extend to more complex tasks, given the availability of Bayesian inference and deep learning techniques. To make this clearer, we have added a sentence at the end of the conclusion section highlighting this limitation and suggesting directions for future research.
>
> ${}$
>
> [1] Luis, Carlos E., et al. "Model-based uncertainty in value functions." International Conference on Artificial Intelligence and Statistics. PMLR, 2023.

---

> ### Comment · Reviewer_kthm · 2025-11-22
>
> I thank the authors for their detailed reply. I am (almost entirely) satisfied, and looking forward to be able to increase the score to 8. Specifically:
>
> 1. Algorithms: the value iteration is solving the *belief* MDP (the MDP constructed over the belief space), is that correct? Just for clarity, please add this to the phrasing (unless it is already mentioned / explained previously in the paper, in which case I'm satisfied). Just to make it clear to the reader how exactly the process operates (even though I would agree with the authors that by now this process is rather standard).
>
> 2. Citation: I must have missed it. Well done.
>
> 3. LazyChain: I think the description still needs iteration. Remaining questions: Does the agent get a reward at either end? Or only the right? If it's either, why is the optimal policy go only and always to the right? Is it accurate to say that "the cumulative reward remains zero", or is it only zero for the optimal policy which reaches the goal? Why does the agent have to reach both ends of the chain, if the optimal policy is always go right? What is the myopia referred to here? The adversarial reward function (the cost of moving)? Are there terminal states (the end of the chain)? Please rewrite the description such that the answers to these questions are all clear. If you're running out of space, I would be satisfied with a very brief description followed by a more detailed description in the appendix. Currently in the appendix there is Chain, not LazyChain, and the description leaves room for these (or similar) questions.
>
> 4. PSRL: Well done. I'm satisfied.
>
> 5. Limitations: Perhaps I misunderstand (please correct me if this is the case), but I believe that this is not sufficiently accurate. Algorithm 1 appears to be running *exact* value iteration *on the entire belief MDP* in *every step in the environment* (or at least every period). This seems to me like almost by-design unextendable beyond small problems where the belief MDP can be *both* represented well and *fully solved* (surely intractable for any but toy problems?). The only direction I can see for extending this method beyond toy domains is by using sufficiently efficient approximate solvers (which to my knowledge, do not exactly exist?). This goes (in my opinion) well beyond "the availability of Bayesian inference and deep learning techniques".
>
> To emphasize: the existence of the limitation is a non-issue for me, by itself. It does not detract from the valuable theoretical work done here by the authors. I would merely like the phrasing and description of the limitation and scope of the contribution to be as clear as possible to the reader. Please let me know what you think, correct me if I misunderstand or discuss this more robustly in the paper if you agree.
>
> 6. Theoretical motivation: CAI - control as inference. Why is CAI chosen to formulate the epistemic uncertainty? and why is epistemic uncertainty chosen to be formulated in CAI in this way? This comment is not so much about the motivation itself (I will be satisfied with any), but more so about including it in the paper.
>
> 7. Nit: the colour used to mark new changes is a bit hard to see (for me, at least). Maybe use something with sharper contrast next time (for this rebuttal I wouldn't bother).

---

> ### Author Response · Authors · 2025-11-22
>
> Thank you for your valuable feedback! I would like to clarify a few misunderstandings.
>
> ## **Algorithm and Function Approximation**
> To be more precise, value iteration solves the MDP constructed ***from the belief*** rather than ***over the belief space***, where the former fixes a belief while the latter enumerates all possible beliefs. This distinction accounts for the significant computational difference. In fact, we have discussed this in **Preliminary 2.1**, where we motivate why we use a mean-MDP instead of a vanilla BAMDP.
>
> To provide a more concrete explanation, let us go through the process of how the MDP is constructed. At each time step, we obtain new observations $(r_t, s_{t+1})$ from the environment. These observations allow us to update the belief in closed form due to conjugacy; let us denote the updated belief as $b_{t+1}(\mathbf{w})$, which is a distribution over the model parameters $\mathbf{w}$, e.g., the transition probability vector $\mathbf{p}$ or reward parameters $\mu$ and $\sigma$.
>
> Once we have the belief, we compute the posterior predictive transition model $P_b$ and the epistemically guided reward $r_b^{\text{EUBRL}}$. Based on these, we construct a mean-MDP $(\mathcal{S}, \mathcal{A}, P_b, r_b^{\text{EUBRL}}, \gamma)$. We then solve this MDP using value iteration to obtain a new policy and interact with the environment again. The whole process repeats.
>
> The **core question** is: do we operate on the belief space? The answer is obviously **no**. Once the belief is obtained, we can immediately derive both the transition and reward models. This addresses the first question.
>
> Bearing this in mind, the extension to function approximation settings becomes clearer. We can maintain an approximate posterior $b$ over the full or partial parameters $\mathbf{w}$ of either the transition or reward model—for example, using a standard Bayesian neural network (BNN). Can we then obtain $P_b$ and epistemic uncertainty approximately? **Yes**. We can draw samples $\mathbf{w}_1, \mathbf{w}_2, \dots, \mathbf{w}_N$ from the approximate posterior and aggregate them as
>
> $$
> \hat{P}_b(s' \mid s, a) = \frac{1}{N} \sum_i P(s' \mid s, a, \mathbf{w}_i).
> $$
>
> But is this necessary? Not always, because $P_b$ is essentially a mixture distribution. We can use Monte Carlo estimates for planning, similar to Dreamer [1], by following a two-step sampling procedure $\mathbf{w} \sim b(\mathbf{w}), s' \sim P(s' \mid s, a, \mathbf{w})$. Estimating epistemic uncertainty can be done similarly, though it usually requires multiple samples from the belief. An open question remains: can we construct a well-calibrated epistemic uncertainty estimator that does not heavily rely on sampling? I believe these are fundamental questions for enabling scalable active exploration in more complex environments. We are currently working on this.
>
> ## **LazyChain**
> You may have missed the following page; please refer to **lines 810–832**, where we provide both visualizations and more detailed descriptions.
>
> ## **Control as Inference**
> Thanks for clarifying this. That's an excellent question. Put simply, we choose CAI to incorporate epistemic uncertainty into the agent's learning objective because we want to disentangle exploration from exploitation. Probabilistic graphical models provide a principled way to condition on additional hidden variables, e.g., $U$. This conditioning allows us to partition optimality $\mathcal{O}$ into two cases: one when the agent is uncertain, corresponding to epistemic uncertainty, and the other when it is certain, corresponding to reward estimates. Everything else then follows from our reward formulation—the epistemically guided reward. We believe we have stated this motivation briefly in **lines 137–139**.
>
> Lastly, we sincerely thank you for your invaluable suggestions and feedback, which have greatly improved the quality and clarity of our work.
>
> ${}$
>
>
> [1] Hafner, Danijar, et al. "Dream to control: Learning behaviors by latent imagination." ICLR, 2020.

---

> > ### Comment · Reviewer_kthm · 2025-11-22
> >
> > I thank the authors for the quick and informative reply.
> >
> > 1. Algorithm: I thank the authors for the additional explanation, it seems indeed that the misunderstanding is mine. If the authors are satisfied that after all recent changes the presentation is clear (it's always hard to produce super-coherent text in the short frame of the rebutal), I am satisfied, and I would also trust the authors to make further edits to the final version if necessary.
> >
> > 2. Function approximation, approximate posteriors etc.: It seems to me to be the standard exact / approximate Bayesian considerations. I am satisfied. Please make sure that the final version contains a brief discussion summarizing the authors' comments. Perhaps  it would be useful to rely / include some citations of existing "Bayesian" works that are exposed to the same challenges of exact -> approximate posteriors.
> >
> > 3. Limitations of value iteration as an "MDP solver": this remains unaddressed in the current draft and reply by the authors, is that correct?
> >
> > 4. Lazy chain: indeed I missed lines 810–832, which provide (some of the explanation) I was looking for. Thank you for correctly pointing that out. However, the description in these lines is still incomplete (or I misunderstand it): Do we terminate at the end of the chain? I dont believe this is specified. This also relates to the following question: why is right better than left? the return of the path from the center seems to be the same, to me. Finally, and perhaps more importantly - are there any challenges in this environment that are not fully covered by Deep Sea already? Does it give us any additional information on the performance of the approach? It seems to me that Deep Sea has the same adverserial reward function, possible stochasticity, and even stronger - the possibility to randomize the action labelings, preventing agents from being able to generalize over actions and resulting in a much harder problem. Please give the description of the environment both in the main text as well as the appendix another pass. I would say that the motivation is more important in the main text, and a clear complete description in the appendix (unless of course these comments are a result of a misunderstanding on my side).
> >
> > 5. Motivation for using CAI, and this choice of incorporating epistemic uncertainty into CAI: remains unaddressed as well, correct?
> >
> > I thank the authors again for their rebutal and look forward to further discussion.

---

> ### Author Response · Authors · 2025-11-22
>
> Thank you once again for the follow-up. We address the remaining concerns of the reviewer as follows.
>
> > **Limitations of value iteration as an "MDP solver"**
>
> Thanks for raising this again. However, in our previous response, we clarified that we are not solving a belief-MDP, which addresses the misunderstanding that we need to deal with a huge belief space. This is not the case: we use a fixed belief to construct both the transition and reward models and solve a mean-MDP using value iteration. We then counter the reviewer's point that the method is ***by-design unextendable beyond small problems*** with a tractable BNN example. It is worth noting that even in the field of Bayesian deep learning, there is no universal solution for uncertainty quantification. However, this does not falsify the possibility of extending our method beyond the current setting.
>
> > **Lazy Chain**
>
> For the terminal state, the agent will be reset to the middle of the chain if either end is reached. We thank the reviewer for pointing out this oversight and will revise it accordingly in the appendix.
>
> Regarding the second question, "why is right better than left," there may be a misunderstanding. The statement that ***the return of the path from the center seems to be the same*** is incorrect. For brevity, consider $\gamma = 1$: the left end gives a positive reward $r = N - 1$, leading to a return of $0$ for the initial state. In contrast, the right end gives a positive reward of $r = 2N - 1$, resulting in a return of $N$, which indicates that the right end is optimal.
>
> Regarding the third question about additional information provided by LazyChain, we note the following. First, LazyChain introduces myopia, meaning the agent can do nothing and receive no cost, but it is impossible to obtain higher rewards. Second, LazyChain hinders effective credit assignment: even when reaching the left end, the positive immediate reward provides no meaningful learning signal for states near the start. Third, for algorithms that cannot sustain exploration, it is easy to get stuck in multiple suboptimal states. Finally, LazyChain exhibits stronger stochasticity in transitions than DeepSea. At each step, there is a probability $p$ that the effect of actions will be flipped. Even if the agent follows the optimal policy (always turning right), there is a high chance that it will take an arbitrarily long trajectory to reach the right end. This can be modeled by a sequence of Bernoulli random variables:
>
> $\\\{ X_t \\\}_{t=1}^{\infty} \sim \text{i.i.d. Bernoulli}(1 - p)$,
>
> with $X_t \in \\\{+1, -1\\\}$ representing whether the action is intended. The probability that the first hitting time
>
> $\tau_{N} = \text{inf} \\\{ t \geq 1: \sum_{i=1}^t X_i = N \\\}$
>
> occurs within a fixed but moderate time $T$ decays exponentially in the balanced length $N$ or as a power of the probability $p$, making the problem technically more challenging.
>
> In practice, LazyChain uses $p=0.2$. DeepSea also has a probability of failing the "right" action without flipping its effect; however, this failure does not apply to other actions. Moreover, the stochasticity in DeepSea decays as the problem size grows, i.e., $p = \frac{1}{N}$, whereas in LazyChain it remains constant, implying that the larger the problem size, the more difficult the task becomes. Additionally, DeepSea is episodic, terminating exactly in $N$ steps, whereas LazyChain may take an arbitrarily longer time to explore the chain.
>
> In summary, LazyChain introduces new challenges, including myopia, difficulty in credit assignment, and higher degree of stochasticity in transitions, all of which require deep exploration to solve the task.
>
> > **CAI**
>
> This issue was addressed at the end of our last response.
>
> Finally, we will adopt the reviewer's suggestion on incorporating a discussion on the approximate setting.

---

> ### Comment · Reviewer_kthm · 2025-11-25
>
> I thank the authors for the detailed reply.
>
> 1. **CAI:** I see, I've missed it in your previous reply. Thank you for pointing lines 137-139, as well. I find the authors motivation both clever and a non-negligable part of the contribution. The description in lines 137-139 is a bit too concise in my opinion. Could the authors incorporate the more elaborate motivation they've provided in the rebutal into the paper (i.e. elaborate lines 137-139)? Specifically / additionaly, if the authors had additional ways they have considered to incoporate the uncertainty into CAI and decided not to follow up on them (/tried them and they didnt work well), I think this is also a valuable part of the contribution. If the authors find there is information they can include here but run out of space, I think that a more elaborate discussion in the appendix will address this point well, as well. I trust the authors to incorporate the relevant information, and view this point as addressed.
>
> 2. **Lazy chain:** (i) *terminals:* good. (ii) *length of path right / left:* good. (iii) *Doing nothing and receiving no cost:* isn't that the same as going in the wrong direction in deep sea and receiving no cost? (iv) *effective credit assignment:* isn't this the same as in any sparse-reward, single-goal environments, where the reward information must "travel" through the value function to the value of the starting states? i.e., the same as in DeepSea? (iv) *stochasticity:* I see, it is indeed a little different. Please make sure this information is included briefly in the main paper, and that the differences are detailed in the appendix (as motivation for including both environments, and to make clear to the reader the slight differences between them). I view this point as addressed.
>
> 3. **Limitation of an MDP solver:** Let's try an example. Say we apply EUBRL (/Algorithm 1) to Chess. For simplicity, let us imagine a hardcoded opponent (unknown to the EUBRL agent, of course), so the environment is a regular single-agent environment. Let us imagine we are able to accurately describe the full posterior over models, extract the mean MDP and update the posterior without any problems. Is line 182, "$\pi -\gets$ solve $\mathcal{M}$" tractable? It seems to me fundemntally intractable. As a sidenote, I note that other algorithms of a *somewhat* similar nature (AlphaZero [1] / MuZero [2] operate according to almost the exact same loop) this step *is* tractable, because instead of "solve $\mathcal{M}$" (intractable when the MDP is even a little bit hard?) they call "*policy improvement* using $\mathcal{M}$" (which is essentially always tractable).
>
> 4. **Limitation of uncertainty quantification**: I don't believe the challenge I'm referring to in the previous point has anything to do with uncertainty quantification. I also agree with the authors that uncertainty quantification is a major open problem, and while it could be relevantly mentioned in a challenges / limitations section for this paper, I view it as orthogonal to and not detracting from the authors' contributions.
>
> [1] Silver et al. "Mastering Chess and Shogi by Self-Play with a General Reinforcement Learning Algorithm." Science, 2018.
>
> [2] Schrittwieser et al. "Mastering atari, go, chess and shogi by planning with a learned model." Nature 2020.

---

> > ### Author Response · Authors · 2025-11-27
> >
> > We sincerely thank the reviewer for helping us improve our work. Before addressing their remaining concerns, we outline the changes made in the latest revision in response to the reviewer's suggestions, highlighted in ***pink***.
> >
> > * Expanded the discussion on the motivation of CAI, including PGMs visualization (**Appendix A** and **Figure 4**, linked at **line 139**).
> > * Added a discussion on function approximation (**Appendix B.3**, linked at **line 539**).
> > * Provided a more detailed description of LazyChain and a comparison with DeepSea (**lines 894-913**, linked at **line 431**).
> >
> > We now address the reviewer's remaining concerns.
> >
> > ### **LazyChain**
> > > **Do nothing**
> >
> > While they incur the same costs, we argue that straying from the correct path is fundamentally different from idling in place. In LazyChain, the optimal path can be arbitrarily extended with "do nothing" actions while still yielding the same return. This makes LazyChain more difficult because it admits many more suboptimal solutions. In contrast, the optimal path in DeepSea remains intact, as moving in the wrong direction does not obscure it. Moreover, doing nothing provides no learning signal, whereas moving in the wrong direction in DeepSea still contributes to model learning through collected data and can potentially yield rewards from the additional bottom-left cell in the stochastic variant.
> > > **Credit assignment**
> >
> > Yes, it indeed inherits the issue of credit assignment as in any sparse-reward environment, where exploration is the bottleneck even to find actions leading to the goal. However, additional factors exist in LazyChain. First, although turning left at the initial state is a necessary condition for reaching the left end, the cumulative rewards signal to the agent that this action contributes nothing, resulting in incorrect credit assignment. Secondly, "do nothing" can occur along any trajectory, making movement actions indistinguishable from inaction, which further complicates credit assignment.
> > ### **Limitation of an MDP solver**
> >
> > I see what you mean. You are correct: solving an MDP in high-dimensional and continuous state and action spaces is intractable. In practice, methods such as AlphaZero and MuZero replace this exact "solve" with approximate policy improvement (via MCTS and value function approximation), which is tractable. Similarly, EUBRL can be implemented using approximate solvers or rollout-based policy improvement to scale to complex domains. The exact "solve" in the pseudocode is meant for clarity and precision in theoretical analysis, not as a claim of tractability in general. We have elaborated on its practical implications in **lines 169–170**, which you may have missed.
> >
> > We would like to thank the reviewer once again for their valuable input in improving our work throughout the discussion and for enhancing its accessibility to a general audience. Please let us know if you have any further concerns.

---

> > > ### Comment · Reviewer_kthm · 2025-11-27
> > >
> > > I thank the authors for the details and comprehensive reply.
> > >
> > > 1. Motivation for CAI: Good.
> > >
> > > 2. Function approximation: Good.
> > >
> > > 3. LazyChain: Good. Please make sure to include this information in the paper / appendix.
> > >
> > > 4. **Limitation of an MDP solver**: We understand each other. I remain with one question / concern - from my perspective, the main contributions of the paper are the strong theoretical results. I presume that these results do not survive the replacement of an exact solver (ValueIteration), *which correctly identifies the optimal policy in the mean-MDP*, with a solver that isnt even exactly approximate, it merely provides policy improvement from a prior policy (in a previous mean-MDP), towards the information in this mean MDP. Of course, if this is not the case, please correct me. For the above reason, it seems to me that it is not the case that "For large-scale settings, approximate methods such as tree search may be required" (lines 168-169), as much as "this family of approaches, and their theoretical guarentees are fundamentally intractable in all but very small and easy to solve MDPs. In order to extend this family of methods and / or their guarentees outside of this space of problems, additional research is necessary to connect the theory driving this family of methods with approximate solvers or *policy improvement* based planning".
> > >
> > > Alternatively, I would like to see experiments that show that this approach works fine / great with approximate planners. It is very late in the rebutal period, and I imagine that the authors do not already have these results to hand as they would otherwise probably already include them. I will be completely satisfied with a clear description of this limitation (or if you can convince me that it's not as much of a limitation as it seems to me), and leaving these results to future work.
> > >
> > > This comment also relates to the content of the second paragraph of Appendix B.3 ("the planning"). It's not clear to me if the discussion there is aiming to address this concern.
> > >
> > > A suggestion that could help connect the approximate line to the exact: There is this recent work [1], which incorporates essentially the idea that the authors propose in this discussion, as I understand it, to MCTS and AlphaZero. Instead of solving the model, the method explicitly maintains an exploration policy which approximates the actions that would be chosen by UCB exploration. This policy is iteratively "improved" in the direction of the largest UCB in the current model (/"mean MDP") using planning with (Epistemic) MCTS.
> > > Perhaps it could serve the authors as a citation motivating that although this paper does not provide experiments with approximate solvers, similar ideas have been explored and been succesful in the past (albeit with the more standard UCB exploration rather than the author's exact approach).
> > >
> > > This is the last concern from my side, and upon its address, I will increase the score to 8.
> > >
> > > [1] Oren, et al. "Epistemic Monte Carlo Tree Search." ICLR 2025.

---

> ### Author Response · Authors · 2025-11-27
>
> We thank the reviewer for their prompt feedback and valuable input. We now address the reviewer's final concern.
>
> ### **The Concern Restated**
> The crux of the concern is ***whether the strong theoretical results of the paper, which rely on an exact (or nearly) MDP solver, still hold if it is replaced with an approximate one***. Our answer is that the theoretical results remain valuable in the approximate setting. We would like to begin by discussing several lines of work that transition from exact to approximate.
>
> ### **From Exact to Approximate**
> The first method is PSRL, which relies on an exact MDP solver to achieve a strong regret guarantee [1-3]. It has later been shown to be effective in high-dimensional and continuous state and action spaces [4–6].
>
> The second method is UCRL, which also requires an exact MDP solver to solve the MDPs constructed from confidence intervals [7]. It has since dominated all optimism-based methods for exploration that use additive bonuses.
>
> There are more examples that could be mentioned, but the main takeaway is that strongly theoretically grounded methods have the potential to be effective in more complex environments, as illustrated above.
>
> ### **Theoretical Justification**
> Let us scrutinize the argument from a theoretical perspective. We argue that our theoretical results remain meaningful in the approximate setting. This can be seen by following the regret decomposition, except that we now introduce an additional policy, $\hat{\pi}_t$, which approximately solves the mean MDP. The per-step regret (suboptimality gap) can then be expressed as follows:
>
> $$
> V^{\star}(s) - V^{\hat{\pi}_t}(s) = V^{\star}(s) - V^{\pi_t}(s) + \underbrace{V^{\pi_t}(s) - V^{\hat{\pi}_t}(s)} _{ \textbf{Approximation Error} },
> $$
>
> As shown, the per-step regret can be decomposed into two components: the first part corresponds to the results established in the paper, while the second part captures the quality of the approximate solver. If a reasonably good approximation is available, we can derive similar regret and sample complexity bounds, albeit with an additional term reflecting the approximation error.
>
> This property is appealing because, given the same solver, our method will always outperform alternative approaches. It also implies that EUBRL can be integrated with existing solvers, such as tree search-based or rollout-based methods. The planning part discussed in the section on function approximation provides viable options.
>
> ### **Citation on Related Work**
> Lastly, regarding the suggestion to cite the work mentioned: as far as we can tell, it operates in a purely frequentist setting and measures epistemic uncertainty using the variance of the empirical mean. This departs from the standard notion of epistemic uncertainty, which is intended to quantify model uncertainty. Moreover, because a similar tree-search approach [8] is already covered in the paper, we believe it is not necessary to include this additional citation.
>
> ${}$
>
> [1] Osband, Ian, Daniel Russo, and Benjamin Van Roy. "(More) efficient reinforcement learning via posterior sampling." Advances in Neural Information Processing Systems 26 (2013).
>
> [2] Osband, Ian, and Benjamin Van Roy. "Why is posterior sampling better than optimism for reinforcement learning?." International conference on machine learning. PMLR, 2017.
>
> [3] Agrawal, Shipra, and Randy Jia. "Optimistic posterior sampling for reinforcement learning: worst-case regret bounds." Advances in neural information processing systems 30 (2017).
>
> [4] Osband, Ian, et al. "Deep exploration via bootstrapped DQN." Advances in neural information processing systems 29 (2016).
>
> [5] Fan, Ying, and Yifei Ming. "Model-based reinforcement learning for continuous control with posterior sampling." international conference on machine learning. PMLR, 2021.
>
> [6] Sasso, Remo, Michelangelo Conserva, and Paulo Rauber. "Posterior sampling for deep reinforcement learning." International Conference on Machine Learning. PMLR, 2023.
>
> [7] Auer, Peter, Thomas Jaksch, and Ronald Ortner. "Near-optimal regret bounds for reinforcement learning." Advances in neural information processing systems 21 (2008).
>
> [8] Guez, Arthur, David Silver, and Peter Dayan. "Efficient Bayes-adaptive reinforcement learning using sample-based search." Advances in neural information processing systems 25 (2012).

---

> > ### Comment · Reviewer_kthm · 2025-11-27
> >
> > I thank the authors for their quick and excellent  reply.
> >
> > Please make sure to include this discussion and above citations in the paper.
> >
> > I am satisfied that this concern is addressed, and I increase the score to 8.
> >
> > In regards to the citation of Epistemic MCTS:
> >
> > The epistemic uncertainty is explicitly modelled as the variance in a posterior over possible models (see Equation 5, the UBE), and analyzed and propagated from this perspective (see Equations 7 and 8). I do not read this as a frequentist approach.
> > To me this method seems like a direct alternative to EUBRL, like UCRL, that is based directly in approximation of an "optimal UCB exploration policy" with respect to the variance in a distribution of models - as far as we choose to observe the policy improvement of MCTS as an approximation to the optimal policy. For that reason, it seems to me to remain relevant to include in the related work / in the area that discusses the feasibility of this family of methods with "approximate" solvers.

---

> > > ### Author Response · Authors · 2025-12-02
> > >
> > > We are pleased that all concerns have been satisfactorily addressed and sincerely thank the reviewer for increasing the score.
> > >
> > > In response to the reviewer's request, we have revised the manuscript to include a discussion of the extensibility of our theoretical results in the approximate setting, detailed in **Appendix B.3**, as highlighted in ***green***.
> > >
> > > Regarding the work recommended for citation, it is worth clarifying that UBE is a Bayesian approach, like PSRL, explicitly maintaining a belief over models, which gives rise to variance stemming from model uncertainty. In contrast, although E-MCTS claims to use UBE to estimate "epistemic uncertainty," it does not maintain a belief over models. Instead, it relies on the variance of the **empirical mean** of rewards, propagated via UBE, as shown in Section 3.1 in the definition of $\hat{R}(s, a)$. This differs from our perspective, in which epistemic uncertainty is a measure of lack of knowledge, rooted in the Bayesian framework. E-MCTS is better understood as a frequentist method that modifies guided sampling for action selection compared to traditional MCTS. Therefore, it is more aligned with that particular direction and is orthogonal to our framework and its derivatives. For these reasons, we believe citing this work is unnecessary.

---

### Official Review · Reviewer_Sc6X · 2025-10-31

**Soundness:** 3
**Presentation:** 3
**Contribution:** 3
**Rating:** 6
**Confidence:** 3

**Summary:**

This paper proposes EUBRL, a Bayesian RL algorithm that steers exploration using epistemic uncertainty. At a high level, the per-step reward is interpolated using a learned "probability of uncertainty," $P(U=1 \mid s, a)$, which balances exploitation (the current mean reward estimate) against exploration (an intrinsic term derived from epistemic uncertainty). The authors prove nearly minimax-optimal regret and sample-complexity bounds for infinite-horizon discounted MDPs under a class of priors. Experiments on Chain, Loop ,DeepSea, and a new LazyChain benchmark compare EUBRL to several baselines and suggest strong exploratory behavior.

**Strengths:**

1.To the best of my knowledge, this is the first work to convert epistemic uncertainty into an explicit guidance weight $P(U=1 \mid s, a)$. This idea is novel, naturally decouples exploration from uncertain reward estimates, and provides an adaptive interpolation weight of the reward signal.

2.Theoretical guarantees are strong.  The paper gives (i) a regret bound $\tilde{O}(\sqrt{S A T} /(1- \gamma)^{1.5}+S^2 A /(1-\gamma)^2$ ) that matches known lower bounds when $T$ is large enough, and (ii) a sample-complexity bound that matches lower bounds for small $\varepsilon$.

3.The paper formalizes a class
$\mathcal{C}$ of decomposable/weakly-informative priors and shows nearly-minimax bounds for uniform bounded priors.

4.The experimental section covers both deterministic and stochastic settings with many random seeds, which helps demonstrate robustness.

**Weaknesses:**

1.The algorithmic description is quite high-level. More concrete details and examples would help reproducibility-for instance: how $\mathcal{E}(s, a)$ is computed in practice; how the on-policy estimate $P(U=1| s, a)= \mathcal{E}\_{b}/ \mathcal{E}\_{\max}$ is formed; and how $\mathcal{E}_{\max }$ is chosen.

2.Some notation and concepts in the main text need clearer definitions. For example: what is the role of $w$ in Section 3.1? How is a "prior" specified precisely (i.e., prior over what objects/parameters)? It would also help to include concrete examples illustrating Definition 1.

**Questions:**

1.Details in Algorithm 1: How do you compute $\mathcal{E}(s, a)$ exactly in the tabular/prior choices you study? How is $P(U=1| s, a)=\mathcal{E}\_{b}/ \mathcal{E}\_{\max}$ estimated on-policy, and how is $\mathcal{E}_{\text {max }}$ selected (global constant, rolling maximum, or theoretical bound)? Do you require assumptions on $f$ and $g$ ? How is Valuelteration(b) implemented under the updated belief?

2.Theorem 3 suggests $\eta$ is important. How do you choose $\eta$ across tasks, and how sensitive are results to it?

3.$\widetilde{V}^t(s)$ is introduced to ensure quasi-optimism. Could you give its precise definition and a short intuition for how it drives the regret decomposition?

Suggestions:
1. If some definitions or derivations are omitted due to space, please add precise references to the appendix, so readers can quickly locate the formal statements and implementation details.

---

> ### Author Response · Authors · 2025-11-21
>
> We sincerely thank the reviewer for their insightful and constructive comments. We address the reviewer's concerns in detail below.
>
> ## **Notations and Concepts**
> > **What is inference target?**
>
> We thank the reviewer for pointing out this. $\mathbf{w}$ refers to the model parameters. For example, it can be the transition probability vector $\mathbf{p}$, for which we can have a Dirichlet prior $\textbf{Dir}(\mathbf{p}; \boldsymbol{\alpha})$; or the reward location $\mu$ and scale $\sigma$, for which we can have a Normal-Gamma prior. We have made this clear in the text at **line 120-121**.
>
> > **Examples on Definition 1**
>
> **Corollary 1** provides one concrete example of this definition, which achieves nearly minimax-optimality. It is worth noting that the majority of priors (e.g. Dirichlet, Normal-Normal, Normal-Gamma) are ***decomposable***, although their rate of epistemic uncertainty may differ. A detailed analysis is provided in **Appendix G**.
>
> > **Composition functions**
>
> While we focus on variance or mutual information as epistemic uncertainty measures, the abstraction provided by the composition of $f$ and $g$ enables the use of other viable alternatives.
>
> ## **Algorithm**
> > **How is epistemic uncertainty calculated?**
>
> Epistemic uncertainty can be expressed in closed form in the tabular setting. The formulas for all priors are presented in **Appendix G.2** and mutual information in **Appendix A.2**.
>
> > **How is maximum epistemic uncertainty calculated?**
>
> Typically, $\mathcal{E}_{\text{max}}$ is determined by the priors and generally non-increasing. However, we note that when epistemic uncertainty depends on the sample variance, it may not be strictly so. Therefore, we use a rolling maximum in practice to ensure that $P_U$ remains well-defined. A description has been added in **line 161**, and a detailed discussion is provided in **Appendix A.2**.
>
> > **Details on value iteration**
>
> We apply the standard value iteration [1] to the MDP constructed from the posterior predictive $P_b$ and the epistemically guided reward $r_b^{\text{EUBRL}}$, where the only differences lie in the transition and reward models. A detailed description is added in **Section 3.3**.
>
> > **Scaling factor**
>
> From a theoretical perspective, both the positive (**Theorem 3-4**) and negative (**Theorem 5**) results suggest that the scaling factor should not be too small. This is reasonably intuitive, since if it is too small, there is not enough "driving force" to explore, and the agent can easily get stuck in suboptimal actions. In practice, the scaling factor $\eta$ is tuned via a line search for each algorithm, where applicable, typically in proportion to either the maximum reward or the size of the state space. We found that EUBRL is more robust to value changes. A more detailed discussion of hyperparameters can be found in **Appendix B.3**.
>
> ## **Theoretical Intuition**
> > **On auxiliary value function**
>
> This is a very good point. We have supplemented the definitions and explained the purpose of each term in lines **223–228**. In addition, I would like to provide some intuition for why it is important: we found that our reward formulation is more aggressive and may not satisfy the requirements for proving quasi-optimism on its own. However, if we can identify another value function (e.g., $\widetilde{V}^t$) that satisfies quasi-optimism while remaining sufficiently close to the original value function $V^t$, then we are done. For more technical details, please refer to **Appendix E.1-2**.
>
> ${}$
>
> [1] Sutton, Richard S., and Andrew G. Barto. Reinforcement learning: An introduction. Vol. 1. No. 1. Cambridge: MIT press, 1998.

---

> > ### Comment · Reviewer_Sc6X · 2025-11-28
> >
> > Thanks for authors' detailed reply.  I am willing to increase the score to 8.

---

### Official Review · Reviewer_cKEm · 2025-10-31

**Soundness:** 3
**Presentation:** 4
**Contribution:** 3
**Rating:** 8
**Confidence:** 3

**Summary:**

The authors propose the Epistemic Uncertainty directed Bayesian Reinforcement Learning (EUBRL) algorithm which achieves nearly minimax-optimal regret and sample complexity in infinite-horizon Markov Decision Processes (MDPs). EUBRL directs RL exploration by utilising probabilistic inference to model epistemic uncertainty as part of the agent’s objective (i.e., adaptively weighting the mean reward and uncertainty term based on the probability of uncertainty). Nearly minimax-optimal regret bounds and sample-complexity guarantees are established for the class of priors that are decomposable, or weakly-informative, and whose rate of epistemic uncertainty is $\mathcal{O}(1\sqrt{n})$. Empirically, EUBRL is shown to outperform frequentist (RMAX, MBIE-EB), sampling-based (PSRL, BOSS), optimism-based Bayesian (BEB, VBRB), and classical Bayesian (BEETLE, Mean-MDP) baselines across tasks with sparse rewards, long horizons and stochasticity (Chain, Loop, DeepSea, and the newly introduced LazyChain).

**Strengths:**

The paper is eloquently written and introduces a novel theoretical proof which, for the first time (to the best of my knowledge also), achieves nearly minimax-optimal sample complexity in infinite-horizon discounted MDPs without assuming a generative model. This result improves on He at al. 2021 which shows nearly minimax-optimal regret but doesn’t extend to sample complexity. The theoretical results are backed up by convincing empirical results (using multiple seeds, reporting standard errors etc) covering a wide range of appropriate baselines (including frequentist and Bayesian), three environments from the literature, and newly introduced fourth task which targets algorithm “myopia”. In all tasks EUBRL is shown to improve upon prior works.

Overall I think the claims in the paper are both significant and well-supported both theoretically and empirically.

**Weaknesses:**

Towards the goal of disentangling exploration and exploitation, the evidence could be strengthened by e,g., considering an ablation (see questions).

The accessibility of the paper to a wider audience could also benefit from adding short intuitive summaries after key lemmas in the appendices.

**Questions:**

Would it be possible to compare EUBRL to a variant that uses the exact same uncertainty estimates but applied as an additive bonus? I think such a result would help further empirically support the benefits of disentangling exploration and exploitation versus improved UQ.

---

> ### Author Response · Authors · 2025-11-21
>
> We sincerely thank the reviewer for their positive assessment of our arguments and contributions. We address the main concern—an ablation on the additive bonus—below.
>
> As discussed in the paper, VBRB employs an ***additive bonus*** of variance similar to ours. Therefore, it serves as an ideal baseline for evaluating the advantages of the epistemically guided reward and the benefits of disentangling exploration from exploitation. We find that VBRB is less sample-efficient and does not scale with problem size as gracefully as EUBRL, and it can sometimes be volatile, especially for the problem size of 100 in the stochastic LazyChain. It is also worth noting that VBRB’s reward is motivated by Chebyshev’s inequality for theoretical purposes rather than being rooted in epistemic uncertainty.
>
> In addition, to better elaborate the key lemmas and theorems, we have revised the theoretical section, clarifying the motivations, explanations, logical flow, and resulting consequences.

---

### Official Review · Reviewer_12wQ · 2025-11-01

**Soundness:** 2
**Presentation:** 2
**Contribution:** 2
**Rating:** 4
**Confidence:** 3

**Summary:**

The paper proposes a novel Bayesian reinforcement learning algorithm, EUBRL, to solve the explore-exploit dilemma, especially in environments with sparse rewards. The paper argues that common "optimism-based" exploration (adding an uncertainty bonus) is flawed because it can magnify errors when the agent's reward estimates are unreliable. EUBRL uses an "epistemically guided reward." This is a weighted average that exploits (uses the estimated reward) when the agent is confident, but explores (uses uncertainty itself as an intrinsic reward) when the agent is uncertain. Theoretically, they show that EUBRL achieves nearly minimax-optimal regret and more importantly sample efficiency which hasn't been done in prior work. Empirically, they show that their method leads to imporved sampled efficiency on different tasks.

**Strengths:**

1. The paper tackles the fundamental exploration–exploitation dilemma in reinforcement learning by introducing a novel approach termed “epistemic guidance.” The method is well-motivated, and its effectiveness is supported through both rigorous theoretical analysis and comprehensive empirical evaluation.

2. The paper argues that adding an exploration bonus to the reward estimate is a flawed way to do exploration in BAMDPs because the reward estimate can be highly uncertain and can result in a poorly specified reward. Instead they propose to weigh the two rewards (reward estimate and the exploration bonus) by the uncertainty associated with the state-action pair. If the agent is uncertain it uses the exploration bonus if the agent is certain it uses the reward estimate. The paper also proves that this method achieves nearly minimax-optimal regret in both regret and sample complexity.

**Weaknesses:**

1. The paper's analysis is confined to discrete state-action spaces, and it does not address the challenges of integrating its method with deep function approximation. The reliance on maintaining an explicit Bayesian posterior is computationally intractable for the high-dimensional environments where deep RL is typically applied. Therefore, it is unclear how the 'epistemically guided reward' could be effectively approximated to improve sample efficiency in practical deep RL algorithms.

2. The paper suffers from a separation of theory from implementation. While the main text provides a compelling theoretical justification for the "epistemically guided reward," it lacks a clear, procedural description of the full algorithmic loop. To understand precisely how the belief posterior is updated after each transition and how the new policy is extracted, the reader is required to hunt for the algorithm in the appendix.

3. The adaptive weighting scheme is particularly vulnerable to the degeneracy of the uncertainty estimates. If uncertainty degenerates to zero, the agent will become fully exploitative.

4. Similar to the last point, a mis-specified prior will also impact this algorithm a lot more than standard exploration bonus algorithms.

**Questions:**

My questions are based on the practicality of the algorithm for more realistic applications:

Exploration bonuses have been shown to work with Deep Reinforcement learning algorithms. Do the authors think that their method can be deployed with a neural-network based policy in a model-based setting.

1. The paper claims "superior scalability" but only demonstrates it in tabular, low-state-space environments (DeepSea). Isn't this claim misleading, as the algorithm's tabular, model-based nature makes it computationally intractable for the large-scale, high-dimensional problems where scalability is actually needed?

3. Since maintaining a full Bayesian posterior is impossible, which practical proxy for uncertainty would work with this method? For example, would you use the variance from a deep ensemble or a novelty score from a pseudo-count method?

4. Could this "guided reward" concept be adapted to improve sample efficiency in a model-free deep RL algorithm (like DQN or SAC), or is it fundamentally tied to a model-based approach?

---

> ### Author Response · Authors · 2025-11-21
>
> We thank the reviewer for their insightful questions and constructive feedback. Below, we address the reviewer's concerns in detail.
>
> ## **Function Approximation**
> > **Scalability**
>
> We recognize that the term “scalability” might be misleading if interpreted as applicability to large-scale neural network-based problems. However, as clearly stated at the beginning of the Experiments section, this refers to the scalability of different metrics with respect to problem size. It examines how an algorithm behaves in larger state spaces and whether it can sustain exploration, which is often hindered by increasing sparsity. This contrasts with previous Bayesian RL approaches, which are limited to small-scale tasks (e.g., Chain), and with some theoretical works that, although aimed at improving exploration, do not provide any empirical evaluation.
>
> > **Integration with Deep RL**
>
> First and foremost, we would like to clarify that this work primarily focuses on the tabular setting, as indicated in the preliminaries, considering only finite state and action spaces. This allows us to study the fundamental limits of exploration in RL within a clean setting, isolating the compounding effects of function approximation. While our work is not intended for immediate deployment with deep function approximators, we believe that the conceptual idea of epistemically guided reward could inspire future research.
>
> - **Approximate Posterior**
> For example, one could integrate Bayesian deep learning into our framework (e.g., deep ensembles [1], Bayes by Backprop [2], or MC dropout [3]) to approximate epistemic uncertainty. Several works have also shown that it is possible to maintain an approximate posterior over a world model (e.g., using Gaussian processes [4]) for active exploration in the high-dimensional and continuous state and action spaces. However, their computational efficiency and the quality of uncertainty calibration are still far from ideal. Therefore, scalable epistemic uncertainty estimation and efficient Bayesian planning with function approximation remain open and promising directions for future research, as noted in the Conclusion.
>
> - **Model-free RL**
> This is a very good question. I would say no, it doesn’t need to be tied to a model-based approach. Since our framework is a general recipe and epistemic uncertainty naturally arises from Bayesian inference, one could have different inference targets and probabilisitc interpretations. For example, it could be integrated with distributional RL [5], or simply used to infer the epistemic uncertainty in a policy or value function.
>
> ## **Algorithm**
> > **Missing details**
>
> We thank the reviewer for pointing this out. To clarify, we have moved the algorithm to the main body and added a separate section detailing its procedure, including the belief update, posterior predictive, epistemic uncertainty, and policy optimization.
>
> > **Prior Misspecification**
>
> This is a very good catch. The agent can indeed become exploitative, as identified in **Theorem 5**. However, this occurs only under a caveat: it becomes problematic ***only if*** both the prior is heavily misspecified and the scaling factor is small. In contrast, prior misspecification is no longer an issue when the scaling factor is appropriately chosen, as formalized in **Theorem 4** and **Corollary 1**. Most importantly, **Corollary 1** provides a ***worst-case*** guarantee: regardless of how severely the prior is misspecified, EUBRL still achieves near-optimal regret and sample complexity.
>
> From a practical standpoint, all Bayesian RL algorithms may be affected by prior misspecification. Hierarchical priors can help alleviate this issue, as we adopt and detail in **Appendix A.1**. Weakly informative priors are also beneficial, as they allow the belief to be incorrect without being overly confident, making it easier for observations to correct the belief.
>
> ${}$
>
> [1] Lakshminarayanan, Balaji, Alexander Pritzel, and Charles Blundell. "Simple and scalable predictive uncertainty estimation using deep ensembles." Advances in neural information processing systems 30 (2017).
>
> [2] Blundell, Charles, et al. "Weight uncertainty in neural network." International conference on machine learning. PMLR, 2015.
>
> [3] Gal, Yarin, and Zoubin Ghahramani. "Dropout as a bayesian approximation: Representing model uncertainty in deep learning." international conference on machine learning. PMLR, 2016.
>
> [4] Iten, Klemens, and Andreas Krause. "Scalable and Efficient Exploration via Intrinsic Rewards in Continuous-time Dynamical Systems." The Exploration in AI Today Workshop at ICML 2025. 2025.
>
> [5] Bellemare, Marc G., Will Dabney, and Rémi Munos. "A distributional perspective on reinforcement learning." International conference on machine learning. PMLR, 2017.

---

### Author Response · Authors · 2025-12-02
**General Response**

**Dear Reviewers, ACs, and SACs**,

${}$

### **Preface**

We sincerely appreciate the constructive comments and insightful questions from the reviewers, which help us continue improving our paper throughout the discussion.

In particular, we are grateful for the reviewers' positive feedback, highlighted as follows:

- Epistemic guidance is novel and well-motivated (**`Reviewers 12wQ, Sc6X`**).
- The theoretical results are significant (**`unanimously noted`**).
- Effectiveness of the approach is well-supported empirically (**`unanimously noted`**).

Despite the regrettable OpenReview incident, we remained actively engaged with the reviewers, addressing their concerns and adopting their suggestions, especially those of **`Reviewer kthm`**. We are grateful to have received an average score of 7 (from 8, 6, 6, 4 to 8, 8, 8, 4); in particular, both **`Reviewer Sc6X`** and **`Reviewer kthm`** increased their scores by 2 points. Although **`Reviewer 12wQ`** did not participate in the discussion, we note that their initial review focused on aspects beyond the scope of our setting, misinterpreted *scalability*, and overlooked the theoretical result on prior misspecification. All of these points are clearly stated and studied in the paper.

### **Our Contributions**
To better understand our work, we would like to briefly outline the contributions we have made:

- $\texttt{EUBRL}$, a Bayesian RL algorithm, achieves principled exploration with epistemic uncertainty.
- $\texttt{EUBRL}$ achieves nearly minimax-optimal regret and sample complexity aross a range of settings.
- $\texttt{EUBRL}$ exhibits strong exploration capabilities across various tasks, improving sample efficiency.

Why is this significant? First, this work represents the first systematic study of epistemic uncertainty in Bayesian RL. Rooted in Bayesian inference, epistemic uncertainty is inherently rich and can provide diverse intrinsic motivations for unsupervised exploration. Secondly, EUBRL is the first algorithm ever to achieve both nearly minimax-optimal regret and sample complexity across a wide range of settings, including finite-horizon episodic and infinite-horizon discounted MDPs, under both frequentist and Bayesian frameworks. In contrast, previous works typically focus on either a single metric or a single setting. This demonstrates not only stronger evidence of EUBRL's exploration capabilities but also broader applicability to both episodic and continuous learning tasks, including those leveraging informative priors. Moreover, we empirically validate the effectiveness of our approach across diverse tasks, contrasting with prior Bayesian RL methods that are limited to small-scale problems and some theoretical works that, while aiming to improve exploration, provide no empirical evaluation.

### **Summary of Revision**
In response to reviewers' questions and concerns, we have revised the paper based on their suggestions, improving clarity, breadth of discussion, motivation, and experimental details. We have made three major revisions, highlighted in three colors in the following order: dark blue, pink, and green. The updates are summarized as follows:

- Moved the algorithm to the main body in **Section 3.3**, with additional elaboration in **Appendix B** (**`Reviewers 12wQ, Sc6X, kthm`**)
- Polished the theoretical section for better flow and clarity in **Section 4** (**`Reviewers cKEm, Sc6X`**)
- Detailed experimental setups and descriptions in **Appendix C** (**`Reviewer kthm`**)
- Expanded discussion on motivation of probabilistic inference in **Appendix A** (**`Reviewer kthm`**)
- Provided a discussion on function approximation in **Appendix B.3** (**`Reviewers 12wQ, kthm`**)
- Miscs: corrected typos, clarified the inference target, added pointers to the corresponding appendix, and anticipated future works. (**`Reviewers Sc6X, kthm`**)

Lastly, we thank you once again for your time and dedication.

${}$

Best regards,

Authors

---

### Meta-Review · Area_Chair_F42x · 2026-01-08

**Summary:**

Reviewer 12wQ's  (score 4)  main concerns were that the analysis/claims are confined to discrete/tabular state-action spaces and that the practicality of the approach in more realistic settings is unclear; they explicitly questioned how the method would translate to deep RL and how posterior/uncertainty estimation would be done approximately (e.g., via ensembles or pseudo-count–style novelty) while still supporting the claimed “scalability.” The reviewer also raised a concrete failure-mode concern: the adaptive weighting can be vulnerable if uncertainty degenerates (leading to fully exploitative behavior), and a mis-specified prior may impact this method more than standard bonus approaches.

Reviewer cKEm was overall positive (score 8) but still requested stronger evidence around specific design choices, e.g., strengthening the empirical case with an ablation related to the additive bonus, and also suggested improving accessibility via more intuitive explanations for technical results.

Reviewer Sc6X  (score 6) mentioned presentation/clarity issues, stating the algorithmic description is high-level and that several definitions/notation details need to be clearer.

Reviewer kthm (score 6) emphasized missing theoretical motivation for specific design choices (e.g., why epistemic uncertainty is formulated in the chosen way).

**Reviewer Concerns:**

In response to Reviewer 12wQ's, the rebuttal clarifies that “scalability” may be misleading in the deep-RL sense and reframes the claim more carefully. Reviewer cKEm’s main request was for an ablation regarding the additive bonus; the rebuttal states it addresses this concern directly and motivates the role of the additive bonus relative to epistemic uncertainty. Responding to Reviewers Sc6X and kthm,  the rebuttal explicitly acknowledged missing-detail concerns, noting that the algorithm was moved to the main paper and that additional explanation/procedure details and missing citations were added.

**Reviewer Scores:**

Two reviewers who initially gave a score of 6 have already indicated that they will increase their score to 8. The reviewer who gave 4 is very likely to increase the score to 6 as most of the concerns are addressed.

---

### Decision · Program_Chairs · 2026-01-26

Accept (Poster)